# Modeling tissue-specific *Drosophila* metabolism identifies high sugar diet-induced metabolic dysregulation in muscle at reaction and pathway levels

Sun Jin Moon[1] ✉, Yanhui Hu [1], Monika Dzieciatkowska [2], Ah-Ram Kim [1], John M. Asara [3,4], Angelo D'Alessandro [2] & Norbert Perrimon [1,5] ✉

Individual tissues perform highly specialized metabolic functions to maintain whole-body metabolic homeostasis. Although *Drosophila* serves as a powerful model for studying human metabolic diseases, modeling tissue-specific metabolism has been limited in this organism. To address this gap, we reconstruct 32 tissue-specific genome-scale metabolic models (GEMs) by integrating a curated *Drosophila* metabolic network with pseudo-bulk single-nuclei transcriptomics data, revealing distinct metabolic network structures and subsystem coverage across tissues. We validate enriched pathways identified through tissue-specific GEMs, particularly in muscle and fat body, using metabolomics and pathway analysis. Moreover, to demonstrate the utility in disease modeling, we apply muscle-GEM to investigate high sugar diet (HSD)-induced metabolic dysregulation. Constraint-based semi-quantitative flux and sensitivity analyses identify altered NAD(H)-dependent reactions and distributed control of glycolytic flux, including GAPDH. This prediction is further validated through in vivo $^{13}$C-glucose isotope tracing study. Notably, decreased glycolytic flux, including GAPDH, is linked to increased redox modifications. Finally, our pathway-level flux analyses identify dysregulation in fructose metabolism. Together, this work establishes a quantitative framework for tissue-specific metabolic modeling in *Drosophila*, demonstrating its utility for identifying dysregulated reactions and pathways in muscle in response to HSD.

Multicellular organisms consist of tissues that perform highly specialized metabolic functions[1]. As such, analyzing tissue-level metabolism is critical to delineate the complex metabolic interplay among tissues and understand organismal physiology[2]. However, direct measurements of tissue-specific enzyme levels or fluxes are often unavailable.

Instead, mRNA levels have served as a proxy to reconstruct context-relevant or tissue-specific genome-scale metabolic models (GEMs)[3–5], enabling quantitative metabolic network analyses. In fact, numerous computational algorithms such as mCADRE[6], tINIT[7,8], CORDA[5], and MERGE[4], have been developed and applied for reconstruction of

$^1$Department of Genetics, Blavatnik Institute, Harvard Medical School, Boston, MA, USA. $^2$Department of Biochemistry and Molecular Genetics, University of Colorado Anschutz, Aurora, CO, USA. $^3$Division of Signal Transduction, Beth Israel Deaconess Medical Center, Boston, MA, USA. $^4$Department of Medicine, Harvard Medical School, Boston, MA, USA. $^5$Howard Hughes Medical Institute, Harvard Medical School, Boston, MA, USA. ✉e-mail: sunjin_moon@hms.harvard.edu; perrimon@genetics.med.harvard.edu

tissue-specific GEMs across complex organisms from *C. elegans*[4] to *Homo sapiens*[6]. Yet, tissue-specific metabolic models in *Drosophila melanogaster* are unavailable, limiting quantitative assessments of metabolic processes at tissue-level in this organism[9,10].

The high conservation of metabolic genes and functionally analogous organs between humans and fruit flies, together with the short generation time, lifespan, and availability of numerous genetic and analytical tools for *Drosophila*, has made flies an excellent organism in which to model human metabolic diseases[11,12]. Various mechanisms underlying dysregulated metabolism associated with type 2 diabetes[13,14], aging[15], and cancer[16] have been elucidated in flies. Moreover, several generic GEMs, such as *FlySilico*[17], *iDrosophila*[18], and *Fruitfly1*[19], have recently been developed, enabling systems-level evaluations of metabolic networks in this organism. However, they lack the resolution needed for tissue-specific metabolic analysis.

While GEMs are powerful computational tools for analyzing metabolic network structures and simulating fluxes, experimental techniques such as metabolomics, ¹³C isotope tracing, and redox proteomics also serve as valuable complementary and validation approaches for quantitative metabolic analyses. Metabolomics provides a comprehensive snapshot of metabolite profiles[20,21], while ¹³C isotope tracing techniques can estimate specific metabolic pathway activity[22,23]. Furthermore, redox proteomics reveals the extent of redox modifications on sensitive amino acid residues (e.g., cysteine and methionine), providing insight into the post-translational regulation of enzyme activity[24–27]. However, such experimental analyses are often underrepresented in GEM-focused metabolic studies, limiting both quantitative metabolic investigation and empirical validation when studying complex metabolic diseases. As such, leveraging both computational modeling and experimental approaches is essential for advancing quantitative understanding of complex metabolic disorders like type 2 diabetes (T2D).

T2D is characterized by insulin resistance and hyperglycemia, posing a significant global health challenge[28–30]. *Drosophila* effectively models key metabolic features of human T2D through high-sugar diet (HSD) feeding, which impairs glucose homeostasis, and induces mitochondrial dysfunction and tissue-specific metabolic dysregulation[13,14,31,32]. Among affected tissues, muscle plays a central role in glucose metabolism and insulin sensitivity. While perturbations in glycolytic and TCA cycle activities have been observed in diabetic muscle[29,33,34], many questions remain unanswered. Specifically, the extent to which NAD(H)-dependent reactions are altered, the identity of potential rate-controlling steps within glycolysis, and the additionally dysregulated metabolic pathways beyond central carbon metabolism remain to be elucidated.

In this study, we first reconstructed 32 tissue-specific GEMs for adult *Drosophila* by integrating a curated *Drosophila* metabolic network model with pseudo-bulk single-nuclei transcriptomics data. These GEMs enabled us to systematically evaluate similarities and differences in metabolic network structures across individual tissues. After validating GEM-based predictions on enriched pathways using targeted metabolomics and pathway enrichment analyses, we demonstrated the utility of tissue-specific GEMs in modeling human metabolic disease. Specifically, we applied the muscle-GEM to simulate metabolic changes induced by a HSD, a well-established *Drosophila* model of T2D. Constraint-based flux analyses revealed altered fluxes in NAD(H)-dependent fluxes, including a decreased maximal NADH production capacity, and sensitivity analysis indicated distributed control of glycolytic fluxes across several enzymes, including GAPDH. [U-¹³C]-glucose tracing and redox proteomics further confirmed decreased glycolytic flux and revealed that these changes were associated with increased redox modifications of glycolytic enzymes. Pathway-level flux comparisons further highlighted dysregulated fructose metabolism. Together, this work establishes a quantitative framework for tissue-specific metabolic modeling in *Drosophila* and demonstrates its utility for identifying high sugar diet-induced metabolic perturbations at reaction and pathway levels.

## Results

### Reconstruction of 32 tissue-specific genome-scale metabolic models (GEMs) in *Drosophila melanogaster*

To evaluate tissue-specific metabolism in *Drosophila*, we designed a strategy to reconstruct 32 tissue-specific GEMs and analyze their metabolic network structures and functions (Fig. 1a and Supplementary Data 1a–e). The reconstructed tissue-specific GEMs revealed variations in their metabolic network structures, reflected in the differing numbers of reactions, metabolites, and genes across tissues (Supplementary Fig. 1a). Among 32 tissue-specific GEMs, fat body and oenocytes, analogous to human adipose tissue and liver, contained the highest number of reactions ($n = 5447$ and $5015$), while most neuronal tissues had the fewest number of reactions ($n = 2533$ to $3077$). A similar trend was observed for the metabolite numbers across tissue-specific GEMs. Additionally, we observed a positive correlation between the number of reactions and metabolites ($R^2 = 0.92$), consistent with the expected increase of metabolite numbers as reactions number increases. The number of genes also varied across models, ranging from 565 to 858, with the germline and fat body containing the higher numbers. Similarly, we observed a positive correlation between the number of reactions and genes ($R^2 = 0.54$) (Supplementary Fig. 1b).

Based on these differences, we hypothesized that the tissues performing similar metabolic functions would show similar metabolic network structures. To test this hypothesis, we compared the metabolic network structures of 32 tissue-specific GEMs by specifically comparing their reaction contents (Fig. 1b). Hierarchical clustering analysis revealed 13 distinct clusters of similar metabolic network structures (Supplementary Fig. 1c and Supplementary Data 1f). As expected, tissues of similar functions were grouped together—muscle and indirect muscle; fat body and oenocytes; hindgut and enteroendocrine cells; seven neuron GEMs, and six glia cells distributed across three different clusters. Additionally, to assess how these GEM-based clusters compared to transcriptomics-based clustering, we performed hierarchical clustering of gene expression data and calculated Jaccard indices to quantify overlap (Supplementary Fig. 1d, e). While we observed high overlap between the two approaches for muscle and neuron clusters (Jaccard index = 1) and fat body/oenocytes (0.75), clustering of glia and gut tissues showed lower overlap between the two approaches (Fig. 1c and Supplementary Data 1g). These results indicate that clustering based on metabolic network structures, which incorporate stoichiometric constraints of biochemical reactions, reveals metabolic distinctions not apparent from transcriptomics alone.

Next, to further evaluate differences in metabolic network structures across tissue-specific GEMs, we analyzed their metabolic subsystem coverage. We found that transport reaction subsystem accounted for the largest proportion of all subsystems, comprising approximately $35 \pm 4\%$ of all reactions (Fig. 1d). Within this subsystem, extracellular transporter reactions accounted for approximately $80 \pm 6\%$. Interestingly, while muscle group did not have the most transport reactions compared to other tissue groups, it had the highest fraction of the extracellular transport reactions ($f = 0.86 \pm 0.02$) (Fig. 1e and Supplementary Data 1h). Additionally, we calculated the percent difference in subsystem coverage and identified 33 subsystems with more than 50% deviation from mean coverage across tissues (Fig. 1f and Supplementary Data 1i, j). Notably, fat body and oenocyte GEMs had the highest reaction counts, particularly in beta-oxidation subsystems ($n_{reactions} = 36.6 \pm 15$, $p < 0.0001$) (Supplementary Fig. 1f), supporting their known function in beta-oxidation of fatty acids[11,35].

Moreover, we performed a metabolic task analysis to determine the extent to which tissue-specific GEMs could perform 219 predefined metabolic tasks (Supplementary Data 1k)[7,36]. Tissue-specific GEMs could pass an average of $40 \pm 10$ tasks, with germline cells

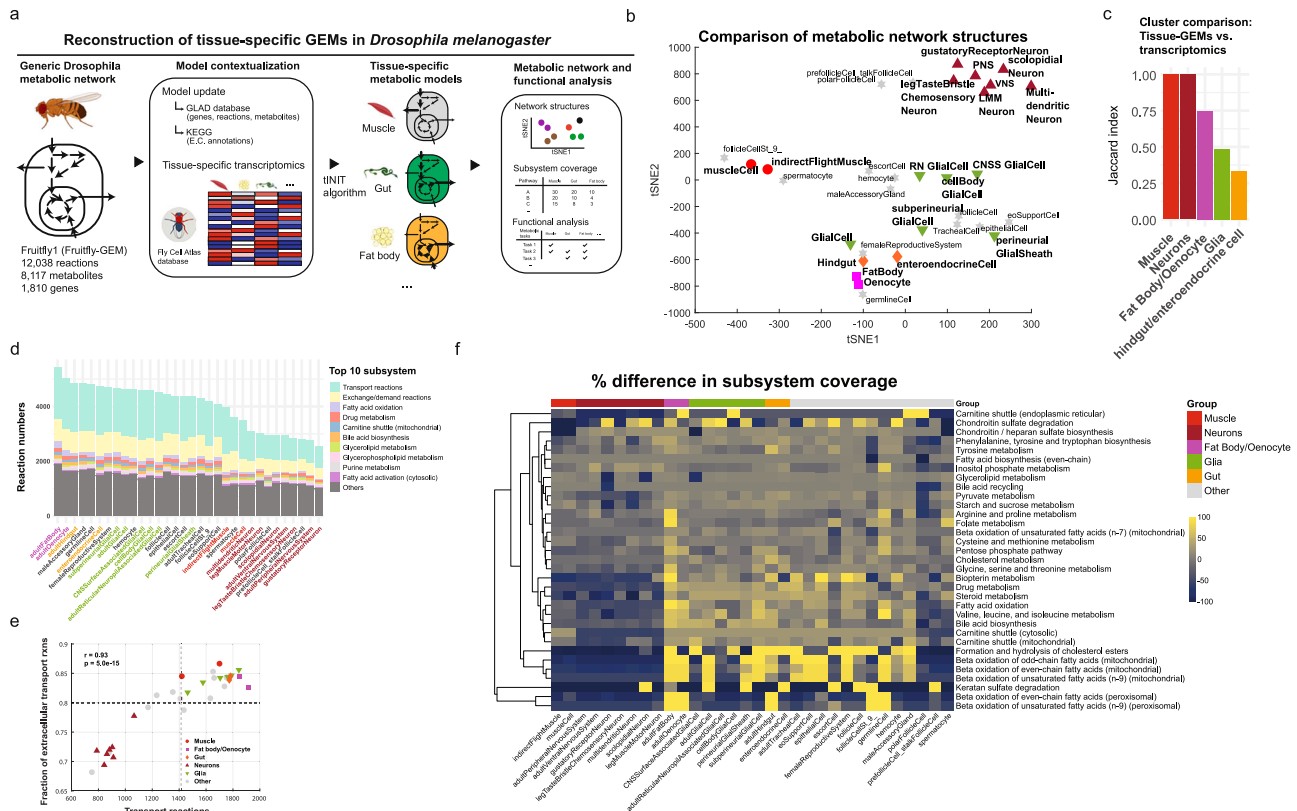

**Fig. 1 | Reconstruction of 32 tissue-specific genome-scale metabolic models (GEMs) in *Drosophila melanogaster*. a** Pipeline for reconstruction of tissue-specific GEMs in *Drosophila*. Elements created in BioRender. Moon, S. (2026) https://BioRender.com/1nasxv2. **b** t-SNE plot comparing the metabolic network structures of 32 tissue-specific GEMs using Hamming similarity. Each point represents one tissue-specific GEM, with marker shapes and colors indicating major tissue groups (muscle: red circles; fat body/oenocyte: purple squares; gut: orange diamonds; glia: green downward triangles; neurons: dark red upward triangles; others: gray stars). PNS peripheral nervous system, VNS ventral nervous system, LMM neuron leg muscle motor neuron, RN glial cell reticular neuropil associated glial cell, CNSS glial cell central nervous system surface associated glial cell.

**c** Jaccard index analysis comparing overlap between clusters defined by tissue-GEM and those based on gene expression. **d** Number of reactions per subsystem across GEMs, highlighting the top 10 subsystems by reaction count. **e** Fraction of extracellular transport reactions relative to all transport reactions, with major tissue groups highlighted. Dashed lines indicate mean values. Correlation was assessed using a two-sided Pearson correlation test ($r = 0.93$, $p = 5e-15$). **f** Heatmap of differential subsystem coverage across tissues, with bright yellow indicating higher coverage and dark blue lower coverage. Subsystems with ≥15 reactions and >50% deviation from mean coverage are shown. Source data are provided as a Source Data file.

completing the most metabolic tasks ($n_{pass} = 66$) and polar follicle cells the least ($n_{pass} = 17$) (Supplementary Fig. 1g, h). Additionally, consistent with the known role of fat body in performing gluconeogenesis and trehalose synthesis[37], we confirmed that fat body-GEM could synthesize trehalose from various substrates, including alanine, pyruvate, glutamine, and glycerol (Supplementary Data 1l). Furthermore, after categorizing metabolic tasks into seven metabolic systems, we performed Fisher's exact test and found significant tissue-specific associations with the specific metabolic systems: fat body with carbohydrate/amino acid metabolism, and CNS glia/gustatory neurons with lipid metabolism (Supplementary Fig. 1i and Supplementary Data 1m, n). Taken together, our reconstructed tissue-specific GEMs provide a systems-level framework to evaluate metabolic differences and similarities across tissues in *Drosophila*, enabling quantitative comparisons of tissue-specific metabolic network structures and metabolic capabilities.

## Validation of tissue-specific GEMs through regional metabolomics and pathway analysis

Validating the predictions from tissue-specific GEMs is essential to establish confidence in GEM-based analyses. Although gene essentiality analysis is commonly used for validation[38,39], in vivo growth rate data were unavailable for the tissues we investigated. Thus, we performed targeted metabolomics and pathway enrichment analysis,

while comparing these results to GEM-based pathway analysis. We hypothesized that enriched pathways identified through metabolomics would also be represented in the corresponding tissue-specific GEMs. To test the hypothesis, we first profiled 303 polar metabolites in four dissected *Drosophila* regions—head (containing neuronal and glial cells), thorax (containing muscle cells), gut (containing hindgut/enteroendocrine cells), and abdomen (containing fat body, oenocytes, and other tissues) (Fig. 2a and Supplementary Data 2a).

Principal component analysis and hierarchical clustering revealed distinct metabolite profiles among these regions (Fig. 2b, c). Specifically, we identified 10 significantly enriched metabolites in thorax, 13 in head, 20 in gut, and 1 in abdomen (Log₂FC > 2 and p-value < 0.05) (Fig. 2d and Supplementary Data 2b). Qualitatively, these enriched metabolites aligned with known tissue functions. For instance, in thorax, energy-related metabolites such as adenosine, adenylosuccinate, adenine, and AMP were enriched, consistent with muscle's high demand on energy metabolism compared to other tissues[11] (Supplementary Fig. 2a). In heads, 4-aminobutyrate (GABA), N-acetyl-L-aspartate, GMP, and ascorbates, were enriched. GABA is a major inhibitory neurotransmitter, and N-acetyl-L-aspartate is often used as a marker for neuronal health[40,41]. In gut, including Malpighian tubules (MT), dietary-related metabolites, such as pyridoxine, betaine, and uric acids, were enriched, consistent with their dietary origin and uric acids generated from MT's purine excretion[42–45]. In abdominal carcass,

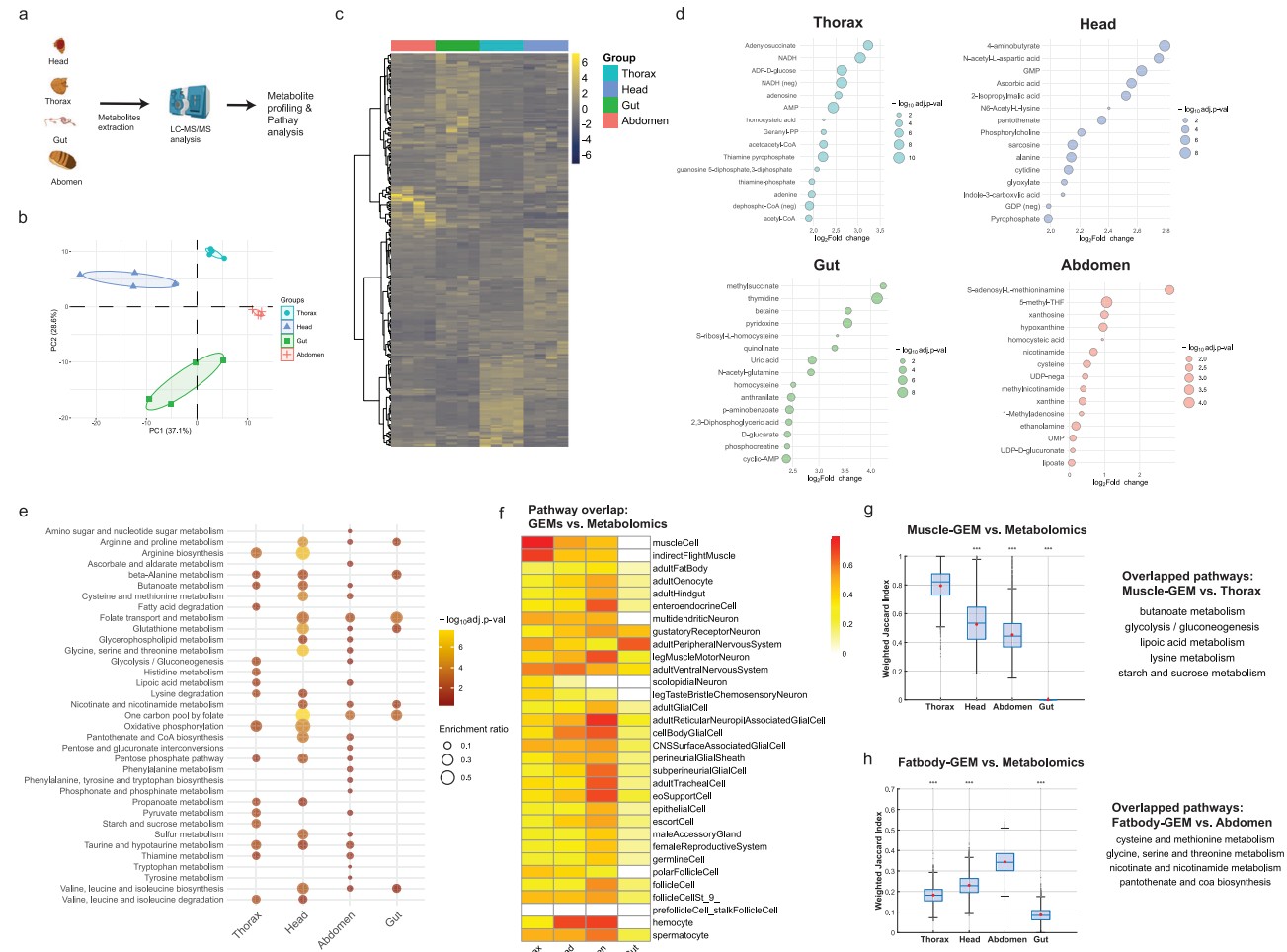

**Fig. 2 | Validation of tissue-specific GEMs through regional metabolomics and pathway analysis. a** Schematics showing the metabolomics analysis performed on four *Drosophila* regions, followed by KEGG over-representation analysis. Elements created in BioRender. Moon, S. (2026) https://BioRender.com/1nasxv2. **b** Principal component analysis (PCA) of metabolite profiles across regions (*n* = 4). **c** Heatmap showing the metabolite profiles across thorax, head, abdomen, and gut. **d** Bubble plot showing enriched metabolites in each region compared to others. One-way ANOVA followed by Benjamini–Hochberg (BH) correction was used for multiple comparisons. **e** Over-representation analysis of enriched metabolite sets across regions, using a one-sided hypergeometric test, followed by BH correction for multiple comparisons. **f** Heatmap of weighted Jaccard index values showing

pathway overlaps between metabolomics-derived and GEM-predicted pathways. **g, h** Boxplots showing weighted Jaccard index distributions between metabolomics-derived pathways and either **g** Muscle-GEM or **h** Fat body-GEM. Center line is median; box limits are first and third quartiles; whiskers are 1.5 × interquartile range; red points are mean; gray points present individual bootstrap samples (*n* = 10,000), shown in full including those outside the whiskers. Statistical significance is based on comparisons against the reference region (**g**: thorax; **h**: abdomen) using one-way ANOVA followed by Bonferroni-corrected pairwise tests; *** *p* < 0.001. Overlapping pathways between GEM-predicted and region-specific enriched pathways are shown to the right. Source data are provided as a Source Data file.

containing fat body and oenocytes, nucleotide and one-carbon metabolites were enriched, such as xanthosine and hypoxanthine, methionine s-adenosyl-l-methioninamine and 5-methyl-THF.

Next, we performed KEGG pathway enrichment analysis using the hypergeometric test based on the enriched metabolite sets[46] (Fig. 2e and Supplementary Data 2c). The most significantly enriched pathway was purine metabolism (KEGG ID: dme00230) for thorax, pyrimidine metabolism (dme00240) for gut, and one carbon pool by folate (dme00670) for head and abdomen (Supplementary Fig. 2b). We also identified six pathways commonly enriched across all regions, such as TCA cycle, glyoxylate, purine, and pyrimidine metabolism (Supplementary Fig. 2c). Moreover, several region-specific enriched pathways were identified, such as starch/sucrose metabolism, histidine metabolism, and fatty acid degradation in thorax.

Next, we compared these metabolomics-derived enriched pathways to those predicted from subsystem coverage analysis of tissue-specific GEMs, using weighted Jaccard index[47,48] (Fig. 2f and Supplementary Note 1). In brief, the index quantifies the degree of pathway overlap between the two datasets. We expected higher index values for tissues associated with specific regions (e.g., muscle with thorax[49] and fat body/oenocytes with abdomen[50]). Indeed, muscle-GEM showed the highest index value with thorax (index: 0.8 ± 0.1), compared to those for other regions (head: 0.48 ± 0.2, abdomen: 0.44 ± 0.1, gut: 0) (Fig. 2g and Supplementary Data 2d–f). The overlapped pathways included butanoate, glycolysis, lipoic, and starch metabolism. Additionally, fat body-GEM also showed high index value with abdomen, with overlapped pathways including cysteine, nicotinamide, and pantothenate metabolism (Fig. 2h). However, pathway overlaps were less distinct for neuron/glia cells and gut groups, as no uniquely enriched pathways were identified for these regions in our datasets (Supplementary Fig. 2d). Furthermore, when we repeated this analysis using gene expression data, we observed significantly lower Jaccard index values across all tissues, with no distinct trends (Supplementary Fig. 2e and Supplementary Data 2g, h). This indicated that GEM-predicted enriched pathways more accurately reflect metabolomics-derived pathways, compared to gene expression-derived enriched pathways. Taken

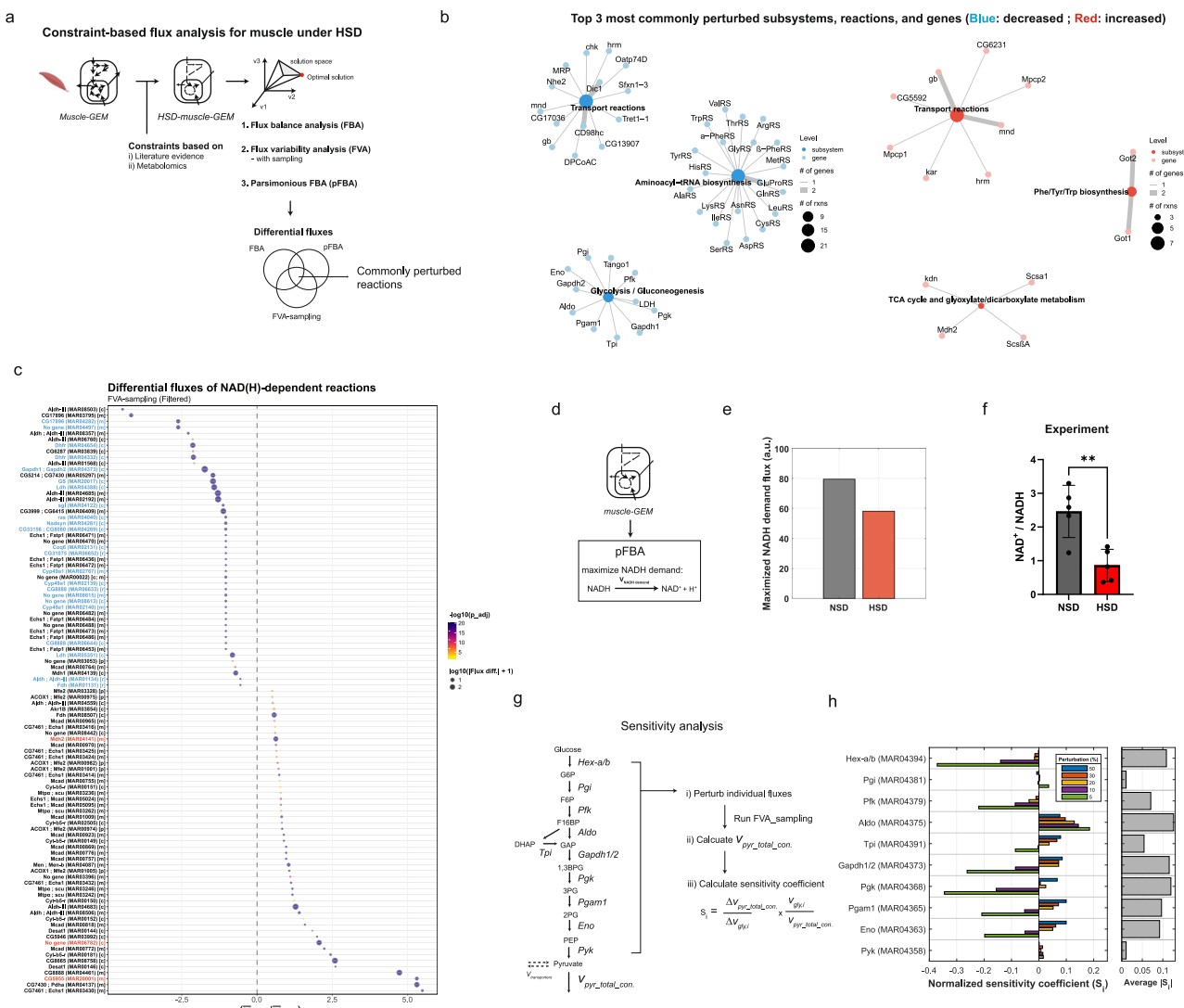

**Fig. 3 | Constraint-based flux analyses predict perturbations in NAD(H)-dependent reactions in muscle under high sugar diet. a** Schematics of constraint-based semi-quantitative flux analyses for muscle under HSD. Constraints were applied to reflect metabolic alterations observed in muscle under HSD, recapitulating features of type 2 diabetes in *Drosophila*. FBA, pFBA, and FVA_sampling analyses were performed to estimate differential reaction fluxes, identifying commonly perturbed reactions across the methods. Elements created in BioRender. Moon, S. (2026) https://BioRender.com/1nasxv2. **b** Networks showing the top three most commonly perturbed subsystems, reactions, and genes from the decreased (blue) and increased (red) reaction sets identified across flux analyses. Node size indicates the number of reactions associated with each subsystem, and edge width represents the number of linked genes. **c** Differential fluxes of NAD(H)-dependent reactions between HSD and NSD from FVA_sampling analysis. Dot size reflects absolute differential flux and color scale indicates adjusted *p*-values from a two-sample *Z*-test. *X*-axis shows log₂ fold change, with threshold of ± 0.5. Reactions consistently decreased or increased across different flux analyses are labeled in

blue or red, respectively. **d** Schematic of the pFBA evaluating maximum NADH production capacity under HSD and NSD conditions. An artificial NADH oxidation reaction (NADH → NAD$^+$) was introduced as a total NADH demand across cytosolic, mitochondrial, and peroxisomal compartments. **e** Maximal NADH demand flux estimated by pFBA between NSD and HSD conditions. **f** NAD$^+$/NADH ratio in $w^{1118}$ male flies fed NSD or HSD for five days. Data are shown as mean ± SD from biological replicates ($n = 5$). Statistical significance was assessed using a two-tailed unpaired *t*-test; ** $p < 0.01$. **g** Schematic of the sensitivity analysis workflow for glycolysis. Individual glycolytic fluxes were perturbed, followed by FVA-sampling to quantify changes in total pyruvate consumption flux, which served as a proxy for glycolytic output. Normalized sensitivity coefficients were calculated to assess each reaction's influence on the pathway output flux. **h** Bar plot showing normalized sensitivity coefficients ($S_i$) of glycolytic reactions under different perturbation magnitudes (5–50%). The bar plot on the right shows the average absolute sensitivity ($|S_i|$), highlighting reactions with the greatest overall influence under these simulations. Source data are provided as a Source Data file.

together, these results validate GEM-predicted pathway analysis, particularly for muscle and fat body.

## Constraint-based flux analyses predict perturbations in NAD(H)-dependent reactions in muscle under high sugar diet

Among the reconstructed tissue-specific GEMs, we further applied the muscle-GEM to evaluate how muscle metabolism could be rewired in response to high sugar diet (HSD), which induces T2D-like phenotypes in *Drosophila*[14,31,51]. We first defined a HSD-muscle-GEM by constraining

reaction rates to simulate metabolic phenotypes observed in type 2 diabetic muscle based on the literature evidence[28,29,33,34,52–54] (Fig. 3a and Supplementary Data 3a, b). These constraints included glucose uptake rates, GAPDH, and several reactions in TCA cycle[53,55]. Consistent with our applied constraints, flux variability analysis with sampling (FVA-sampling) showed a reduction in glucose uptake rate, along with decreased fluxes through GAPDH, OGDH, and SDH (Supplementary Fig. 3a, b, Supplementary Note 2, and Supplementary Data 3c, d). The decrease in model-generated glucose uptake rate was confirmed

experimentally by measuring glucose uptake rates in $w^{1118}$ male flies fed with HSD (Supplementary Fig. 3c).

Next, to identify perturbed reactions beyond those we directly constrained, we systematically compared differential fluxes obtained from flux balance analysis (FBA), FVA-sampling, and parsimonious FBA (pFBA) (Supplementary Note 3). In brief, we performed pFBA to complement FVA-sampling analysis, which could generate non-physiologically high fluxes in reactions such as NAD(H)-dependent cycling reactions and generate fluxes near boundaries due to under-constrained nature of network (Supplementary Fig. 3d–g and Supplementary Data 3e, f). By comparing these analyses, we identified 77 reactions showing consistently decreased fluxes and 18 with increased fluxes in the HSD-muscle-GEM relative to the unconstrained model (Supplementary Fig. 3h and Supplementary Data 3g). Among the reactions with decreased fluxes, many were associated with transport, aminoacyl-tRNA biosynthesis, and glycolysis, with associated genes including *Dic1*, *Cd98hc*, *Gapdh1/2*, *Aldo*, *Eno*, *Pfk*, and aminoacyl-tRNA synthetases such as *AsnRS* and *AspRS* (Fig. 3b). In contrast, reactions with increased fluxes were mainly linked to transport, aromatic amino acid biosynthesis, and TCA/glyoxylate metabolism, with associated genes including *Hrm*, *Cg6231*, *Got1/2*, *Kdn*, *Scsa1*, and *Mdh2*.

As perturbed NADH metabolism has also been implicated in T2D[56,57], we further investigated individual NAD(H) dependent reactions to identify specific reactions showing altered fluxes. By comparing differential fluxes of 135 active NAD(H) dependent reactions across three different flux simulations, we identified 23 reactions showing consistently decreased fluxes and 3 reactions with increased flux (Supplementary Fig. 3i, j and Supplementary Data 3h–j). Associated genes with decreased fluxes included *Dhfr*, *Ldh*, *Gs*, and *Gapdh1/2*, while those with increased fluxes included *Mdh2* and *CG5955*. Profiling differential fluxes of NAD(H)-dependent reactions further highlighted a variety of significantly perturbed reactions and their magnitudes (Fig. 3c). Next, to further investigate whether the network's maximum capacity of NADH production was perturbed, we introduced an artificial NADH demand reaction and maximized its flux (Fig. 3d). This analysis revealed approximately a 27% reduction in NADH maximum production capacity in the HSD-muscle GEM, accompanied by altered contributions of individual NADH-producing reactions (Fig. 3e and Supplementary Fig. 3k). Although this change does not directly predict the cellular $NAD^+$/NADH ratio, it suggests impaired NADH turnover, consistent with our experimental observation of a decreased $NAD^+$/NADH ratio in thoracic muscle of $w^{1118}$ flies fed with HSD (Fig. 3f). Together, our constraint-based semi-quantitative flux analyses predicted perturbations in specific subsystems, reactions, and genes in muscle under HSD, highlighting disrupted NAD(H)-dependent reactions characterized by altered flux distributions and decreased NADH production capacity.

## Sensitivity analysis reveals distributed control of glycolytic flux

Given that NAD(H)-dependent reaction fluxes were perturbed in the HSD-muscle GEM and GAPDH functions as one of the key $NAD^+$-dependent glycolytic enzymes, we examined whether GAPDH could serve as a rate controlling step in glycolysis under HSD condition. To test this, we performed a sensitivity analysis (See "Methods"). In brief, we systematically decreased the baseline flux of each glycolytic reaction, defined as the median flux from FVA-sampling, and quantified the total pyruvate consumption flux as a proxy for glycolytic output flux (Fig. 3g and Supplementary Fig. 3l). At relatively larger perturbation (e.g., 20 to 50%), most of the lower glycolytic enzymes showed positive sensitivity coefficients, suggesting that decreases in these fluxes led to reductions in total pyruvate consumption (Fig. 3h and Supplementary Data 3k). In contrast, at smaller perturbations (e.g., 5 and 10%), several enzymes, including Hex-a/b, Pgk, and Gapdh1/2, showed strong

negative sensitivity coefficients, suggesting that decreases in these fluxes may trigger compensatory increases in total pyruvate consumption through alternative reactions. Indeed, evaluation of individual pyruvate-consuming reactions revealed diverse responses to perturbations. Specifically, while alanine transaminase (*CG1640*) consistently showed decreased flux in response to high perturbation (e.g., 50%), monocarboxylate transporter (*MCT1*), or other reactions showed increased fluxes in response to varying perturbations, reflecting the nonlinear and compensatory nature of metabolic network (Supplementary Fig. 3m). When comparing absolute magnitudes of sensitivity coefficients across all perturbations, we found Aldo exhibited the highest average sensitivity coefficient (0.13), with Pgk (0.12), Gapdh1/2 (0.12), and Hex-a/b (0.11) showing comparable sensitivity (Fig. 3h). Altogether, these results suggest that glycolytic flux is not solely controlled by GAPDH as initially hypothesized, but is distributed among several other enzymes and varies with perturbation magnitude under this condition.

## Model-predicted decreases in glycolytic flux, including GAPDH, validated through ¹³C-glucose tracing

To validate the predicted fluxes in glycolysis, particularly involving GAPDH, we performed in vivo ¹³C-glucose isotopic tracing experiments. After feeding $w^{1118}$ male flies with a HSD containing uniformly labeled ¹³C-glucose tracer for five days, we dissected the thoracic muscle and evaluated both the metabolite intensities and labeling patterns of downstream metabolites using mass spectrometry (Fig. 4a). As the [U-¹³C]glucose is metabolized in cells, glycolytic intermediates will have M + 6 or M + 3 mass isotopomers, and TCA cycle metabolites will show M + 2 mass isotopomers after the first cycle (Fig. 4b). Based on our sensitivity analysis, where we observed strong control around Aldo, Pgk, and Gapdh, we hypothesized that a bottleneck step would result in decreased fractional labeling of downstream isotopomers, accompanied by accumulation of upstream substrates and decreased levels of downstream products.

Indeed, we observed that upper glycolytic intermediates increased by approximately 23%, including F6P, F16BP, and glyceraldehyde 3-phosphate, a substrate of GAPDH (Supplementary Fig. 4a, b and Supplementary Data 4a, b). Conversely, lower glycolytic intermediates, such as 3PG and lactate, decreased by 27% and 14%, respectively. Moreover, the fractional labeling of lower glycolytic intermediates was significantly decreased compared to that of upper glycolytic intermediates (Fig. 4c). Specifically, the M + 3 fractional labeling of 1,3-BPG, the product GAPDH, was nearly 90 % lower in response to HSD, indicating a significantly reduced GAPDH activity. In TCA cycle, the abundances of citrate, succinate, fumarate, malate, and glutamate, increased by 40%, 34%, 14%, 97%, and 24%, respectively (Supplementary Fig. 4c, d and Supplementary Data 4c). Moreover, their M + 2 fractional labeling decreased relative to M + 6 glucose, suggesting a decreased contribution of glucose-derived carbon to the TCA cycle (Fig. 4d).

Furthermore, to investigate the extent to which our model-predicted fluxes align with experimental measurements, we performed a correlation analysis by mapping the flux of each reaction in the HSD-muscle-GEM to the fractional labeling of its corresponding metabolite product. Notably, we observed a strong correlation between the fractional labeling data and model-predicted relative fluxes within glycolysis ($\rho = 0.81$, $p = 0.03$), supporting the accuracy of model predictions for this pathway (Fig. 4e). In contrast, correlation in the TCA cycle was weaker ($\rho = 0.12$), potentially due to the contributions from other carbon sources (e.g., fatty acids[58]) accounting the predicted fluxes in the TCA cycle (Supplementary Fig. 4e). Taken together, these results confirm the model-predicted flux changes, specifically for glycolysis, and reveal GAPDH as one of the key regulatory steps in muscle in response to HSD.

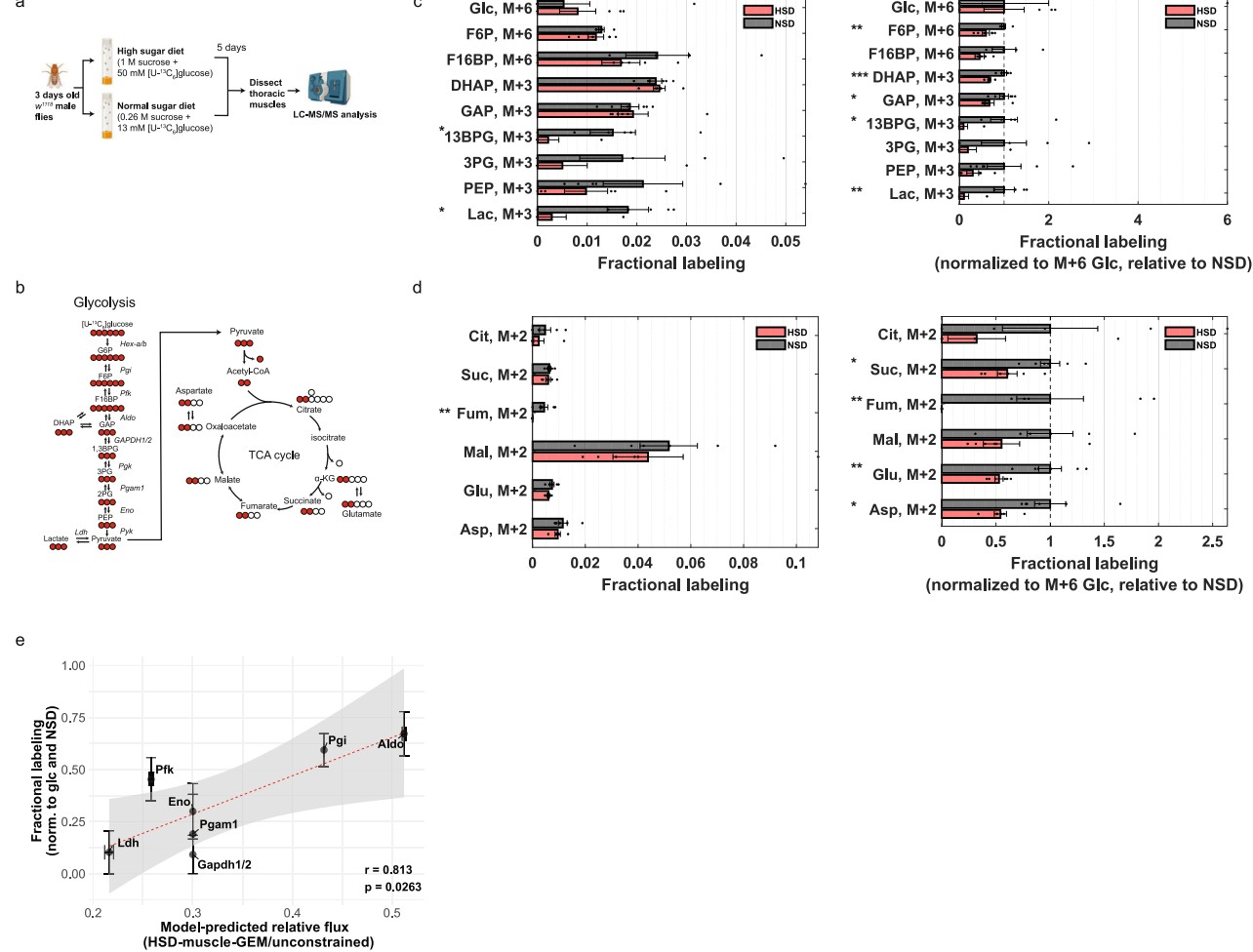

**Fig. 4 | Model-predicted decreases in glycolytic flux, including GAPDH, validated through ¹³C-glucose tracing. a** Experimental design for ¹³C-glucose isotope tracing experiments in *Drosophila* thoracic muscle under high sugar diet (HSD) conditions. Elements created in BioRender. Moon, S. (2026) https://BioRender.com/1nasxv2. **b** Isotopic labeling patterns generated by [U-¹³C₆]glucose in glycolysis and TCA cycle. Red circles represent ¹³C-labeled carbons. **c, d** Fractional labeling of mass isotopomers of **c** glycolytic or **d** TCA cycle intermediates, shown as both fractional labeling (left) and values normalized to M + 6 glucose and scaled relative to normal sugar diet (NSD) (right). Bars represent mean ± SEM, and individual points indicate biological replicates (*n* = 6). M + x denotes a mass isotopomer containing x ¹³C atoms. Statistical significance was assessed using two-tailed unpaired *t*-tests comparing NSD and HSD for each metabolite; * *p* < 0.05, ** *p* < 0.01, *** *p* < 0.001. **e** Correlation between model-predicted relative flux changes (HSD-muscle-GEM/Unconstrained) and fractional labeling of glycolytic intermediates (HSD/NSD). Relative flux of each reaction was mapped to the normalized fractional labeling of its corresponding product metabolite. Fractional labeling values (*y*-axis) are shown as mean ± SEM from biologically independent replicates (*n* = 6), and model-predicted flux values (*x*-axis) are shown as mean ± SEM from flux sampling simulations (*n* = 10,000). The dashed red line represents a linear regression fit, with the shaded region indicating the 95% confidence interval. Correlation was assessed using a two-sided Pearson correlation test (*r* = 0.813, *p* = 0.0263). Source data are provided as a Source Data file.

## Model-predicted decreases in glycolytic flux correlate with increased redox modification of glycolytic enzymes

Since redox modification, including oxidation of cysteine residue, can directly alter enzyme activity such as GAPDH[24,26,59,60], we were wondering whether the decreased glycolytic fluxes, particularly GAPDH, were linked to redox modifications (Fig. 5a). Given the increased oxidative stress observed in diabetic muscle[53,61,62], we hypothesized that redox modification at cysteine or methionine residues would also be elevated in HSD-fed fly muscle. Indeed, PCA of redox proteomics revealed greater variability in peptide oxidation profiles in HSD samples compared to NSD controls (Fig. 5b). Among 1049 detected peptides, 189 peptides exhibited significantly increased redox modifications (|Log₂FC| > 0.5 and adjusted *p*-value < 0.05) (Fig. 5c, Supplementary Fig. 5a and Supplementary Data 5a). Next, based on these significantly oxidized peptides, we performed KEGG overrepresentation analysis[63] to identify pathways enriched for redox-modified enzymes (Fig. 5d and Supplementary Data 5b). Consistent to

prior studies, the most affected pathways were oxidative phosphorylation (dme00190), a well-known site of mitochondrial dysfunction observed in diabetic muscle[64,65]. Within this pathway, we observed increased redox modifications in peptides, including succinate dehydrogenase, NADH dehydrogenase, V-ATPase, citrate synthase, and isocitrate dehydrogenase (Supplementary Fig. 5b). Interestingly, glycolysis was also significantly affected, with increased redox modifications observed in peptides, including Ald1, Gapdh1/2, Pyk, Pgi, Pfk, Pgm1, Eno, Fdh, and Pgk (Fig. 5e).

To further evaluate the relationship between the extent of redox modification and predicted fluxes in glycolysis, we performed a correlation analysis and found a significant negative correlation (*r* = −0.78, *p* = 0.025) (Fig. 5f). Specifically, Pyk exhibited the highest level of redox modification and corresponded to the largest flux decrease, whereas Pgm1 showed minimal redox modification and the smallest flux change (Supplementary Fig. 5c, d). Notably, protein levels of most glycolytic enzymes remained unchanged, except for Pyk, suggesting that redox

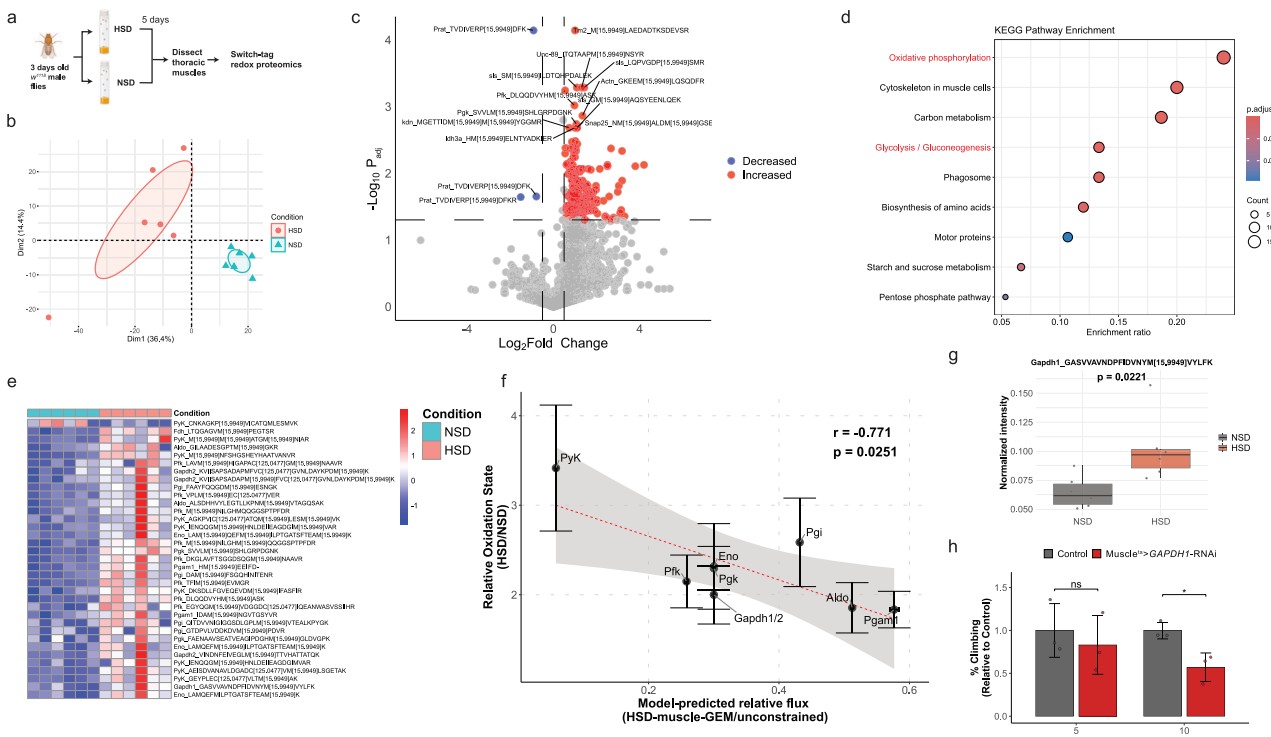

**Fig. 5 | Model-predicted decreases in glycolytic flux correlate with increased redox modification of glycolytic enzymes. a** Experimental design for switch-tag redox proteomics in *Drosophila* thoracic muscle under HSD conditions. Elements created in BioRender. Moon, S. (2026) https://BioRender.com/1nasxv2. **b** PCA plot showing variance in oxidized peptides between HSD and NSD conditions. **c** Volcano plot displaying differentially oxidized peptides in thoracic muscle between HSD and NSD conditions. The intensities were normalized within each protein across samples. **d** Dot plot showing KEGG pathway enrichment (*Drosophila*) based on significantly oxidized peptides. **e** Heatmap showing the significantly oxidized peptides in glycolysis between NSD and HSD conditions. **f** Correlation between model-predicted relative flux changes (HSD-muscle-GEM/Unconstrained) and relative oxidation state changes of glycolytic enzyme peptides (HSD/NSD). Relative oxidation state was calculated as the average oxidation level of peptides from each enzyme in HSD, normalized to NSD. Relative oxidation state values (*y*-axis) are shown as mean ± SEM from biologically independent replicates (*n* = 6), and model-

predicted flux values (*x*-axis) are shown as mean ± SEM from flux sampling simulations (*n* = 10,000). The dashed red line represents a linear regression fit, with the shaded region indicating the 95% confidence interval. Correlation was evaluated using a two-sided Pearson correlation test (*r* = −0.771, *p* = 0.0251). **g** Boxplot showing normalized intensity of significantly oxidized GAPDH peptide. Center line is median; box limits are first and third quartiles; whiskers are 1.5 × interquartile range; points represent biologically independent replicates (*n* = 6). Statistical significance was assessed using a two-tailed unpaired *t*-test and the *p*-value is shown above the plot. **h** Climbing ability of male *Mhc^ts^ > attp40* (control), or *GAPDH1*-RNAi flies, measured at day 5 and 10 after fed with HSD. Bar represent mean ± SD, and individual points indicate biological replicates (*n* = 3). Statistical significance was assessed using two-tailed unpaired *t*-tests comparing control vs. *GAPDH1*-RNAi within each time point; ns not significant, * *p* < 0.05. Source data are provided as a Source Data file.

modifications, rather than enzyme abundance, more strongly associate with changes in glycolytic flux (Supplementary Fig. 5e, f).

Moreover, since our flux analyses and isotope tracing results revealed GAPDH as one of the key perturbed steps in glycolysis, we further evaluated this enzyme. In *Drosophila*, two isoforms of GAPDH are present, both closely related to human GAPDH and known to perform partially redundant functions in glycolysis (Supplementary Fig. 5g)[66]. While the enzyme levels of both GAPDH1 and GAPDH2 remained unchanged (Supplementary Fig. 5h), we observed significant redox modifications at Met-40 residue in GAPDH1 and at Cys-130, Met-127, 141 and 172 in GAPDH2 (referenced to Uniprot:P07486 for GAPDH1 and M9PJN8 for GAPDH2) (Fig. 5g and Supplementary Fig. 5i, j). Structural modeling using AlphaFold further predicted that Met-40 is in near to the NAD⁺ binding site (~9 Å), suggesting that redox modification at this residue could indirectly affect NAD⁺ binding and impair activity (Supplementary Fig. 5k). Additionally, to further investigate the functional role of GAPDH in muscle, we used a muscle-specific *Mhc-Gal4* driver combined with a temperature-sensitive Gal80ts system to knock down GAPDH1 in fly muscle. Indeed, muscle-specific downregulation of GADPH1 led to a significant decline in climbing ability in male flies fed with HSD starting at day 10, suggesting a physiologically

important role for GAPDH1 in maintaining muscle function (Fig. 5h and Supplementary Fig. 5l). In summary, these results indicate a strong association between increased redox modifications and decreased glycolytic fluxes, identifying the redox modification of GAPDH and its knockdown causing climbing defects.

## Pathway-level flux analysis reveals dysregulated fructose metabolism

Next, to identify pathway-level perturbations beyond glycolysis under HSD conditions, we evaluated differential pathway fluxes obtained from FVA-sampling and pFBA analyses (Fig. 6a and "Methods"). In brief, we defined pathway flux as the average of non-zero flux magnitudes within each pathway. Using this approach, we identified 77 significantly perturbed pathways, with 33 pathways overlapping between flux sampling and pFBA analyses (Fig. 6b and Supplementary Data 6a). Consistent with our prior findings, we observed significant decrease in pathway fluxes through glycolysis, oxidative phosphorylation, and the TCA cycle, with fructose showing the most pronounced decrease in the HSD muscle compared to control. Among the top perturbed pathways with increased pathway fluxes, we found fatty acid metabolism (e.g., β-oxidation, desaturation, elongation, and omega-3/6

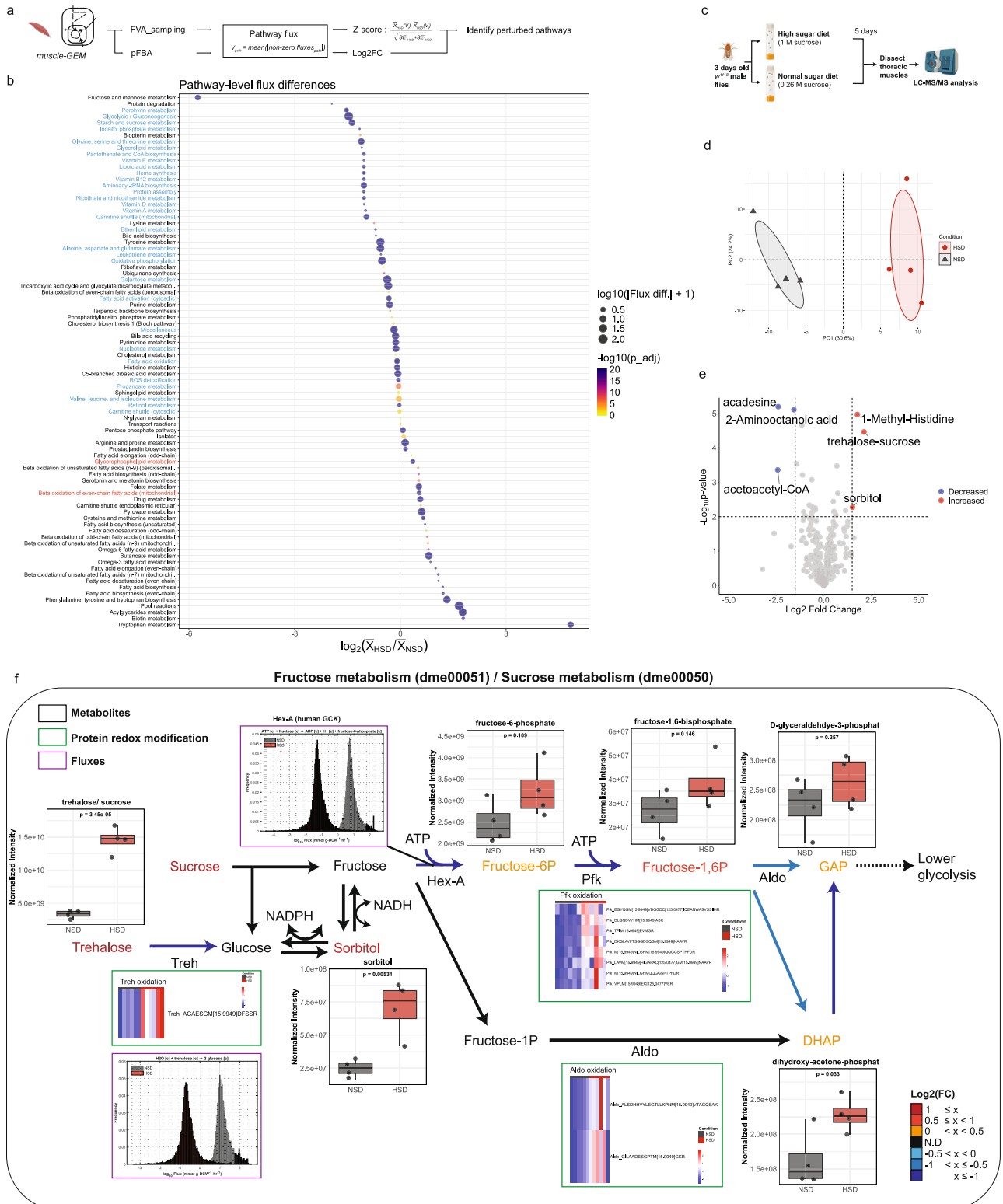

metabolism), lipid droplet turnover (pool reactions), butanoate metabolism, glycerophospholipid, and several amino acid metabolism pathways (e.g., tryptophan, phenylalanine-tyrosine).

To validate the predicted pathway-level perturbations, we performed targeted metabolomics in thoracic muscles in response to HSD (Fig. 6c). PCA revealed distinct metabolite profiles between HSD and NSD conditions (Fig. 6d). Among the six most significantly altered metabolites, three were involved in either fructose/sucrose metabolism (e.g., sorbitol and trehalose/sucrose) or butanoate metabolism

(e.g., acetoacetyl-CoA) ($|Log_2FC| > 2$ and $p$-value < 0.01), consistent with the pathways identified as highly perturbed (Fig. 6e and Supplementary Data 6b). To further investigate how these metabolite changes relate to fluxes and enzyme redox modifications, we created an integrated metabolic map focusing on fructose/sucrose metabolism (Fig. 6f). Beyond the perturbations in upper glycolysis, we found that Trehalase, an enzyme that converts trehalose to glucose, showed a decreased predicted flux ($Log_2FC = -5.4$) and significant oxidation at a methionine residue ($Log_2FC = 1.75$, $p = 0.006$). Collectively, these

**Fig. 6 | Pathway-level flux analysis reveals dysregulated fructose metabolism.**
**a** Schematic of the pathway-level flux analysis pipeline. FVA-sampling and pFBA were used to estimate pathway fluxes under HSD and NSD conditions, followed by calculation of Z-scores and log$_2$ fold changes to identify perturbed pathways. Elements created in BioRender. Moon, S. (2026) https://BioRender.com/1nasxv2.
**b** Pathway-level flux differences between HSD and NSD conditions. Each dot represents a pathway, with its size indicating absolute pathway flux difference and color representing adjusted *p*-value. The *x*-axis shows the log$_2$ fold change in pathway flux. Pathways highlighted in blue or red show commonly decreased or increased fluxes across both FVA-sampling and pFBA. **c** Schematic representing the targeted metabolite profiling of thoracic muscles from *w$^{1118}$* male flies fed either HSD or NSD. Elements created in BioRender. Moon, S. (2026) https://BioRender. com/1nasxv2. **d** PCA of thoracic metabolite profiles under HSD and NSD (*n* = 4). **e** Volcano plot showing significantly altered metabolites in thoracic muscles in

response to HSD. **f** Integrated schematic of fructose and sucrose metabolism (KEGG: dme00051 and dme00050), highlighting key metabolites, enzymes, and reactions. Box plots (black outline) display normalized metabolite levels under NSD and HSD. Center line is median; box limits are first and third quartiles; whiskers are 1.5 × interquartile range; points represent biologically independent replicates (*n* = 4). Statistical significance was assessed using a two-tailed unpaired *t*-test and the *p*-value is shown above the plot. Heatmaps (green outline) indicate significantly altered redox modifications for select enzymes (*p* < 0.05). Histograms (purple outline) show model-predicted flux distributions from flux sampling analysis (*n* = 10,000). Black arrows indicate reactions not detected (ND) in the muscle-GEM. Colored metabolite names and arrows represent log$_2$ fold changes (HSD/NSD) for both fluxes and metabolite levels, as shown in the color scale. Source data are provided as a Source Data file.

results demonstrate that our pathway-level flux analysis, supported by metabolomics and redox proteomics, effectively identifies candidate dysregulated metabolic pathways, such as fructose metabolism in muscle under HSD condition.

## Discussion

Modeling tissue-specific metabolism is essential to dissect the complex interplay of metabolic activities in multicellular organisms. In this study, we reconstructed 32 tissue-specific GEMs by integrating pseudo-bulk single-nuclei transcriptomics datasets with the manually curated generic *Drosophila* metabolic model. These tissue-specific GEMs enable quantitative comparisons of their metabolic network structures, representing a significant advance over existing *Drosophila* generic GEMs, such as Fruitfly1[19], FlySilico[17], and iDrosophila[18], which lack tissue-level metabolic analyses.

Furthermore, our tissue-specific GEMs offer distinct advantages by enabling network-level flux analyses that are difficult to achieve with pathway-specific or resource-intensive isotope tracing experiments. Specifically, using the muscle-GEM, we predicted active NAD(H)-dependent flux changes, identified reactions that may control glycolytic fluxes, and uncovered pathway-level flux perturbations under high sugar diet. These analyses were further integrated with other omics data, such as metabolomics and redox proteomics, confirming pathway perturbations and their link to redox-dependent enzyme modifications. Collectively, our in silico predictions generate testable hypotheses and prioritize key reactions and pathways for targeted experimental investigation, complementing the precise and tissue-specific genetic tools available in *Drosophila*.

Additionally, our tissue-specific GEMs, validated through metabolomics and pathway-level analysis, not only recapitulated known metabolic functions of individual tissue groups, but also revealed distinct tissue metabolism. For instance, fat body and oenocytes exhibited the highest overall reaction numbers, indicating a broad metabolic capacity of these tissues, particularly in carbohydrate and fatty acid metabolism. Conversely, muscle tissue, despite having fewer overall reaction numbers, showed the highest fraction of extracellular transport reactions and enriched reactions in protein degradation and lipoic acid metabolism. This suggests that muscle may serve as a major hub for systemic metabolite exchange and distribution. Indeed, muscle is known to be a major reservoir of amino acids, releasing them into circulation to support other tissue metabolism when needed[67]. As inter-organ communication is essential to maintain whole-body homeostasis and compensate for the limitations of individual tissue functions[2], future studies in evaluating transport reactions and their associated metabolic pathways using these tissue-specific GEMS may reveal additional layers of systemic metabolic coordination among tissues.

Moreover, we demonstrated the utility of our tissue-specific GEMs to evaluate muscle metabolism in response to high sugar diet. Our analyses revealed altered NAD(H)-dependent reactions and suggested

that regulation of glycolytic flux is distributed among several enzymes, including Aldo and GAPDH. Previous studies recognized GAPDH as a rate-limiting step in glycolysis, particularly in the context of aerobic glycolysis in cancer, where it regulates flux and is influenced by fructose 1,6-bisphosphate levels[68]. Our findings expand this role, demonstrating that GAPDH also acts as a regulatory node in muscle in response to HSD and through redox modifications of the enzyme. While GAPDH is typically known to undergo redox regulation at cysteine residues in response to oxidative stress[26,69], our data also revealed increased methionine oxidation. In fact, methionine oxidation has previously been shown to affect GAPDH function. For example, Samson et al., demonstrated that oxidation at Met-46 promoted GAPDH aggregation, a process linked to disease[70]. In our study, methionine oxidation occurred near the NAD$^+$ binding site, which may interfere with cofactor binding and enzymatic activity. Moreover, GAPDH enzyme levels, as well as most other glycolytic enzyme levels, were not significantly altered, indicating a potentially greater role of redox regulation on modulating glycolytic activity in this condition.

Finally, our muscle-GEM enabled predictions of additional dysregulated metabolic pathways, with fructose metabolism among the most substantially perturbed under HSD. While liver, intestine, and kidney are typically considered as primary sites of fructose metabolism due to high ketohexokinase expression, recent evidence showed that muscle can also metabolize fructose and affect muscle glucose handling[71]. Our pathway-level flux analysis predicted muscle could catabolize fructose and showed a significant decrease in the activity of Hex-A, a *Drosophila* hexokinase capable of phosphorylating fructose to fructose-6-phosphate in response to HSD[72]. The accumulation of metabolites such as sorbitol, trehalose and/or sucrose, along with increased oxidation of Trehalase, supported the perturbation of fructose and sucrose metabolism. Yet, these experimental data do not indicate whether fructose catabolic activities were increased or decreased, as elevated concentrations of upstream metabolites can increase flux through mass action. Future studies using fructose isotope tracing will be required to validate the direction and magnitude of flux changes within this pathway.

We acknowledge several limitations in this study. First, the choice of GEM reconstruction algorithm can influence model content and predictive performance[73]. When comparing our tissue-specific GEMs reconstructed through the tINIT algorithm with those generated by CORDA, we found that the final GEMs from CORDA contained fewer reactions and lower metabolic task pass rates compared to those generated by tINIT, but maintained similar metabolic structures for major tissue groups (Supplementary Fig. 6a–c). This underscores the importance of careful selection and parameter tuning of reconstruction algorithms to minimize potential biases. Second, we acknowledge the challenges of validating all 32 tissue-specific GEMs. Additional context-specific validation will be required to improve model accuracy. Third, using a single pseudo-biomass objective for flux balance analysis may not fully predict tissue- or context-specific flux

predictions. Incorporating tailored objectives functions could refine flux predictions across tissues. Fourth, most reactions in the GEMs retained the default flux bounds, representing computational conventions rather than experimentally derived limits. This underscores that our flux predictions are semi-quantitative, given the limited availability of comprehensive in vivo nutrients exchange data. Incorporating high-quality in vivo nutrients exchange rates will improve flux prediction accuracy. Finally, certain reactions within fructose metabolism were absent in the current muscle-GEM, highlighting a need for continued curation and targeted data integration to enhance model completeness and flux predictions.

In summary, we reconstructed tissue-specific GEMs for *Drosophila* and demonstrated the utility of the muscle-GEM to identify high sugar diet-induced metabolic dysregulation at both reaction and pathway levels. These GEMs provide a quantitative, systems-level framework that further complements the diverse experimental and genetic tools available for *Drosophila*, advancing metabolic investigations in this model organism.

## Methods

### Genome-scale metabolic model for *Drosophila*

We selected *Fruitfly1* as a base GEM[19], as it offered the comprehensive coverage of reactions ($n_{rxn} = 12,308$), metabolites ($n_{met} = 8117$), and genes ($n_{gene} = 1810$) compared to alternative GEMs such as *FlySilico* ($n_{rxn} = 363$, $n_{met} = 293$) or *iDrosophila1* ($n_{rxn} = 8230$, $n_{met} = 6990$, and $n_{gene} = 2388$)[17,18].

**Update of genes and gene-transcript-protein-reaction associations in Fruitfly1-GEM.** Fruitfly1.0 is a generic genome-scale metabolic model describing metabolic networks of *Drosophila melanogaster*[19]. To update the gene information within the model, we first used Gene List Annotation for Drosophila (GLAD) database, containing a metabolic gene set of 2629 genes (https://www.flyrnai.org/tools/glad/web/, Metabolic_vs3). We compared these genes with those listed in the GEM. We found that 1,465 genes from GLAD were not included. These missing genes included transcription factors, kinases, phosphatases, and dehydrogenases. Before adding those genes directly into Fruitfly1, we evaluated those genes had human orthologs by using DIOPT score[74]. Of 1465 genes analyzed, 564 genes (38.5%) were predicted to have human orthologs assigned with a high rank score[74]. Among those, 36 genes were already linked to reactions and gene-rules in human-GEM. Thus, we added these genes into Fruitfly1 (see Supplementary Data). For reactions that already had gene-rules (grRules), we integrated the new genes using the Boolean operator "or". To address gene redundancy, we investigated overlapping gene symbols within Fruitfly1 and found that three gene symbols (*Sur/sur, Argk2/CG5144*, and *Argk/Argk1*) were mapped to the same FlyBase gene IDs. To resolve these redundancies, we retained *Sur* (FBgn0028675), *Argk2* (FBgn0035957), *Argk1* (FBgn0000116). Following these updates, we named the revised model Fruitfly2. In summary, we added 36 missing genes (Supplementary Data 1a).

**Update the enzyme commission number.** To update Enzyme commission (EC) numbers, a systematic classification of enzymes based on the reactions, we used KEGG ID to map reactions with EC numbers within the Fruitfly1-GEM. We first examined whether the reactions present in Fruitfly1-GEM were associated with EC numbers and found that 7411 out of 12,038 reactions (approximately 61.6%) lacked EC number annotations in Fruitfly2. We then assessed whether these reactions had associated KEGG reaction IDs, which could be used to retrieve EC numbers. Only 161 reactions were linked to KEGG reaction IDs, of which 78 were associated with EC numbers. Based on this information, we assigned EC numbers to these 78 reactions. Additionally, by using gene–KEGG reaction ID associations, we assigned EC numbers to a further 576 reactions. In total, we assigned EC numbers to

654 reactions in Fruitfly2 (Supplementary Data 1b),and designated the updated GEM as Fruitfly3.

### Reconstruction of tissue-specific genome-scale metabolic models in *Drosophila*

To generate tissue-specific GEMs, we integrated Fruitfly3 with tissue-specific pseudo-bulk single-nuclei transcriptomics data for 32 individual tissues from Fly Cell Atlas, while using the Task-driven Integrative Network Inference for Tissues (tINIT) algorithm, implemented via the getINITModel2 function from the Human-GEM repository (Supplementary Data 1c)[7,75]. For generating tissue-specific pseudo-bulk gene expression data, we first retrieved the Seurat objects from Fly Cell Atlas dataset and used *AggregateExpression* function in Seurat to pseudo-bulk the counts per tissue[76,77]. Gene expression values were converted to counts per million (CPM), rather than TPM or FPKM, by normalizing to library size and scaling by a factor of 1e6[78]. The resulting normalized gene expression values are expressed in unit of CPM. To incorporate these gene expression data as input for tINIT, we applied a global and relaxed expression threshold of 1, selected based on the distribution of average gene expression levels across tissues (mean $- 1 \times$ SD = 1.02). This ensured the inclusion of moderately expressed genes while maintaining consistency across all tissues.

In addition to transcriptomic data, tINIT requires a set of essential metabolic tasks as inputs. Accordingly, we used a 57 predefined essential metabolic tasks, adapted from prior study (Supplementary Data 1d)[75]. These tasks span key metabolic functions, including rephosphorylation of nucleoside triphosphates, de novo synthesis of nucleotides, uptake of essential amino acids, protein turnover, electron transport chain and TCA, beta-oxidation of fatty acids, de novo synthesis of phospholipids, growth based on media components, and synthesis of vitamins and cofactors. Although these tasks were originally designed for human GEM reconstruction, we reviewed available *Drosophila* literature, including KEGG pathway annotations and FlyBase resources, and found no direct evidence that these core tasks would not be performed in *Drosophila*. While we did identify *Drosophila*-specific pathways in the KEGG database, such as dorso-ventral axis formation, insect hormone biosynthesis, and Toll and Imd signaling pathway, we did not include these as essential tasks for reconstruction, as their essentiality has not been validated across all 32 tissues.

To compare the GEM reconstruction outcomes, we also applied the Cost Optimization Reaction Dependency Assessment (CORDA) algorithm[5] to reconstruct tissue-specific GEMs using the same dataset used for tINIT. CORDA categorizes reactions into essential (ES), present (PR), and not present (NP) based on gene expression levels. Specifically, reactions associated with genes expressed above the mean were designated as ES; those with expression between one and the mean as PR; and those less than or equal to 1 as NP.

### Metabolic network structure, subsystem coverage, and metabolic task analysis

**Metabolic network analysis.** The analysis of metabolic network structure, subsystem coverage, and metabolic tasks was performed following a previously established protocol[75]. In brief, to compare model structures, we used *compareMultipleModels* function from the RAVEN package. This function constructs a binary reaction matrix, where rows represent reactions and columns represent individual GEMs. A value of 1 indicates the presence of a reaction in a given GEM, while a value of 0 indicates its absence. For visualization, we applied t-distributed stochastic neighbor embedding (t-SNE) to the binary reaction matrix, using the Hamming distance as the similarity metric. The resulting two-dimensional projection revealed the relative proximity of models based on their metabolic reaction content. Additionally, we performed hierarchical clustering to classify the tissue-specific GEMs. A Euclidean distance matrix was computed from the same

binary reaction matrix, followed by average linkage clustering to generate a dendrogram. Cluster assignments were defined by partitioning the dendrogram into a fixed number of groups, capturing major tissue classes such as muscle, fat body/oenocyte, gut, glia, and neurons. These clusters were annotated and visualized in the t-SNE plot.

**Quantification of cluster overlap using Jaccard index.** To assess the agreement between GEM-derived and gene expression-based tissue clusters, we calculated the Jaccard index. This metric quantifies the similarity between two sets as the size of their intersection divided by the size of their union, ranging from 0 (no overlap) to 1 (complete overlap)[47], as follows:

$$J(A, B) = \frac{|A \cap B|}{|A \cup B|}$$

where A and B represent the sets of tissues assigned to a given cluster based on GEM-derived and gene expression-derived clusters, respectively.

We first performed hierarchical clustering on both datasets. For tissue-specific GEMs, clustering was based on the binary reaction presence matrix, as described in the metabolic network analysis method. For gene expression-based clustering, we applied hierarchical clustering to pseudo-bulk expression data. Jaccard indices were then calculated by comparing each pair of clusters. For glial tissues, which were distributed across three distinct GEM-based clusters, the corresponding Jaccard indices were averaged to represent them as a unified group.

**Metabolic subsystem coverage analysis.** Variation in model structures was further investigated by analyzing the coverage of metabolic subsystems in each tissue-specific GEMs. Subsystem coverage (SC) refers to the total number of reactions present in a specific metabolic subsystem. For each subsystem, the SC was calculated for individual tissue-specific GEMs and compared to the average subsystem coverage across all tissue-specific GEMs ($SC_{mean}$). The relative deviation from the mean coverage was calculated using the following formula:

$$\% \, differences \, in \, subsystem \, coverage = \frac{SC - SC_{mean}}{SC_{mean}} \times 100\%$$

To facilitate the visualization, we applied a threshold of 100%, showing only those subsystems in which at least one tissue-GEM exhibited more than 50% differences from the mean subsystem coverage.

**Metabolic task analysis.** To evaluate and compare each GEM's metabolic functionality, we assessed the ability of each GEM to perform a set of curated 219 metabolic tasks. This list was primarily based on the updated metabolic task collection from Richelle et al.[36]., which includes 195 tasks. This list also includes system and subsystem categories. Given the well-characterized role of trehalose metabolism in *Drosophila*, we added eight trehalose biosynthesis tasks with different substrate conditions. Additionally, we compared this updated list with the earlier Human-GEM task set[75]. From this comparison, we added an additional 16 tasks related to energy metabolism (eight oxidative phosphorylation, one Krebs cycle, four glycolysis) and carbohydrate metabolism (two glycogen metabolism), resulting in the final list of 219 metabolic tasks (Supplementary Data 1k). For clarity, this list is distinct from the set of 57 essential metabolic tasks used for tissue-specific GEM reconstruction (Supplementary Data 1d).

Task performance was evaluated using *checkTasks* function in the RAVEN toolbox. A task was considered "passed" (assigned a value of 1) if the model could carry flux through the required reactions to convert the defined input metabolite(s) into the specified output product(s), indicating that the metabolic function is feasible within the network[79]. Conversely, a task was considered "failed" (assigned a value of 0) if no such flux could be achieved, suggesting that the function was not supported by the model. Additionally, we used Fisher's Exact test to evaluate the association between individual tissue-specific GEMs and metabolic systems, defined as groups of functionally related metabolic tasks[36].

## Fly stocks and maintenance

All flies were reared at 25 °C and 60% humidity with a 12-h on/off light cycle on standard laboratory food. Standard laboratory food is made as follows: 12.7 g/liter deactivated yeast, 7.3 g/liter soy flour, 53.5 g/liter cornmeal, 0.4% agar, 4.2 g/liter malt, 5.6% corn syrup, 0.3% propionic acid, and 1% Tegosept/ethanol. The semi-defined synthetic media for high sugar and normal diet were made according to the previously reported recipe[80], with a slight modification of sugar concentration. The normal diet (NSD) consisted of 10 g/L of agar, 80 g/L of yeast, 20 g/L of yeast extract, 20 g/L of peptone, 262 mM of sucrose, 0.3% propionic acid, and 1% Tegosept/ethanol. The high sugar diet (HSD) consisted of the same amount of all the components except 1 M of sucrose. For diet intervention experiments, we collected 3 days old $w^{1118}$ male flies and transferred approximately 30 flies to vials containing either NSD or HSD. The food was replaced every two days. Bloomington *Drosophila* stock center: attp40 (36304) and UAS-GAPDH1 RNAi (62212). Laboratory stocks: $w^{1118}$ and *w; tub-Gal80[TS]; MHC-Gal4/TM6b*.

## Intracellular metabolites extraction and LC/MS analysis

Metabolites quantification in *Drosophila* were adapted from a published protocol[81,82]. 20 thoraces, head, gut, and abdomen were dissected in ice-cold 0.9% normal saline buffer after placing the flies under light CO2 anesthesia. Dissected thoraces were immediately placed in 1.5 mL tubes in dry ice. 450 µL of cold 80% (v/v) aqueous methanol were added to the tubes, and intracellular metabolites were extracted by homogenizing tissues with pellet pestle for 1 min by repeating a cycle of 20 s of grinding and 10 s of break. Samples were centrifuged at maximum speed for 10 min in the cold room. Supernatants were transferred to a new 1.5 mL tube, pelleted by vacuum centrifugation using speedvac and stored in −80 °C until mass spectrometry analysis.

Samples were re-suspended using 22 µL HPLC grade water for mass spectrometry. 5–7 µL were injected and analyzed using a hybrid 6500 QTRAP triple quadrupole mass spectrometer (AB/SCIEX) coupled to a Prominence UFLC HPLC system (Shimadzu) via selected reaction monitoring (SRM) of a total of 300 endogenous water soluble metabolites for steady-state analyses of samples. Some metabolites were targeted in both positive and negative ion mode for a total of 311 SRM transitions using positive/negative ion polarity switching. ESI voltage was +4950 V in positive ion mode and −4500 V in negative ion mode. The dwell time was 3 ms per SRM transition and the total cycle time was 1.55 s. Approximately 9–12 data points were acquired per detected metabolite. For targeted 13 C flux analyses, isotopomers from ~140 polar molecules were targeted with a total of 460 SRM transitions. Samples were delivered to the mass spectrometer via hydrophilic interaction chromatography using a 4.6 mm i.d × 10 cm Amide XBridge column (Waters) at 400 µL/min. Gradients were run starting from 85% buffer B (HPLC grade acetonitrile) to 42% B from 0 to 5 min; 42% B to 0% B from 5 to 16 min; 0% B was held from 16 to 24 min; 0% B to 85% B from 24 to 25 min; 85% B was held for 7 min to re-equilibrate the column. Buffer A was comprised of 20 mM ammonium hydroxide/20 mM ammonium acetate (pH = 9.0) in 95:5 water:acetonitrile. Peak areas from the total ion current for each metabolite SRM transition were integrated using MultiQuant v3.2 software (AB/SCIEX).

## Metabolites profiling data analysis

Statistical analysis was performed using MetaboAnalystR[46]. The hierarchical clustering analysis and heatmap was generated using *PlotHeatMap* function, in which *hclust* function used to normalize the data across the samples and the Euclidean distance and ward.d were used as parameters for clustering algorithm. Principal component analysis was performed using *PCA.anal* function. In comparison of the means of the normalized metabolite peaks, the Anova test was used using *ggpubr* package in R. For evaluation of the normalized metabolites peak areas, the peak areas were normalized to the protein mass of individual tissues.

## KEGG over-representation analysis for metabolomics

To identify pathway-level metabolic changes, we performed over-representation analysis (ORA) using KEGG pathway annotations[63]. First, we constructed a reference pathway and metabolite database by utilizing the *keggrest* R package, retrieving all *D. melanogaster* pathways (e.g., dme00010, dme00020, etc) via *keggList*, and extracting associated metabolites for each pathway using *keggGet*, removing duplicate metabolites to create a unique reference compound list. A total of 105 *Drosophila* pathways and their associated compounds were compiled to serve as the reference database for subsequent over-representation analysis. Region-specific metabolomics data, containing KEGG IDs from MetaboAnalyst, were then analyzed through ANOVA testing for significant metabolite alterations across regions. Log$_2$ fold changes (log$_2$FC) were calculated relative to the mean intensity of other tissues, and metabolites with $p < 0.05$ and log$_2$FC > 0.5 were considered significantly enriched. For the abdomen, due to lower metabolite coverage, a relaxed threshold log$_2$FC > 0 was used, maintaining comparable input metabolite sizes compared to other regions. ORA was conducted using a hypergeometric test to calculate statistical over-representation, comparing altered metabolites against the background of all detected metabolites, with Benjamini–Hochberg adjustment for multiple testing, considering pathways with adjusted $p < 0.05$ as significantly enriched pathways. Enrichment results were visualized as bubble plots, with bubble size representing the enrichment ratio (the proportion of observed significant metabolites within a pathway relative to the expected proportion based on the reference dataset) and color indicating log-transformed adjusted $p$-values.

## Quantification of pathway overlap using weighted Jaccard index

To evaluate the degree of overlap between metabolomics-derived enriched pathways and those predicted by tissue-specific GEMs, we calculated a weighted Jaccard index. A traditional Jaccard index measures the similarity between two sets as the size of their intersection divided by their union[47]. We extended this to incorporate significance derived from metabolomics data, allowing pathways with stronger statistical support to contribute more to the similarity score[48]. For each experimentally enriched pathway, we assigned a weight based on the negative log$_{10}$ of its adjusted $p$-value ($-\log_{10}(q)$), normalized across all pathways within each region. For GEM-based enriched pathways, we determined pathways with reaction content above the median across tissues as the enriched pathways. For each region–tissue pair, the weighted Jaccard index was calculated as:

$$J_{weighted}(A, B) = \frac{\sum_{i \in A \cap B} \min(w_A(i), w_B(i))}{\sum_{i \in A \cup B} \max(w_A(i), w_B(i))}$$

where:

A and B are the sets of enriched pathways in the experimental region and predicted tissue-specific GEMS, respectively;

$w_A(i)$ refers to the normalized significance weights assigned to pathway $i$ from experimentally determined enriched pathways for

region A, calculated as $w_A(i) = \frac{-\log_{10}(q_i)}{\sum_{j \in A} -\log_{10}(q_j)}$, where $q_i$ is the adjusted $p$-value for pathway $i$. For any pathway $i$ that is not part of set A (e.g., $i \in A \cup B$ and $i \notin A$), $w_A(i) = 0$. The sum of all $w_A(i)$ equals 1;

$w_B(i)$ refers to the normalized weights assigned to GEM-based enriched pathways. Since the enriched pathways identified from GEM do not have statistical significance values, each pathway $i$ in set B is assigned an equal weight, calculated as $w_B(i) = 1/n_B$, where $n_B$ is the number of enriched pathways in that tissue-specific GEM (set B). For any pathway $i$ that is not part of set B (e.g., $i \in A \cup B$ and $i \notin B$), $w_B(i) = 0$. The sum of all $w_B(i)$ equals 1.

To assess the statistical robustness of the weighted Jaccard index, we performed 10,000 bootstrap resampling of the experimental enriched pathway dataset. For each bootstrap iteration, enriched pathways were randomly sampled, and the weighted Jaccard index was recalculated against model-predicted tissue-specific enriched pathways. This generated a distribution of indices for each region-tissue pair. We then used one-way ANOVA followed by Bonferroni-corrected post-hoc tests to determine whether a specific region (e.g., thorax) exhibited significantly higher similarity to a given tissue-GEM (e.g., Muscle-GEM) compared to other regions. Significance levels were visualized on box plots to highlight region-specific alignment between experimental data and model predictions.

## Definition of HSD-muscle-GEM

We defined a high sugar diet (HSD)-muscle-GEM by constraining reaction rates to recapitulate key known features of type 2 diabetic muscle based on the literature evidence[28,29,33,34,52–54]. These constraints included reduced upper bounds of reactions rates in glucose uptake, glyceraldehyde-3-phosphate dehydrogenase, citrate synthase, oxoglutarate dehydrogenase, succinate dehydrogenase, and fumarate hydratase (Supplementary Data 3a). Since *Drosophila* also utilizes trehalose as a fuel source, and its circulating levels increase in response to HSD[13,83], we also constrained the upper bounds of trehalose uptake rates. Because glycolysis and TCA cycle activities are closely linked to NAD(P)H redox metabolism[56], we updated reaction directionality of redox-related reactions based on supporting evidence from the literature[9,84–89] (Supplementary Data 3a). Furthermore, to focus the model on metabolic pathways relevant to our measured metabolites, we constrained the upper bounds of exchange rates for the undetected metabolites to one tenth of the default upper bound (Supplementary Data 3b). Lastly, given the muscle tissue serving as a major tissue for energy and redox metabolism, we included artificial demand reactions for ATP production under HSD, NAD$^+$ regeneration, and NADPH regeneration. These reactions reflect fundamental physiological needs of muscle in *Drosophila*: ATP production was required for flight[90], NAD$^+$ regeneration to sustain glycolysis[91], and NADPH regeneration to mitigate oxidative stress[92,93]. While we originally intended to use these reactions as objective functions, similar to prior approaches modeling energy and redox metabolism[84,94], they were not used as objectives in our FBA simulations of HSD-muscle-GEM, but were still retained as auxiliary buffer reactions for energy and cofactor metabolism. Simulations were performed using a biomass objective function (reaction ID: *MAR00021*), which captures broad metabolic maintenance and biosynthesis requirements of muscle tissue. Details of this objective function are provided in the Flux Balance Analysis section.

## Flux balance analysis

Flux balance analysis (FBA) was performed to determine the optimal metabolic flux distributions that maximizes a defined objective function, while satisfying stoichiometric constraints under steady state assumption[95]. A metabolic network is represented by a stoichiometric matrix $S$ of size m × n, in where m and n represents the number of

metabolites and reactions, respectively. The elements in $S_{ij}$ corresponds to the stoichiometric coefficient of metabolites $i$ in reaction j. The relationship between metabolites and reaction rates is given by the following equation:

$$S\mathbf{v} = \frac{d\mathbf{m}}{dt}$$

where $\mathbf{v}$ is the $n \times 1$ vector of reaction rate vector and $\mathbf{m}$ is an $m \times 1$ vector of metabolite concentrations. Under the steady-state assumption, the concentration of metabolites does not change over time, making the right side equal to 0. FBA solves a linear optimization problem to determine flux distributions that satisfy this constraint, formulated as:

$$\text{maximize} \, \mathbf{c}^T \mathbf{v},$$

$$\text{subject to } S\mathbf{v} = 0,$$

$$\text{and} \, \mathbf{v}_{min} \leq \mathbf{v} \leq \mathbf{v}_{max},$$

where

S : Stoichiometry matrix($m \times n$)
v : flux vector representing reaction rates at steady state($v \in R^n$)
c : vetor defining the linear objective function
$v_{min}$, $v_{max}$ : Lower and upper bounds for each flux

For FBA simulations, we used a biomass objective function (reaction ID:*MAR00021*), which aggregates the demand for key metabolic precursors—including amino acids (e.g., alanine, glutamine, valine), lipids (e.g., cholesterol, lipid droplets), glycogen, nucleotides (DNA, RNA), and metabolic pools (e.g., phospholipids, cofactors, vitamins)—into a single pseudo-biomass reaction, as follows:

$$
\begin{aligned}
&CL_{pool}[c] + DNA[n] + DNA - methylcytosine[n] + PI\,pool[c] + RNA[c] \\
&\quad + SM\,pool[c] + alanine[c] + arginine[c] + asparagine[c] \\
&\quad + aspartate[c] + cholesterol[c] + cholesterol - ester\,pool[r] \\
&\quad + cofactors\,and\,vitaminc[c] + cysteine[c] + glutamate[c] \\
&\quad + glutamine[c] + glycine[c] + glycogen[c] + histidine[c] \\
&\quad + isoleucine[c] + leucine[c] + lipid\,droplet[c] + lysine[c] \\
&\quad + methionine[c] + phenylalanine[c] + phosphatidate - LD \\
&\quad - TAG\,pool[c] + proline[c] + serine[c] + threonine[c] \\
&\quad + tryptophan[c] + tyrosine[c] + valine[c] \rightarrow biomass[c]
\end{aligned}
$$

This formulation was used here as a general proxy for basal cell maintenance, including protein turnover, membrane lipid remodeling, and nucleotide recycling. These processes remain active and physiologically relevant to tissues like muscle[67,96,97]. Thus, rather than using energy and redox demands as sole objective functions, which could bias the solution space or do not recapitulate other core metabolic activities, we adopted this biomass objective function to capture broader metabolic requirements. The resulting flux through this reaction was approximately 1.1 (arbitrary unit). While this value represents the demand for cell biosynthesis and maintenance, the energetic and precursor requirements for synthesizing biomass are distributed among the various reaction fluxes throughout the metabolic network.

## Flux variability analysis
Flux Variability Analysis (FVA) was used to evaluate the range of feasible fluxes for each reaction under sub-optimal conditions, characterizing the flexibility of individual reactions[98]. FVA involves solving two optimization problems—one to maximize and one to minimize

each flux $v_i$ of interest,

$$\text{maximize/minimize} \, v_i,$$

$$\text{subject to } S\mathbf{v} = 0,$$

$$\mathbf{c}^T \mathbf{v} \alpha \times z_{common,opt},$$

$$\text{and} \, \mathbf{v}_{min} \leq \mathbf{v} \leq \mathbf{v}_{max},$$

where $\alpha$ is the optimality parameter ($0 \leq \alpha \leq 1$), where $\alpha = 1$ corresponds to the fully optimal solution obtained from FBA. In this study, we set $\alpha = 0.9$ to allow a small deviation from the optimal solution, enabling exploration of alternative flux distributions that are biologically plausible while still near-optimal. We used flux variability analysis to probe the range of fluxes (differences between Vmax and Vmin) as a measure of feasible flux range, not as a proxy for statistical uncertainty.

## Flux sampling analysis
We performed flux sampling analysis after unconstrained reaction bounds with the minimum and maximum flux values obtained from flux variability analysis (FVA). To explore the feasible solution space, we applied the *GpSampler* algorithm, which implements the Artificial Centering Hit-and-Run[99]. A total of 10,000 randomly sampled flux distributions were generated using uniformly distributed initial points.

## Parsimonious flux balance analysis (pFBA)
Parsimonious enzyme usage flux balance analysis (pFBA) is a commonly used extension of standard flux balance analysis that identifies flux distributions with minimal total flux while achieving the optimal value of a specified objective (e.g., biomass production)[100]. The rationale is that cells are expected to minimize enzyme usage and avoid unnecessary internal cycling, leading to more realistic and interpretable flux predictions. It uses a bilevel optimization in which the objective function is first optimized using FBA, followed by the minimization of total flux through all gene-associated reactions. The second part of the optimization is expressed as follows:

$$\min \sum_{j=1}^{m} v_{irrev,j}$$

$$s.t. \, v_{objective,lb} = v_{objective,FBA},$$

$$S_{irrev} * \mathbf{v_{irrev}} = 0$$

$$0 \leq v_{irrev,j} \leq v_{max},$$

where m is the number of irreversible reactions in the network, $v_{objective,FBA}$ is the optimal flux value obtained from FBA and $v_{objective,lb}$ is the lower bound for the objective function for the second optimization problem.

## Differential flux analyses
To identify reactions with altered fluxes between HSD and NSD conditions, we calculated the difference in flux values for each reaction across FBA, pFBA, and FVA_sampling analyses. Flux values from FBA and pFBA represent single optimal solutions, whereas for FVA_sampling, median flux values across the sampling points were used as a representative flux value. Reactions were classified as increased or decreased based on a flux difference threshold of ±1, which can be adjusted for stringency. This analysis produced three independent sets

of perturbed reactions for each method. Then, we identified the commonly perturbed reactions shared across all three methods and visualized their overlaps using *UpSet* and *ggvenn* functions in R. For these shared reaction sets, we assigned each reaction to its corresponding subsystem and associated gene(s). Within each direction (increased or decreased), the top three subsystems with the largest numbers of perturbed reactions were selected. Gene–subsystem relationships were visualized as bipartite network graphs, where node size corresponds to the number of associated reactions and edge width represents the number of shared genes between subsystems and reactions. Networks were generated using the *igraph* (v1.3.5) and *ggraph* (v2.1.0) in R.

For FVA-sampling analysis of NAD(H)-dependent reactions, we additionally performed two-sample *Z*-tests to evaluate the statistical significance of flux changes between NSD and HSD conditions[101]. Z-score was calculated as follows:

$$Z_i = \frac{\bar{X}_{HSD}(v_i) - \bar{X}_{NSD}(v_i)}{\sqrt{\frac{\sigma^2_{HSD}(v_i)}{n_{HSD}} + \frac{\sigma^2_{NSD}(v_i)}{n_{NSD}}}}$$

where i represents each reaction, $\bar{X}_{cond}(v_i)$ is the average of the sampled fluxes of each reaction for condition (NSD or HSD), $\sigma^2_{cond}(v_i)$ is the sample variance, and $n_{cond}$ is the number of flux samples per condition. Two-tailed *p*-values were computed using the standard normal cumulative distribution function, and Bonferroni correction was applied to adjust for multiple testing.

## Evaluation of maximum NADH production capacity

To assess the network's maximum NADH production capacity, we introduced an artificial total NADH demand reaction ($NADH \rightarrow NAD^+ + H^+$) accounting for relevant compartments (cytosol, mitochondria, and peroxisome) into the muscle-GEM. Specifically, the reaction is represented as:

$$NADH[c] + NADH[m] + NADH[p] \rightarrow NAD^+[c] + H^+[c] + NAD^+[m]$$
$$+ H^+[m] + NAD^+[p] + H^+[p]$$

where [c], [m], and [p] refer to the cytosol, mitochondria, and peroxisome, respectively. As solely maximizing the NADH demand reaction may yield biologically nonviable solutions, we first performed FBA by maximizing the original pseudo-biomass objective function[84]. Then, we set the lower bounds of this reaction at 50% of the optimized solution and performed pFBA to maximize the NADH demand reaction.

## Sensitivity analysis

To assess the control of individual glycolytic reactions on overall pathway flux, we performed a sensitivity analysis. This analysis is conceptually analogous to metabolic control analysis, where flux control strength is quantified as the relative change in pathway output flux in response to variations in individual enzyme activities[102,103]. It also aligns with general sensitivity analyses of reaction rates used in reaction networks[85,104]. For each glycolytic reaction, the baseline flux was defined as the median flux of 5000 sampling data obtained from FVA. Then, each reaction was perturbed by constraining its flux to 95, 90, 70, and 50% of the baseline, and FVA-sampling was repeated to obtain the perturbed flux distributions. The total pyruvate consumption flux was used as a proxy for glycolytic output flux, because unlike mammalian systems where lactate production rate by lactate dehydrogenase (LDH) is commonly used as the glycolytic output, LDH activity is relatively low in insect muscle[105], and pyruvate is substantially converted for the production of other metabolites such as alanine and acetate[106,107]. The normalized sensitivity coefficient ($S_i$) was

calculated as follows:

$$Normalized\ sensitivity\ coefficient(S_i) = \frac{\Delta v_{total\_pyr\_con}}{\Delta v_i} \times \frac{v_{i,base}}{v_{total\_pyr\_con,base}}$$

where $v_{total\_pyr\_con,base}$ represents the sum of fluxes through all pyruvate-consuming reactions at baseline, $\Delta v_{total\_pyr\_con}$ represents the difference in the summed flux of all pyruvate-consuming reactions between the baseline and perturbed states, $v_i$ represents the baseline flux of with glycolytic reaction, and $\Delta v_i$ represents the difference in flux of the ith reaction between the baseline and perturbed states.

## Pathway-level flux analysis

To evaluate the pathway-level flux changes and statistical significance between NSD and HSD conditions, we defined pathway flux ($V_i$) as the average of non-zero flux magnitudes within each pathway. For FVA-sampling analysis, we used 10,000 flux sampling points per condition and performed a two-sample *Z*-test, adapting the approach from a previous study[101]:

$$Z_i = \frac{\bar{X}_{HSD}(V_i) - \bar{X}_{NSD}(V_i)}{\sqrt{\frac{\sigma^2_{HSD}(V_i)}{n_{HSD}} + \frac{\sigma^2_{NSD}(V_i)}{n_{NSD}}}}$$

where i represents each pathway, $\bar{X}_{cond}(V_i)$ is the average of the pathway flux across all sampling points for condition (NSD or HSD), $\sigma^2_{cond}(V_i)$ is the sample variance of pathway flux across all sampling points for each condition, and $n_{cond}$ is the number of flux sampling points per condition. Two-tailed *p*-values were computed using the standard normal cumulative distribution function. For pFBA, which yields a single optimal flux solution per condition, no statistical test was performed and pathway flux changes were quantified using the log2 fold change between HSD and NSD.

## $^{13}$C-isotopic labeling experiments and mass isotopomer analysis

We prepared the normal and high sugar diet with 13 mM and 50 mM of [U-$^{13}$C$_6$]glucose (Cambridge Isotope Laboratories, Inc. CLM-1396), respectively, maintaining the molar ratio of the labeled glucose to sucrose consistent. For $^{13}$C-isotopic labeling experiments, we collected 3 days old $w^{1118}$ male flies and placed approximately 30 flies in vials containing either NSD or HSD with isotope tracers at room temperature. The food was replaced every two days. On day five, twenty thoraces were dissected per biological replicate and intracellular metabolites were extracted using 80% (v/v) aqueous methanol. Q1/Q3 SRM transitions for incorporation of 13C-labeled metabolites were established for polar metabolite isotopomers, and data were acquired by LC-MS/MS. Peak areas were generated using MultiQuant version 2.1 software. The natural isotope abundance was corrected using AccuCor (Github: https://github.com/XiaoyangSu/AccuCor)[108].

## Switch-tagging redox proteomics

Muscle tissues were solubilized in 4% SDS in 100 mM triethylammonium bicarbonate pH 7.0 lysis buffer and digested using S-Trap micro columns (Protifi, Huntington, NY) following the manufacturer's procedure. Each sample was loaded onto individual Evotips for desalting and washing. The Evosep One system (Evosep, Odense, Denmark) was used to separate peptides on a Pepsep colum (150 um inter diameter, 15 cm) packed with ReproSil C18 1.9 um, 120 A resin using pre-set 15 samples per day gradient. The system was coupled to the timsTOF Pro mass spectrometer (Bruker Daltonics, Bremen, Germany) via the nano-electrospray ion source (Captive Spray, Bruker Daltonics). Raw data files conversion to peak lists in the MGF format, downstream identification, validation, filtering and quantification were managed using FragPipe version 13.0. MSFragger version 3.0 was used for

database searches against a *Drosophila* database with decoys and common contaminants added.

## Prediction of *Drosophila* GAPDH protein complexes using AlphaFold-Multimer

For the prediction of protein complexes using AlphaFold-Multimer (AFM), we employed LocalColabFold version 1.5.2[109]. It integrates AFM version 2.3.1[110] and utilizes MMseqs2 for generating multiple sequence alignments. Computations were performed on the Harvard O2 high-computing cluster. Our prediction involved five models for each complex, each undergoing five recycling iterations. The model displaying the highest prediction quality was selected for each protein complex. The protein complexes examined in this study include *Drosophila* GAPDH1 homodimer, GAPDH1:GAPDH2 heterodimer, and GAPDH2 homodimer. The prediction results have been made publicly available on the Fly Predictome website (https://www.flyrnai.org/tools/fly_predictome/web/)[111]. For visualization of the predicted structures we utilized ChimeraX[112].

## KEGG over-representation analysis (ORA) for redox proteomics

Similar to the metabolomics-based ORA analysis, we conducted KEGG over-representation analysis using proteins with significantly altered oxidation states. Significantly altered peptides of proteins were identified by calculating the $\log_2$ fold change between high sugar diet and normal sugar diet conditions. Statistical significance was determined using an unpaired two-sample *t*-test, with *p*-values corrected for multiple testing using the Benjamini–Hochberg (FDR) method. Proteins were considered significantly altered if they met the thresholds of $|\log_2 FC| > 0.5$ and adjusted *p*-value $< 0.05$. After obtaining a list of significantly altered protein sets, we performed KEGG over-representation analysis using the *clusterProfiler* R package. First, protein names were converted to gene symbols. Next, these gene symbols were mapped to entrez IDs using the *bitr* function in the *org.Dm.eg.db* database. Only unique Entrez Gene IDs were retained for downstream analysis. Finally, over-representation analysis was performed using the *enrichKEGG* function. The enrichment results were visualized using dot plots using *dotplot* function.

## Climbing assay

Flies were tested for vertical climbing ability at day 5 and 10[82,113]. In this assay, male flies separated by genotypes were transferred to empty vials. Flies were tapped three times and observed for 10 s. The percentage of flies that climbed above 5 cm was recorded. Consecutive trials were separated by 30 s of rest.

## RNA extraction and qRT-PCR

RNA was extracted and analyzed by qRT-PCR[49]. Five thoraces were collected per sample. Samples were homogenized in TRIzol reagent (Ambion). The RNAs were isolated and purified using Direct-zol RNA MicroPrep columns (Zymo Research) according to the manufacturer instructions. Reverse transcription was done by using iScript cDNA Synthesis Kit (Bio-Rad). qRT-PCR was done in a CFX96 Real-Time System (Bio-Rad) using iQ SYBRGreen Supermix (Bio-Rad). Relative mRNA levels were calculated using the ΔΔCt method. For thoraces, values were normalized to the housekeeping gene a-tubulin84.

## Quantification of cellular NADH/NAD+

Measurements of NADH/NAD+ was adapted from a published protocol[86], using the NADH/NAD+ Glo Assay (Promega: G9071, G9072) with a modification in manufacturer instruction. In brief, five thoraces were dissected, weighted, and stored in 1.5 mL tubes in dry ice. 300 μL of PBS with 1% DTAB solution was added the tubes and the thoraces were homogenized with the pellet and homogenizer. 110 μL of the cell lysates were split into two separate ice-cold 1.5 mL tubes: one for NADH

and the other for NAD+. For NADP+ tube, 55 μL of 0.4 M HCl with 1 mM ascorbate were added. We added ascorbic acid to prevent the oxidation of NADH into NAD+ under low pH, avoiding the overestimation of intracellular NAD+ levels[114]. The other NADH tube remained untreated. Both tubes were placed into the heat block with 60 °C temperature for 20 min to destroy NAD+. Tubes were equilibrated at room temperature for 10 min. 55 μL 0.5 M Trizma/base was added to the NAD+ tubes to neutralize the acid, while 110 μL of HCl/ascorbate/base buffer was added to the untreated NADH tube. Afterwards, NADH, NAD+, and the ratio were measured following the manufacturer's instruction with the generation of standard curves with known concentration.

## Data analysis, statistics, and reproducibility

Statistical analyses were performed in GraphPad Prism (v9.5.1), MATLAB (2020b), or R (4.2.1). For analysis of two groups, including metabolomics, isotope tracing, and redox proteomics between HSD and NSD conditions, Welch's unequal variances two-sided *t*-test was used to calculate a *p*-value. For metabolomics data involving multiple group comparisons, one-way ANOVA was conducted followed by Benjamini–Hochberg (BH) correction for controlling the false discovery rate (FDR). The significance was represented as follows: $*P < 0.05, **P < 0.01, ***P < 0.001, ****P < 0.0001$. In pathway enrichment analyses, hypergeometric tests were applied to identify over-represented pathways. To account for multiple comparisons, *p*-values were adjusted for false discovery rate (FDR) using the Benjamini–Hochberg (BH) method. For weighted Jaccard index analyses, pathway enrichment data were bootstrap-resampled ($n = 10,000$) and analyzed using one-way ANOVA, followed by Bonferroni-corrected pairwise comparisons relative to a predefined reference region. For correlation analyses between model-generated flux changes and metabolite or proteomic measurements, Pearson's correlation coefficient ($r$) and associated *p*-values were calculated. For model-generated flux sampling analysis, the Mann–Whitney $U$ test (ranksum test) was used to compare flux distributions between NSD and HSD conditions, as the distributions are not assumed to be normal. Significance levels were indicated in the boxplots. No samples or data points were excluded from statistical analysis. All analyses from experimental data were based on biologically independent replicates. The statistical methods used for each analysis are described in the corresponding figure legends or methods sections.

## Reporting summary

Further information on research design is available in the Nature Portfolio Reporting Summary linked to this article.

# Data availability

All data generated in this study, including $^{13}$C-glucose isotope tracing, high-sugar-diet (HSD) metabolomics, regional metabolomics, and redox proteomics, are provided in the Source Data. Processed and analyzed data are provided in Supplementary Data. The protein mass spectrometry raw data have been deposited to the ProteomeXchange via the MassIVE with the dataset identifier MSV000100288. Source data are provided with this paper.

# Code availability

MATLAB and R scripts used for reconstruction of genome-scale metabolic models and flux analyses are available on GitHub (https://github.com/sunjjmoon/FlyTissueGEMs). An achieved version of the code used in this study is available on Zenodo: https://doi.org/10.5281/zenodo.17684286[115].

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

## Acknowledgements

We thank members of the Perrimon laboratory for useful discussions and experimental advice, including Dr. Stephanie Mohr for feedback on the manuscript. We also thank to the Research Computing Group at Harvard Medical School for access to the O2 High Performance Compute Cluster. We additionally thank Dr. Anush Chiappino-Pepe and Dr. Steven Marygold for valuable feedback on genome-scale metabolic modeling work; Dr. Jason Tennessen for valuable feedback on fly metabolism; Dr. Travis Nemkov for valuable feedback on metabolomics analysis; and Dr. Safak Yilmaz and Dr. Marian Walhout for valuable feedback on genome-scale metabolic models. A.K. was supported by Postdoctoral Fellowship Program (Nurturing Next-generation Researchers) through the National Research Foundation of Korea (NRF) funded by the Ministry of Education (2021R1A6A3A14039622). The mass spectrometry work was partially funded by NIH grants 5P01CA120964 (J.M.A.) and 5P30CA006516 (J.M.A.). This research was supported by NIH/NIDDK 1R01DK136945, NIH NIAMS R01 AR057352, Cancer 509 Research UK (CGCATF-2021/100022) and the National Cancer Institute (1OT2 CA278685-01). N.P. is an investigator of Howard Hughes Medical Institute. This article is subject to HHMI's Open Access to Publications policy. HHMI lab heads have previously granted a nonexclusive CC BY 4.0 license to the public and a sublicensable license to HHMI in their research articles. Pursuant to those licenses, the author-accepted manuscript of this article can be made freely available under a CC BY 4.0 license immediately upon publication.

## Author contributions

S.J.M. and N.P. designed the study. Y.H. performed bioinformatics analyses related to the Fly Cell Atlas datasets. S.J.M. performed the computational analyses and follow-up experiments. S.J.M. and J.M.A. performed metabolomics and subsequent analyses. S.J.M., M.D., and A.D. performed redox proteomics and subsequent analyses. A.R.K. performed the AlphaFold structural analysis. N.P. supervised the work. S.J.M. and N.P. wrote the manuscript. S.J.M., Y.H., M.D., A.R.K., J.M.A., A.D., and N.P. reviewed and edited the manuscript.

## Competing interests

The authors declare no competing interests.
