## [Transparent Peer Review file · Nature Communications]

Modeling tissue-specific *Drosophila* metabolism identifies high sugar diet-induced metabolic dysregulation in muscle at reaction and pathway levels

Corresponding Author: Professor Norbert Perrimon

Version 0:

Reviewer comments:

Reviewer #1

(Remarks to the Author)

Noteworthy results:

- Moon and colleagues describe an approach to apply genome scale metabolic models (GEMs) to individual tissues using transcriptomic data from each tissue. They then integrate the muscle GEM with results of a labelled glucose tracer study to profile reaction rates for glycolysis and the TCA cycle, identifying GAPDH as a rate limiting enzyme. In a separate study, they integrate the muscle GEM with metabolomics data from flies fed either normal or high sugar diets, identifying sorbitol and ketone body metabolism as dysregulated by high sugar diet.

Significance and Novelty:

- The authors make a valid point that it is crucial to understand the contributions of individual organs to systemic metabolism and efforts towards this aim are likely to be welcomed by the community. However, it is not made clear how the results of this approach differ to that which would be produced by traditional gene set enrichment. Clustering of the raw pseudobulk transcriptomic data presented in figure 1b is remarkably similar to the clustering of tissues in their final results in figure 3f. Thus, it is unclear how flux variability analysis adds to our understanding of metabolic activities beyond simply comparing enzyme expression. A similar concern exists for the results of the ¹³C flux analysis. These experiments are interesting, however, it seems that conclusions are largely apparent from looking at the raw fractional enrichment data.
- The conclusion relating to GAPDH is not novel, as GAPDH is a well known rate limiting enzyme (see for example, PMID: 25009227). Conclusions relating to ketone body metabolism are interesting, however, the provided results would benefit greatly from further analysis of the mechanism by which this occurs, as was reported for oxidation of GAPDH.

Does the work support the conclusions and claims, or is additional evidence needed?

- The message of many figures is unclear, especially early in the manuscript. Specifically figures 2b, 3b-e.
- Figure 4 is interesting, however, the significance of this experiment, and how it relates to the conclusions of the paper (GAPDH, ketone metabolism) are unclear. The conclusion that metabolite abundance is reflected by the expression of related mRNA is not overly surprising. The authors do not quantify the accuracy of their predictions of metabolite flux match and so it would be interesting to see how well their predictions correlate with metabolite abundance (across all metabolites, rather than only for specific examples).

Is the methodology sound? Does the work meet the expected standards in your field?

- I have several concerns relating to the prediction of enzyme activity from mRNA abundance. Protein turnover rates are highly protein-specific and likely vary between tissues. For example, the turnover of beta-oxidation proteins is greater in white than brown adipose tissue given increased exposure of brown adipose proteins to reactive oxygen species and oxidative damage. This is not accounted for in the authors estimation of protein abundances, as data from all tissues is imputed from a whole organism reference dataset. This seems at odds with the motivation to increase the tissue-specificity of analyses.
- Enzymatic activities of proteins are not directly linked to their abundances. For example, the activity of glycogen synthase is regulated exquisitely by protein phosphorylation and allostery. Many metabolic enzymes are regulated in this way, given that they are activated by dynamic signalling processes (often phosphorylation) and changes in intracellular substrate levels, however, the approach taken here does not account for this. The authors argue that they confirm "previous reported

differences in the estimated rates of metabolic processes” between tissues, however, as both sets of conclusions appear to be based on very similar approaches, the results are not truly independent.

Reviewer #2

(Remarks to the Author)

The authors analyze tissue-specific metabolism in *Drosophila melanogaster* using metabolic network modeling, metabolomics, and carbon tracing. They first construct metabolic network models for multiple tissues by integrating a Fly metabolic model with tissue transcriptomics, obtained by aggregating data single cell RNA-sequencing. These networks are then converted to quantitative models using estimated variables of enzyme kinetics together with flux balance analysis (FBA) and flux variability analysis (FVA). They then perform an experimental case study of high (HSD) vs normal (NSD) sugar diet in muscle. They show that glyceraldehyde 3-phosphate dehydrogenase (GAPDH) is very likely a rate limiting step of glycolysis in HSD and they show evidence that this is probably associated with an increased NADH/NAD ratio.

Given that the fly is an important model organism for studies of metabolism and metabolic diseases, bringing tissue-level resolution to the metabolic network analysis of this organism is significant and timely. However, I have concerns about the modeling methods and the overall robustness of the models presented. Enzymatic variables are loosely defined and mixed with FBA using an arbitrary objective function. It is not clear if these models are valid or relevant (in part because flux data is not shared in supplementary tables). Furthermore, it is challenging to assert that the modeling results are validated by the HSD case study, as it appears that the experimental carbon tracing (and some other measurements such as NADH/NAD⁺), rather than the modeling, predominantly drives the discovery around GAPDH. Therefore, the contribution of modeling to the findings is not convincing. Combined with a general deficiency in the quality and thoroughness of the methods section the paper as presented is not a robust advance.

Major Comments

1) V_{max} values are used to constrain FBA. V_{max} is a multiplication of k_{cat} and enzyme abundance, both of which are rough approximations. Thus V_{max} values themselves are likely to have poor precision. Indeed, the analysis in the paper shows that good predictions of just the enzyme abundance are only within an order of magnitude of experimental values. Such variance raises doubts about the robustness of a quantitative model, particularly when multiple imprecise estimates are used within the same network. This could lead to the accumulation or amplification of errors, potentially skewing results significantly. Multiple specific examples would be needed to justify the usefulness of these constraints (such as predictions that are true with these constraints but wrong without them).

Also related to this issue, some reported V_{max} values are extremely large (more than a million). I have never seen such high numbers for flux, especially with the unit of mmol/gdW/h, and I was not able to find anything close with a search. To address these concerns, I recommend the authors provide detailed calculations for some of these V_{max} values, starting from the initial data to the final output. Furthermore, can the authors comment on what these high numbers mean for material conversion rates in a whole fly (can be calculated using the mass of a fly or its relevant organ to get an overall rate per hour, second, or day)? Are they realistic?

2) The objective function of FBA adds up ATP, NAD and NADP demand fluxes together. What is the rationale behind this formulation? And how do these components interact within the model? Does the model sometimes pick one over the others and make some of these fluxes zero? Flux tables would be useful to provide answers.

Here, ATP demand is a classical objective function and quantifies energy production capacity. But what the authors call NAD and NADP demand are reactions that convert NAD to NADH (or NADP to NADPH) freely, and therefore, do not quantify electron demand or capacity to extract reducing power from nutrients. Instead, they might inadvertently diminish the importance of these fluxes, analogous to how a reversed ATP reaction ($ADP + P \rightarrow ATP$) would make the electron transport chain redundant. This raises questions about the effectiveness and accuracy of the simulations. Further explanations and justification are needed for this approach.

3) The authors seem to relate high flux variability to metabolite concentrations, and they use the flux variability in a given pathway to validate the enrichment of metabolites to that pathway in metabolomics data. However, it is difficult to link steady state flux values to metabolite abundance in general. Also, variance of flux is an indicator of uncertainty; how would it be related to metabolite abundance?

Particularly concerning is the formulation of pathway flux index (PFI). This formula involves multiplying the number of reactions in a pathway by a sum that itself multiplies the mean flux of each reaction by its flux range. Typically, one would expect a normalization by the number of reactions rather than a straightforward multiplication, as this would prevent the index from disproportionately escalating with the length of the pathway. Moreover, incorporating flux variability ($v_{max} - v_{min}$) into the index suggests that the PFI might reflect the underlying uncertainty of the flux estimates more than the actual pathway activity. Contrary to this, Figure 6a implies that PFI is low for pathways carrying low flux and vice versa; this is not necessarily true based on the mathematical formulation. Even if this were true, we again have the problem with trying to link flux with metabolite abundance. The lack of a clear theoretical basis for the PFI formula suggests potential overfitting of the metabolomics data.

The authors should first show the relationship between flux variability and metabolite enrichment or concentration by doing statistical analysis, such as randomization. If there is indeed a significant relationship, then they could comment on why this would be the case.

4) The manuscript suffers from significant issues with clarity, most notably within the Methods section but also in other parts of the text. These issues affect the overall quality and robustness of the paper.

Minor Comments

1) Ln 124, Figs 1b and S1b: Mentioned clusters are not clear in these figures. S1b does not have a dendrogram to show clusters.

2) Ln 133: "The analysis revealed that germline cells, male accessory glands, and the fat body were able to complete greater number of metabolic tasks than other tissues"
What is the significance of this and how do we know these chosen tasks truly represent the metabolic capabilities of a cell? Isn't this a biased analysis?

3) Fig S1e: Color legend missing.

4) Fig S1f: I believe y-axis is indicating a fraction, not number of transporters?

5) Fig S2c: what is rho for this plot and how does it compare with S2b? It seems from the plot that there is no real improvement in predicting protein abundance using transformed data instead of original mRNA. It is possible that I am not getting it because S2d is not clear (please see below)

6) Fig S2d: what is plotted as a histogram is not clear from legend or text.

7) Fig. S2j: Please explain gene ratio in the legend.

8) Ln 162: "found that nearly 92 % of the maximum rates were within the default maximum values set in the Fruitfly1, with the median Vmax of 23.5 mmol/g-DCW/hr"

It is not significant that predicted values are consistent with the default maxima. The default setting of upper and lower reaction boundaries (nearly all of which are 1000 and -1000, respectively), just indicate an arbitrarily high flux which you don't expect to reach.

9) Ln 178: "certain tissues with higher Vmax were also capable of completing more metabolic tasks than those with lower Vmax."

What is the significance or biological relevance of this?

10) In multiple figures, the font is too small to even read by zooming. The most notable example is Fig. S3.

11) Lns 214-216: "Further, for the protein degradation pathway, muscle cells showed higher flux variability, an observation that agrees with the established function of muscle as a major site of amino acid breakdown."

Why should flux variability indicate activity? Doesn't it just mean we are uncertain about the flux level? Also, the exact range of the flux is important for interpretation. If zero is within the range, then we cannot even conclude if the reaction is predicted to carry flux. I would think reactions that are predicted to be active with high confidence are those with high mean flux and low variability.

12) Ln 226, Figure 3f: It is not clear how data points are grouped together. There are no apparent clusters.

13) Ln 244: "GABA serves as an inhibitor of neurotransmitters"
I think this is not accurate. GABA is an inhibitory neurotransmitter itself.

14) Fig 5i: Not clear how FBA is done and what is shown. There are several problems here: (i) UB and LB of some reactions seem to be the standard value, 1000. So, does this mean v_{max} constraints on muscle are not used here? (ii) I think GAPD in other figures is labeled GAP in this one. (iii) Total flux into GAP is more than the flux out of it. Where does excess GAP go? (iv) What is imposed as a constraint and what is obtained? Please give examples to clarify the method, and provide the data. (v) Flux variability is not shown on the HSD map. The authors can show the numbers as in the NSD pathway figure and still keep the colors to indicate ratios.

15) Ln 360-366: The metabolites discussed in the context of a specific butanoate metabolism pathway, e.g. acetoacetate, are connected to other pathways of the metabolic network, too. So how do we know that the change in proportions are due to enzymatic activities in this particular pathway?

16) Ln 521: "T is the mRNA nucleotides per cell."
How is this obtained and how can it be way less than 1 (1E-15)?

- 17) In methods, equations are not numbered, making it difficult to follow and refer to.
- 18) In equations of Lns 524 and 526, do mRNA and [mRNA] mean the same thing?
- 19) Ln 535: "Mw = 110 and mtot = 150 × 10⁻¹² were used"
Please indicate the unit of mtot. Also I think 1.5E-10 would be the correct form in scientific notation.
- 20) Ln 539: PaxDb appears as abbreviation before this line.
- 21) Ln 533, ATP demand equation: When this equation is used, what happens to the original maintenance requirement reaction?
- 22) Ln 597: "The objective function with cT of 1/3 is defined as"
cT is a vector, so this statement or the equation needs to be modified.
- 23) Equation after Ln 609: What is the rationale for using this constraint?
- 24) Ln 623: is alpha always taken as 0.9 in all analyses or is it varied as the equations suggest?
- 25) Ln 633-639:"The perturbation effect was defined by the summed reactions rates in glycolysis, normalized to the change of maximum reaction rates of perturbed enzyme rate."
This paragraph is not clear and this is the most confusing sentence. Is the quantity that is summed "mean reaction rate"? It is not clear what this is normalized with. Please provide an example or specific equations?

Reviewer #3

(Remarks to the Author)

Reviewer #4

(Remarks to the Author)

Overview:

This manuscript leverages single-nuclei gene expression data to infer tissue-specific genome-scale metabolic models in *Drosophila*. These are constraint-based models that enable prediction of the flux through all metabolic reactions for a given tissue. Additionally, kinetic information (maximum reaction rates, estimated from enzyme abundances and kcat values) is used to further constrain the predicted flux values. The authors compare features of the resulting metabolic networks (network size and structure, ability to perform defined metabolic tasks, max reaction rates, flux values). The models are used to predict the metabolic features in the context of a high sugar diet, and metabolite profiling and heavy-labeled isotope tracing are used to qualitatively test some model predictions.

Overall, the methods are well laid out, appropriate and rigorous. The results presented are clear. However, a primary issue with the paper is regarding significance/impact – it is not clear how the many results presented could be leveraged (by the authors or by others in the field). Specific comments are below.

Major issues:

- A significant weakness is that the paper does not provide a compelling argument that the insights uncovered through metabolic modeling are useful. Thus, it is not clear what the significance and potential impact of this work could be. The discussion mentions cancer cachexia as a future application, but in general, the implications of the metabolic differences between tissues are not presented.
- The title emphasizes the application of studying the effects of high sugar diet; however, most of the paper focuses on comparing the models (p. 6-10; figs 1-4). Only the last part is about high sugar diet (p. 11-13). And related to comment 1 above, the question of how we can use their results remains.
- A central finding for predicting metabolic features with high sugar diet is that GAPDH is a bottleneck reaction (** note, in multiple places, "GADPH" is incorrectly used). This is not a novel result, as GAPDH is widely known to be a limiting reaction in many conditions. This further emphasizes the need to explain why the reader should be excited about the results presented.
- The authors should add a paragraph to the Discussion about limitations and assumptions that affect their findings. For example, using estimates of protein abundances and limited availability of kcat values are expected to influence the results and subsequent conclusions.

Additional issues:

- The “clusters” shown in Figure 1b are not distinguishable. A more quantitative/rigorous definition of the clusters is needed, as many appear to overlap.
- The paper presents that tissues containing enzymes with high V_{max} are able to complete more metabolic tasks (fig. 2b). There appear to be several counter examples to this that should be discussed. Do the points corresponding to low V_{max} represent a low ability to perform tasks? Is this meaningful?
- A finding from the flux variability analysis is there are “similarities and differences across tissues” (line 225). This is a very vague sentence that is not insightful or actionable. Again, it aligns with the major critique that the results are not presented in a way that highlights their utility.
- It would be helpful to the reader to have a table that presents the major model predictions and whether each aligns with experimental evidence (either from this paper or from published works). This would better support the authors’ statements that model predictions are consistent with experimental findings.

Reviewer #5

(Remarks to the Author)

The manuscript by Sun Jin Moon et al. represents a comprehensive study on the use of tissue-specific GEMs to study the effect of a high glucose diet in drosophila. This is quite an impressive study in scope, and will be a good contribution and highly cited study. I think the authors should feel proud of such comprehensive work. However, I think there are still some very important items that need to be addressed regarding the model construction and validation process, along with statistical analyses, which the authors should address, given that others in our community will be using these models, and the models themselves are critical to the claims of the paper. Also there are many critical details that are glossed over, along with community standards in tissue-specific model construction that need to be addressed. However, I hope to see these concerns addressed since I’m excited about this fine work!

Major Comments:

- We applaud the effort to further refine the drosophila reconstruction. These efforts for improving network curation are valuable to the community.
- Numerous algorithms exist for generating tissue specific models, and a number of decisions need to be made in processing the data. Such decisions can have a huge impact on the content that is obtained in tissue-specific models, and thus any interpretations of model analyses. Why was tINIT selected, as opposed to mCADRE, CODA, or other high-quality methods? What validations were done to believe that the models obtained by tINIT were valid? Often, community best-practices validate models by using gene knockouts, correlations to experimental metabolic fluxes (though this is confounded by the models used for fitting C-13 data), cell culture growth rates compared to model growth rates, other experimental validations, or comparison of model functions to comprehensive lists of known metabolic functions (not just citing a few papers that suggest a metabolic function could be active in a couple models, as done here for muscle metabolism). This is critical since other studies (e.g., Opdam, Cell Systems, 2017; Machado, PLoS Comp Bio, 2014) showed some methods were quite poor at making reliable models.
- tINIT, as its name suggests, requires defined metabolic tasks. How were these defined? The methods just said they used a list of 57 tasks in building the models. What were these tasks, how were they defined, and what is the support for each task? (see, e.g., Thiele, Nat Biotech, 2013; Blais Nat Commun, 2017; Richelle, PLoS Comp Bio, 2019). They later mentioned using 256 human metabolism tasks to test model functions. How many of these tasks are relevant to drosophila, as opposed to just human?
- Many tissue-specific model generation methods require one to define which genes are expressed in a given tissue. How was this performed? Was it done by global or local thresholding? (Richelle, PLoS Comp Bio, 2019)
- In figure 1c only 37/256 tasks are shown. It is stated that these were significant. How was significance defined and determined?
- Figure 1d is unclear, and just reporting a Z-score provides limited insight. What % of tasks were functional for the different tissues? Do these conform with known tissue and cell-type specific tasks better than permuting tissue labels and comparing tasks with known tissue-specific functions? On a related topic, it seems that the authors provide some literature support for a couple tissue-specific functions, but can they define tissue-specific functions prior to the analysis so that they can do a proper assessment of true/false positives/negatives to make sure their models are better than randomly-generated models?
- How much does the V_{max} for specific reactions vary across tissues? Do you see, e.g., glycolytic and lactogenic V_{max} values to be higher for more metabolically active cells, like neurons?
- In setting up simulations for FBA and related approaches, one must consider several things, including the objective function. Along these lines, multiple studies have benchmarked these, but I don’t see them referenced. For example, in lines 183-186 the authors explain that all flux analyses in the paper are being conducted by setting the objective with a DM reaction that takes into consideration ATP, NAD and NADPH production. The authors state that this is a common metabolic objective across tissues. However, they are not encountering other metabolic activities that the cell is subjected to and is represented by the biomass function. In this regard the authors should cite previous work that have carried a similar approach and also justify the addition of the NAD^+ to the objective function as they only explained the incorporation of NADPH and ATP to the objective. This could be easily done by referencing the data and rationale in papers such as Savinell and Palsson, J Theor Bio, 1992; Schinn, Metabolic Engineering, 2021; Schuetz, Mol Syst Bio, 2007
- A lot of flux spans are reported from flux variability analysis. These are merely descriptions of the models and parameters that were inferred. I only see one single citation mentioning how the span on follicle and germ cells could be meaningful. Is

there any biological support for all of the differences discussed between lines 194-229?

- Supplementary Fig 1 needs to incorporate labels that indicates the meaning of each color. Specifically panel A and E.
- Supplementary Fig 3 needs to incorporate labels that indicates the meaning of each color. Specifically panel B, H and I.
- In the metabolic profiling section starting with line 231, along with figures 4 and S4, the authors have generated a fantastic dataset and provided a mildly qualitative analysis, but with no statistical tests to see if there is a significant overlap between the metabolomic data and their flux analyses. However, in theory, metabolic task analysis, which they did for figure 1, should allow one to see, for each tissue, which metabolic tasks are more active (i.e., which metabolites could increase in activity). I believe Sanjay Nigam's lab at UCSD did such an analysis that he presented 3 or 4 years ago. Thus, it would provide much greater confidence in their tissue-specific models, if the authors here did a quantitative analysis comparing the metabolic data to the tissue-specific metabolic task activities.
- Line 342, the authors say there's a decrease in PFI. What was the magnitude and statistical significance of the decreases? Bordel, et al. PLoS Comp Bio, 2010 and Bar Evan, PNAS, 2010 from the Nielsen and Milo labs, respectively, presented ways to quantify significance of changes in pathway fluxes using different sampling approaches. We've personally seen in many instances increases and decreases in flux are not significant if using empirical p-values, but the approach is valuable for finding the ones that really are.
- The PFV subtracts the max and min fluxes for the reactions in a pathway, from what I understand. Shouldn't this consider the absolute value, since sometimes reactions are written backwards from the way they're used, and thus would contribute a negative flux value. On a related note, some reactions will be heavily skewed when doing sampling. Did you try using the mode instead for the mean flux for PFI?

Minor Comments:

- Line 136: Figure incorrectly referenced. Correct reference is Supplementary fig. 1e, instead of Supplementary fig. 1d.
- Line 137: Is not clear what the authors are stating. I suggest rephrasing this sentence for better understanding.
- Figure 2a should be incorporated as an equation in the manuscript and being removed from Figure 2.
- As in Figure 2a, Supplementary Figure 2 equations in a and e should be addressed as equations rather than a panel in the figure.
- Figure legend of Supplementary Fig 4 panels h and I are wrongly referenced. I suggest reorganizing the figure.
- Supplementary Fig 5, heatmaps corresponding to panels b and c should be increased in resolution for better readability.
- Line 289: Add "Glyceraldehyde 3-phosphate" initials for better localization in the bar plot of Supplementary Fig.5a.

Reviewer #6

(Remarks to the Author)

Version 1:

Reviewer comments:

Reviewer #1

(Remarks to the Author)

The authors have performed considerable additional experimentation in the revision process. This has addressed many/most of the comments raised. The manuscript is much improved.

Reviewer #2

(Remarks to the Author)

GENERAL COMMENTS

The authors made significant changes in their predictive simulations with the fly tissue GEMs and in the organization of the manuscript. The methodology has been simplified by replacing the previous modeling approach, which relied on kinetic parameters, with regular FBA. Despite this simplification, the predictive power is maintained (or possibly improved), consistent with earlier criticism that the added complexity from inaccurate kinetic parameters was not necessarily beneficial. The methodology is now clearer, aided by better manuscript organization and more straightforward language, in addition to the simplifications in modeling.

However, some methodological concerns remain. A major issue is the treatment of reaction boundaries. As in many GEMs, including the human model used here, reaction flux bounds are uniformly set to +/-1000. This arbitrary value is not biologically meaningful but is meant to represent an effectively unbounded range, under the assumption that biologically

relevant fluxes (e.g., those obtained with reasonable nutrient uptake rates and predicting reasonable biomass production rates) will not approach these limits. The authors' approach, however, treats these arbitrary values as literal and perturbs them in some of their tests (see comments below), which raises technical concerns.

In addition, there are some remaining issues related to quality and clarity. Detailed comments are provided below for another round of revision.

MAJOR COMMENTS

1) The rationale for applying perturbations by altering reaction boundaries is unclear. A 20% reduction in the upper flux bound (or increase in lower bound) is arbitrary and does not affect all reactions equally. For example, if Reaction 1 and Reaction 2 have fluxes of 700 and 850 mmol/gDW/hr, lowering the upper bound from 1000 to 800 will have no impact on Reaction 1, but will constrain Reaction 2, as its reference flux exceeds the new boundary. This inconsistency undermines the interpretation of such perturbations. A more principled approach would be to apply perturbations directly to the flux values themselves; for example, by imposing a constraint that forces a reaction to carry no more than 80% of its unperturbed flux. While I acknowledge that the authors rely on flux sampling and do not define a single reference flux, it would be more appropriate to apply perturbations based on representative statistics of the sampled distribution (e.g., the mode or median), rather than manipulating arbitrary boundary values. This would avoid the conceptual problem of treating +/-1000 as meaningful limits and ensure a more uniform and interpretable perturbation strategy.

2) Related to the issue above, some of the flux sampling distributions shown in Supplementary Figure 49 appear to cluster near the flux boundaries. This raises three concerning possibilities:

(i) The default flux bounds (+/-1000), which are conventionally intended to approximate infinity, may be too restrictive relative to the feasible solution space. In this case, increasing the bounds (e.g., to +/-10000) might be appropriate to avoid artificial truncation of the distributions.

(ii) The nutrient uptake rates may not be well constrained, allowing unrealistically high fluxes. As a result, key fluxes, particularly for the uptake of limiting nutrients, may saturate at the default boundary of 1000, leading to biologically implausible solutions.

(iii) The high-magnitude fluxes may reflect the presence of internal flux loops. For example, one reaction ($A + \text{NADH} \leftrightarrow C + \text{NAD}$) may carry a flux of +1000, while a parallel reaction ($A + \text{NADPH} \leftrightarrow C + \text{NADP}$) carries -990, yielding a net flux of 10 while recycling redox cofactors through an internal loop. This scenario could be evaluated by performing parsimonious FBA (pFBA), which minimizes the total flux and suppresses such loops by favoring net conversions over unnecessary cycling.

More broadly, it would be informative to compare the predictive power of simple pFBA solutions to that of flux samples. Given the potential ambiguity in sampled distributions, especially when they include artificial extremes or loops, pFBA could offer a more interpretable baseline for certain predictions.

3) The rationale for using flux variability range as a multiplier in the Pathway Flux Variability Index (PFVI) is not well justified. The interpretation of this index appears problematic, particularly in cases where pathways are essential and consistently active. In such cases, reactions may exhibit narrow flux ranges that do not include zero (reflecting obligatory activity) but the PFVI will be suppressed due to the low variability. According to the current interpretation, this could misleadingly suggest low pathway activity.

Conversely, as illustrated in Supplementary Figure 50a, a reaction with a broad flux range that spans zero (e.g., the red curve) may be inactive in many feasible solutions, yet it receives a high PFVI score due to its variability. This undermines the biological interpretation of PFVI as a proxy for pathway activity.

This issue should be addressed either by adjusting the PFVI formulation or clarifying its intended meaning. One might also question whether similar qualitative predictions could be made without using variability at all; e.g., by relying solely on representative flux modes (or pFBA fluxes, as discussed in the previous comment). This would avoid conflating variability with activity and may yield more interpretable results.

4) Despite the significant improvements, several figures and supplementary tables still suffer from quality and clarity issues. For example, Supplementary Figure 12 has unreadable column labels; Supplementary Data 2a includes #VALUE! errors in the Excel file; Supplementary Figure 15 and 38, as well as Supplementary Data 2b, also require attention. Please see additional issues detailed in the minor comments below.

MINOR COMMENTS

1) Figure 2f (pathway overlap between GEMs and metabolomics): It appears the authors use KEGG pathways as an intermediary to match GEM activity with metabolomics data. Why is this done indirectly through KEGG? Moreover, the method used to map GEM reactions to KEGG pathways is not clearly explained.

2) Figure 3b: The numbers and parenthetical expressions in the panel are not explained in the legend. Please clarify.

3) Page 9:

"Specifically, glucose uptake rate was reduced to about 55%, while fluxes through glyceraldehyde-3-phosphate dehydrogenase..."

This sounds like a circular argument. If glucose uptake was manually reduced by constraining the upper bound, then observing a reduction in glycolytic flux may simply reflect that constraint. It may still be a valid observation if the flux remains well below the new boundary, but this should be clarified (See Major Comments 1 and 2).

4) Page 9:

“Overall, TCA cycle fluxes were decreased by approximately 50% in the HSD-muscle-GEM, consistent with decreased TCA flux observed in diabetic muscle.”

This does not appear consistent with Figure 3b, which shows an increase in upper TCA cycle flux. Additionally, please explain where excess α -ketoglutarate goes and where the additional malate comes from, as the alternate pathways consuming or producing these intermediates must account for any internal flux imbalance in the cycle.

5) Page 10:

“Comparison of these four reactions revealed approximately 2-fold increase in NADH generating fluxes and a 2-fold decrease in NAD⁺ regenerating fluxes (Fig. 3f and Supplementary Fig. 27) ... confirming the perturbed NADH metabolism.” This reasoning is problematic. By mass balance, any NADH generation must be matched by NAD⁺ regeneration elsewhere in the network. How, then, is this imbalance quantified and connected to the NADH/NAD⁺ ratio? Are the authors referring to net electron flow (e.g., from NADH to other carriers)? Furthermore, focusing on just a couple reactions is difficult to justify when the rest of the network may compensate.

To assess differences in the network's capacity to extract electrons, the authors could add a demand reaction (NADH \rightarrow NAD⁺; in mass and charge balanced form and not in reverse direction as in the original submission) and maximize its flux across conditions.

6) Figure 4c and the like: Normalizing everything with NSD eliminates valuable information. Why not normalize both NSD and HSD with their respective Glc uptake?

7) Supp. Fig. 31: How can downstream metabolites show higher fractional labeling than glucose, the uniformly labeled carbon source used in the experiment?

8) Fig. 6g: Black arrow means Flux is ND. What does it mean for a model prediction? Please clarify.

9) Page 15: “Consistent with this, our flux analysis predicted muscle could catabolize fructose and showed a significant decrease in the activity of Hex-A, a *Drosophila* hexokinase capable of phosphorylating fructose to fructose-6-phosphate⁷⁵ in response to HSD. This aligned with the accumulation of upstream metabolic intermediates, such as sorbitol and trehalose.”

An alternative hypothesis is that metabolite levels (rather than enzyme abundance) drive flux in high-sugar conditions. In that case, low enzyme expression (inferred from mRNA) does not necessarily indicate low flux. This complicates interpretation of perturbed states. Do the authors have data to rule out this alternative?

10) Supp. Data 1c: What is the type and unit of the data presented in this table?

11) Page 18, last paragraph: It is not clear what was achieved with `getINITModel2()`.

12) Pages 25,26: “This formulation was used here as a general proxy for basal cell maintenance, including protein turnover, membrane lipid remodeling, and nucleotide recycling. These processes remain active and physiologically relevant to tissues like muscle^{70,95,96}. Thus, rather than using energy and redox demands as sole objective functions, which could bias the solution space or do not recapitulate other core metabolic activities, we adopted this biomass objective function to capture broader metabolic requirements. We confirmed the flux through this reaction was approximately 1.1 mmol/gDCW/hr, which is orders of magnitude lower than the major glycolytic and TCA cycle fluxes, indicating that the objective function serves as a minimal biosynthetic demand (Supplementary Fig. 25, Supplementary Data 3h).”

I do not understand the reasoning in flux comparisons. Biomass contains pools such as TAG. For example, fatty acid synthesis required for biomass may translate into a very large energetic demand, depending on the biomass composition.

13) Page 28: “For visualization of the predicted structures we utilized ChimeraX (PMID: 37774136).” There appears to be a formatting issue with this reference.

14) Page 30: “To evaluate the statistical significance of pathway-level flux changes between NSD and HSD conditions, we performed a two-sample Z-test using flux distributions obtained from sampling analysis, following an approach described in a previous study¹⁰⁵.”

However, the bibliography does not contain 105 references. Please correct this.

15) Page 30: “where \bar{X} represents the mean pathway flux”

It is unclear how this average is computed. For example, in glycolysis, flux doubles mid-pathway due to carbon splitting (from 6C to 3C intermediates). How is this handled when averaging across a pathway?

Reviewer #3

(Remarks to the Author)

Reviewer #4

(Remarks to the Author)

Overview:

This manuscript leverages single-nuclei gene expression data to infer tissue-specific genome-scale metabolic models in *Drosophila*. These are constraint-based models that enable prediction of the flux through all metabolic reactions for a given tissue. The authors first briefly compare features of the resulting metabolic networks (network size and structure, ability to perform defined metabolic tasks, representation of metabolic subsystems). The muscle-specific model is used to explore the effects in the context of a high sugar diet, and metabolite profiling and heavy-labeled isotope tracing are used to validate some model findings.

Overall, the revised manuscript is much improved. The paper is more streamlined and better demonstrates the utility of tissue-specific GEMs. The methods are well laid out and explained. The results presented are clear, and the title and introduction align with the main message of the paper. Only a few smaller issues remain.

Specific comments:

- The authors have used experimental methods to confirm several model predictions. This is laudable and nicely bridges communities using computational and experimental methods to study metabolism. However, the authors have not sufficiently emphasized what computational modeling using GEMs uniquely offers above and beyond experimental approaches. This should be better highlighted. This is an important point to address.
- More justification is needed for the perturbation-based sensitivity analysis. In the first submission, the upper flux bound was decreased by 10%. Here, 20% is used. Either seems arbitrary and should be justified why that perturbation is meaningful. I could not find any support in the paper cited (ref. 99).
- The number of supplementary figures grew by nearly 10-fold, from 6 to 56! It is the responsibility of the authors to provide a well-curated story – this task has not been met if there are 56 supplementary figures. I believe the authors must reduce this number to only include those that are absolutely needed to help tell the story (rather than adding figures that are “nice to have”). This is a critical point to address.

Evaluating the response to the points raised by Reviewer 5:

In my reading, most of the comments from Reviewer 5 were addressed appropriately. I believe major points related to comparing to experimental data/observations were useful. Additionally, the changes to the manuscript to remove flux distributions and V_{max} calculations made some of the reviewer points no longer relevant.

List of metabolic tasks. This response is still unclear. The authors should mention all essential tasks (57 from human metabolic task list and 219 metabolic functions) in a single place within the methods. As it stands, these two lists are described in different locations in the manuscript. Furthermore, it is not clear whether the 57 is part of the 219. This requires greater clarity from the authors.

Experimental validation. The authors can go further with their validation, as comparing enriched pathways (data) with subsystem coverage (model) does not consider if the reactions present carry flux or not, it just checks if the subsystem is there or not. I believe it is important to perform the same model predicted enrichment after flux values are calculated. It is not clear that the predictions and FEA would agree.

Version 2:

Reviewer comments:

Reviewer #2

(Remarks to the Author)

The authors addressed all of my previous comments and I agree that the manuscript has improved further. Below are two minor issues with the updated sections:

1. NADH \rightarrow NAD conversion reaction: I could not find this in supplementary tables. The authors should make sure this reaction is mass-balanced using a proton, so that the observed effect is not due to proton consumption (but not charge-balanced, since the reaction represents electron demand).
2. Methods, "Quantification of pathway overlap using weighted Jaccard index": This section is rewritten but is still unclear. In particular, the explanation on $wB(i)$ (normalized weight for pathway i) is confusing. For example, in the equation $wB(i) = 1/|B|$, there is no i on the right hand side. What does this mean? Also, why is B in absolute value brackets if it is a number of

enriched pathways, which should always be positive?

Reviewer #4

(Remarks to the Author)

The authors have addressed my previous comments. I particularly appreciate the changes made to address concerns related to the perturbation analysis (which another reviewer also raised).

I do not have any further comments to address.

Reviewer #1 (Remarks to the Author):

Noteworthy results:

- Moon and colleagues describe an approach to apply genome scale metabolic models (GEMs) to individual tissues using transcriptomic data from each tissue. They then integrate the muscle GEM with results of a labelled glucose tracer study to profile reaction rates for glycolysis and the TCA cycle, identifying GAPDH as a rate limiting enzyme. In a separate study, they integrate the muscle GEM with metabolomics data from flies fed either normal or high sugar diets, identifying sorbitol and ketone body metabolism as dysregulated by high sugar diet.

Significance and Novelty:

- “The authors make a valid point that it is crucial to understand the contributions of individual organs to systemic metabolism and efforts towards this aim are likely to be welcomed by the community. However, not made clear how the results of this approach differ to that which would be produced by traditional gene set enrichment. Clustering of the raw pseudobulk transcriptomic data presented in figure 1b is remarkably similar to the clustering of tissues in their final results in figure 3f. Thus, it is unclear how flux variability analysis adds to our understanding of metabolic activities beyond simply comparing enzyme expression.”

We thank the reviewer for the valuable comment. We agree that transcriptomic clustering or gene set enrichment analysis can also reveal metabolic similarities and differences across individual tissues. In the revised manuscript, we included additional analyses by directly comparing clustering derived from either gene expression and tissue-specific GEMs.

These comparisons have been added to the revised manuscript (**Lines 126–133, Fig. 1c, Supplementary Fig. 8–9**). In brief, while we observed strong agreement for certain tissues such as muscle, neurons, and fat body/oenocyte, agreement was lower for glial cells and gut tissues. These results suggest that GEM-based clustering, which incorporates stoichiometric constraints of biochemical reactions, reveals metabolic features not apparent from transcriptomics alone.

Additionally, we agree that the observed similarity between the clustering based on GEMs and the flux variability analysis-based clustering (previous Fig. 3f) may raise questions about the added value of FVA-based analysis. However, this similarity is in fact expected, as the flux variability analyses were performed using the reconstructed tissue-specific GEMs. We acknowledge that we did not sufficiently clarify this point in the original submission. To improve clarity and focus the scope of

the current study, we have removed the tissue-wide FVA-based clustering results from the revised manuscript (original figure 3).

Additionally, we would like to emphasize that unlike traditional gene set enrichment analysis, our tissue-specific GEMs enabled:

- Comparison of metabolic network structures across tissues, including reaction contents, metabolite composition, subsystem coverage, and predicted metabolic task performance (**Fig 1, and supplementary fig1-6, and 10 - 13**)
- Simulation of pseudo steady-state flux distributions (**Fig 3**)
- Identification of rate-controlling steps (e.g., GAPDH) via perturbation-based sensitivity analysis (**Fig3h**)
- Discovery of pathway-level metabolic perturbations using the pathway flux variability index (PFVI) (**Fig 6**)

We recognize that these features were not clearly emphasized in the original manuscript and have revised the text accordingly (**Line 105 – 158**).

“A similar concern exists for the results of the ¹³C flux analysis. These experiments are interesting, however, it seems that conclusions are largely apparent from looking at the raw fractional enrichment data.”

We thank the reviewer for this important point. In the revised manuscript, we clarified that the primary purpose of the ¹³C-glucose tracing experiments was to experimentally validate model-predicted changes in glycolytic fluxes derived from the muscle-GEM (**line 252 – 283**). To directly link model predictions with experimental data, we included a correlation analysis between predicted fluxes and the fractional labeling of corresponding downstream metabolites (**Fig 4e**).

While we agree that fractional enrichment itself can indicate glycolytic activities, our GEM-based analyses provided additional layers of metabolic analysis, including the identification of altered NADH-dependent reactions (**Fig 3c-f,h**) and additional perturbed pathways beyond glycolysis (**Fig 6a-c**).

• “The conclusion relating to GAPDH is not novel, as GAPDH is a well known rate limiting enzyme (see for example, PMID: 25009227).”

We appreciate the reviewer’s reference to prior literature identifying GAPDH as a rate-limiting enzyme. We have cited this study in the revised manuscript (**line**

382-384). While this study has implicated GAPDH in aerobic glycolysis in cancer cell lines, our work expands these findings by demonstrating that GAPDH also serves as a regulatory node in *Drosophila* muscle in response to high sugar diet.

This conclusion was supported by multiple independent analyses, including GEM-based perturbation sensitivity analysis, and *in vivo* ¹³C-glucose tracing (**Fig3h, Fig4**). Importantly, we showed that the decreased GAPDH activity was not associated with enzyme abundance, but rather by redox modification — specifically methionine oxidation near the NAD⁺ binding site (**line 390-393, Fig5g**). These findings, to our knowledge, has not been previously described *in vivo* in muscle in the context of high sugar diet feeding. We have clarified this point in the revised discussion (**lines 380–393**).

“Conclusions relating to ketone body metabolism are interesting, however, the provided results would benefit greatly from further analysis of the mechanism by which this occurs, as was reported for oxidation of GAPDH.”

We also thank the reviewer for highlighting the findings on ketone body metabolism (butanoate metabolism), which was identified as upregulated via our pathway flux variability index analysis (**Figure 6a–b**). We acknowledge that the mechanistic exploration of this pathway was limited in our current study. To maintain a focused narrative, we did not further pursue ketone body metabolism in the revised manuscript and instead prioritized fructose metabolism, which exhibited the most pronounced downregulation (**line 346– 358**). We have included an additional figure (**Fig. 6g**) integrating results from flux analysis, metabolomics, and redox proteomics to support this focus. We agree that ketone body metabolism remains an interesting area for future investigation.

Does the work support the conclusions and claims, or is additional evidence needed?

• “The message of many figures is unclear, especially early in the manuscript. Specifically figures 2b, 3b-e.”

We thank the reviewer for this helpful feedback. In the revised manuscript, we removed the original Figures 2 and 3, which focused on tissue-wide metabolic comparisons based on Vmax and flux variability. These analyses were deprioritized to better focus the manuscript on our core narrative — comparison of metabolic network structures, subsystem, and task analyses, and flux analyses in muscle under high sugar diet condition. As such, the concerns

regarding unclear messaging in these figures have been addressed through their removal.

• Figure 4 is interesting, however, the significance of this experiment, and how it relates to the conclusions of the paper (GAPDH, ketone metabolism) are unclear. The conclusion that metabolite abundance is reflected by the expression of related mRNA is not overly surprising. The authors do not quantify the accuracy of their predictions of metabolite flux match and so it would be interesting to see how well their predictions correlate with metabolite abundance (across all metabolites, rather than only for specific examples).

We thank the reviewer for the valuable comment regarding the original figure 4. In the initial version of the manuscript, our goal was to highlight regionally enriched metabolites and use them to validate known tissue functions derived from tissue-specific GEMs. However, we agree that the manuscript lacked a quantitative assessment of model predictions against experimental metabolomics data.

In the revised version (now **Figure 2**), we addressed this by: (1) performing pathway enrichment analysis on region-specific metabolomics data, and (2) comparing the enriched pathways to those predicted by tissue-specific GEMs. To quantify the agreement, we computed weighted Jaccard indices, which represent pathway overlap similarities (see **Methods, line 625 - 651**). This quantification systematically evaluated the performance of tissue-specific GEMs in identifying enriched pathways for each tissue, with muscle and fat-body showing strong agreement (**Figure 2g–h, Line 193 - 208**). Additionally, we relocated metabolites comparisons found in original figure 4c-f to the supplementary figure 16 – 21.

Additionally, while these additional analyses supported the utility of GEMs for identifying tissue-specific metabolic pathways, we also acknowledge the limitations. As noted in the revised Discussion (**Lines 425–434**), not all GEMs could be fully validated using metabolomics alone, and additional data such as ¹³C-based flux analysis would be important to strengthen model confidence across all 32 tissues.

Furthermore, in response to the reviewer's suggestion to assess prediction accuracy at individual metabolite level, we conducted correlation analyses between simulated metabolite demand fluxes and measured metabolite abundances, following the methodology of Lewis et, Cell Systems, 2021. However, we observed weak correlation ($R < 0.14$; $p > 0.05$; data not shown), likely due to the fundamental difference between production flux (simulated) and steady-state abundance (measured), which is a function of both production and

consumption rates. As such, we concluded that pathway-level comparisons offer a more robust and informative strategy for model validation in this context.

In summary, the metabolomics analysis originally shown in Figure 4 was reanalyzed to validate model-predicted pathway enrichments. We have clarified this purpose in a dedicated results section in the revised manuscript (**Lines 160–208**).

Is the methodology sound? Does the work meet the expected standards in your field?

• I have several concerns relating to the prediction of enzyme activity from mRNA abundance. Protein turnover rates are highly protein-specific and likely vary between tissues. For example, the turnover of beta-oxidation proteins is greater in white than brown adipose tissue given increased exposure of brown adipose proteins to reactive oxygen species and oxidative damage. This is not accounted for in the authors estimation of protein abundances, as data from all tissues is imputed from a whole organism reference dataset. This seems at odds with the motivation to increase the tissue-specificity of analyses.

We appreciate the reviewer's comment and agree that estimating enzyme activity from tissue-specific mRNA abundance with whole-organism kinetic reference datasets has important limitations (Originally presented in figure 2 and 3). While we acknowledged these assumptions in the original manuscript, we have now removed all enzyme activity-based analyses from the revised version.

Instead, we focus on comparing tissue-specific metabolic network structures and applying GEM-based simulations to investigate muscle metabolism in response to high sugar diet feeding (now **Fig3 - 6**). These analyses are supported by multiple layers of experimental validation, including ¹³C-glucose tracing, metabolomics, and pathway enrichment analyses.

• Enzymatic activities of proteins are not directly linked to their abundances. For example, the activity of glycogen synthase is regulated exquisitely by protein phosphorylation and allostery. Many metabolic enzymes are regulated in this way, given that they are activated by dynamic signalling processes (often phosphorylation) and changes in intracellular substrate levels, however, the approach taken here does not account for this. The authors argue that they confirm "previous reported differences in the estimated rates of metabolic processes" between tissues, however, as both sets of conclusions appear to be based on very similar approaches, the results are not truly independent.

We agree with the reviewer that enzyme activity is not solely determined by protein abundance but is also regulated through mechanisms such as phosphorylation, allostery, and substrate availability. These regulatory layers are not explicitly captured by the current constraint-based GEM framework. However, by simulating a range of feasible fluxes (e.g., flux variability analysis), GEMs can still reflect the potential regulatory flexibility within metabolic networks. Nonetheless, identifying the exact regulatory mechanisms that control these fluxes—such as post-translational modifications—remains an important future direction.

As the main goal of this study is not to evaluate these regulatory effects (e.g., enzyme abundance on flux activities), we have removed enzyme-based analyses (original Fig 2 and 3) from the revised manuscript and no longer claim that our results "confirm previously reported differences in metabolic rates." In fact, because enzyme activity was inferred using transcript levels combined with global kinetic parameters, the two approaches were not fully independent, as the reviewer indicated.

In regard to regulatory effects, we further reanalyzed the redox proteomics data independently, and demonstrated that glycolytic fluxes were not associated with their enzyme levels, but rather strongly associated with redox modifications of those enzymes, including GAPDH (**Fig 5f and line 286 - 325**). Accordingly, we thank reviewer of this point and highlighted the importance of redox modification rather than enzyme abundance in the revised discussion (**line 391 – 393**).

Reviewer #2 (Remarks to the Author):

The authors analyze tissue-specific metabolism in *Drosophila melanogaster* using metabolic network modeling, metabolomics, and carbon tracing. They first construct metabolic network models for multiple tissues by integrating a Fly metabolic model with tissue transcriptomics, obtained by aggregating data single cell RNA-sequencing. These networks are then converted to quantitative models using estimated variables of enzyme kinetics together with flux balance analysis (FBA) and flux variability analysis (FVA). They then perform an experimental case study of high (HSD) vs normal (NSD) sugar diet in muscle. They show that glyceraldehyde 3-phosphate dehydrogenase (GAPDH) is very likely a rate limiting step of glycolysis in HSD and they show evidence that this is probably associated with an increased NADH/NAD ratio.

Given that the fly is an important model organism for studies of metabolism and metabolic diseases, bringing tissue-level resolution to the metabolic network analysis of this organism is significant and timely. However, I have concerns about the modeling methods and the overall robustness of the models presented. Enzymatic variables are loosely defined and mixed with FBA using an arbitrary objective function. It is not clear if these models are valid or relevant (in part because flux data is not shared in supplementary tables). Furthermore, it is challenging to assert that the modeling results are validated by the HSD case study, as it appears that the experimental carbon tracing (and some other measurements such as NADH/NAD⁺), rather than the modeling, predominantly drives the discovery around GAPDH. Therefore, the contribution of modeling to the findings is not convincing. Combined with a general deficiency in the quality and thoroughness of the methods section the paper as presented is not a robust advance.

Major Comments

1) V_{max} values are used to constrain FBA. V_{max} is a multiplication of k_{cat} and enzyme abundance, both of which are rough approximations. Thus V_{max} values themselves are likely to have poor precision. Indeed, the analysis in the paper shows that good predictions of just the enzyme abundance are only within an order of magnitude of experimental values. Such variance raises doubts about the robustness of a quantitative model, particularly when multiple imprecise estimates are used within the same network. This could lead to the accumulation or amplification of errors, potentially skewing results significantly. Multiple specific examples would be needed to justify the usefulness of these constraints (such as predictions that are true with these constraints but wrong without them).

Also related to this issue, some reported V_{max} values are extremely large (more than a million). I have never seen such high numbers for flux, especially with the unit of mmol/gdW/h, and I was not able to find anything close with a search. To address these concerns, I recommend the authors provide detailed calculations for some of these V_{max} values, starting from the initial data to the final output. Furthermore, can the authors comment on what these high numbers mean for material conversion rates in a whole fly (can be calculated using the mass of a fly or its relevant organ to get an overall rate per hour, second, or day)? Are they realistic?

We thank the reviewer for raising important concerns about the use of V_{max} constraints derived from estimated enzyme abundances and k_{cat} values. We agree that V_{max} values are approximations and that both k_{cat} and enzyme abundance estimates carry inherent uncertainty. While we incorporated V_{max} analyses to advance GEM-based analyses, we recognized the lack of robust validation methods to confirm tissue-specific *in vivo* k_{cat} values and steady-state protein levels.

In response, we have removed all V_{max} associated constraint modeling and related analyses from the revised manuscript, including those previously presented in original **figures 2 and 3**. In the revised version, we instead focus on the application of constraint-based flux simulations (**Figure 4 – 6**). All conclusions and predictions are now derived from GEM simulations using established methods such as flux balance analysis (FBA), flux variability analysis (FVA), and sensitivity analysis, without reliance on V_{max} associated analyses.

2) The objective function of FBA adds up ATP, NAD and NADP demand fluxes together. What is the rationale behind this formulation? And how do these components interact within the model? Does the model sometimes pick one over the others and make some of these fluxes zero? Flux tables would be useful to provide answers. Here, ATP demand is a classical objective function and quantifies energy production capacity. But what the authors call NAD and NADP demand are reactions that convert NAD to NADH (or NADP to NADPH) freely, and therefore, do not quantify electron demand or capacity to extract reducing power from nutrients. Instead, they might inadvertently diminish the importance of these fluxes, analogous to how a reversed ATP reaction ($ADP + P \rightarrow ATP$) would make the electron transport chain redundant. This raises questions about the effectiveness and accuracy of the simulations. Further explanations and justification are needed for this approach.

We thank the reviewer for this important comment. In the original version of the manuscript, we intended to use a composite objective function combining ATP,

NAD⁺, and NADPH demands to reflect the major energetic and redox needs of metabolically active tissues like muscle. This was conceptually motivated by experimental evidence showing reductions in ATP levels, NAD⁺/NADH, and NADPH/NADP⁺ ratios in muscle under high sugar diet conditions, as well as prior modeling studies incorporating redox demand objectives (e.g., Lewis et al., *Cell Systems* 2018) (**Line 664 – 669**).

However, upon further reflection and in response to reviewer feedback, we recognized that using redox-related demand reactions as standalone or combined objectives could introduce artificial fluxes and potentially bias interpretation of central carbon metabolism, particularly if not well-justified across all tissues.

In the revised manuscript, we have not used the ATP/NAD/NADPH composite objective function and instead adopted a biomass objective function (reaction ID: *MAR00021*) for all simulations (**Line 696 - 706**). We used this formulation as a general proxy for basal cellular maintenance, including protein turnover, membrane lipid remodeling, and nucleotide recycling, which remain physiologically active in adult muscle tissue.

This adjustment resolves the reviewer's concern that maximizing NAD⁺ and NADPH demand reactions as part of the objective function might act as artificial sinks — freely converting cofactors without accounting for upstream metabolic requirements. By removing these from the objective, we avoided unrealistic cofactor cycling that could otherwise bypass physiologically important reactions, such as the electron transport chain. We clarified this explicitly in the revised Methods section (**Lines 707–714**) and ensured that all modeling results presented in the revised manuscript (e.g., **Figures 3–6**) are based on the biomass objective function. As such, our conclusions regarding glycolytic and TCA cycle flux changes under HSD conditions are not dependent on the earlier ATP/NAD/NADPH objective formulation. We also ensured that the flux through biomass was maintained and confirmed at least two orders of magnitude lower than glycolysis (**Line 711-714**). Additionally, we acknowledge the limitation of using this objective function in discussion (**Line 429 - 432**)

Importantly, we ensured that the following GEM-based predictions were validated with orthogonal experimental measurements, such as ¹³C-glucose tracing and metabolomics-based pathway enrichment analysis. To clarify that these experiments served as independent validation—rather than the primary drivers of our conclusions—we now present them in two dedicated validation sections: **Lines 160–208** for pathway enrichment analysis and **Lines 251–283** for ¹³C-glucose tracing.

3) The authors seem to relate high flux variability to metabolite concentrations, and they use the flux variability in a given pathway to validate the enrichment of metabolites to that pathway in metabolomics data. However, it is difficult to link steady state flux values to metabolite abundance in general. Also, variance of flux is an indicator of uncertainty; how would it be related to metabolite abundance?

We thank the reviewer for this thoughtful comment and appreciate the opportunity to clarify our approach. We agree that directly correlating steady-state fluxes or flux variability with metabolite abundance is challenging.

In our original submission, we may not have clearly conveyed the intent of our analysis, which led to confusion. We did not aim to correlate flux variability directly with individual metabolite concentrations. Instead, in the revised manuscript (**Figure 2, Lines 160–208**), we clarified that our comparison was performed at the pathway level, evaluating whether enriched pathways predicted from tissue-specific GEMs aligned with those enriched based on region-specific metabolomics data.

Importantly, we used flux variability values (i.e., the difference between V_{max} and V_{min} obtained from FVA) as a measure of feasible flux range, not as a proxy for statistical uncertainty. We clarified this in the method section (**Line 724 - 728**).

To prevent further confusion, we have reorganized and clarified these analyses in the revised Results (particularly **figure 2**) and Methods sections regarding GEM and flux analyses.

“Particularly concerning is the formulation of pathway flux index (PFI). This formula involves multiplying the number of reactions in a pathway by a sum that itself multiplies the mean flux of each reaction by its flux range. Typically, one would expect a normalization by the number of reactions rather than a straightforward multiplication, as this would prevent the index from disproportionately escalating with the length of the pathway. Moreover, incorporating flux variability ($v_{max}-v_{min}$) into the index suggests that the PFI might reflect the underlying uncertainty of the flux estimates more than the actual pathway activity. Contrary to this, Figure 6a implies that PFI is low for pathways carrying low flux and vice versa; this is not necessarily true based on the mathematical formulation. Even if this were true, we again have the problem with trying to link flux with metabolite abundance. The lack of a clear theoretical basis for the PFI formula suggests potential overfitting of the metabolomics data.”

We thank the reviewer for the thoughtful and detailed critique regarding the formulation and interpretation of the Pathway Flux Index, now revised and

referred to as the Pathway Flux Variability Index (PFVI). We acknowledge that our original explanation may have led to confusion about its purpose and theoretical basis. The revised formulation and rationale are now fully described in the Methods section (**Line 816 – 826**).

In response to the reviewer's concern, we now normalize by the number of reactions in a pathway by calculating PFVI as a weighted average, rather than a sum, which prevents the metric from disproportionately scaling with pathway length. Additionally, we incorporate the mode of flux distributions from sampling to better reflect the central tendency of pathway activity, rather than relying solely on extremes. These adjustments allow PFVI to capture both the magnitude and variability of fluxes within a pathway, while mitigating the risk of overinterpreting uncertainty as activity.

To further support the interpretation of pathway-level analysis, we performed an additional Z-test-based statistical analysis comparing mean pathway fluxes between HSD and NSD conditions using flux sampling distributions (**Line 828-837**). This provided a statistical assessment of pathway perturbations. Together, these improvements strengthen the conceptual and statistical basis of our pathway-level flux comparisons using PFVI.

Finally, our aim was not to directly correlate PFVI with individual metabolite abundances. Rather, PFVI served as a model-derived index to identify pathways with altered metabolic activity under high sugar diet (HSD) conditions. We then independently assessed whether significantly altered metabolites identified through metabolomics were associated with PFVI-predicted pathways. This approach allowed us to validate the biological relevance of PFVI results using orthogonal experimental data, as described in the revised Results (**Figure 6f, Lines 327–358**). Notably, we found that three of the six most significantly altered metabolites (from over 200 detected) were indeed associated with the top pathways identified by PFVI, supporting the utility of this index in revealing statistically meaningful metabolic changes (**Line 346 – 358**).

The authors should first show the relationship between flux variability and metabolite enrichment or concentration by doing statistical analysis, such as randomization. If there is indeed a significant relationship, then they could comment on why this would be the case.

We thank the reviewer for this thoughtful suggestion. In our revised manuscript, we have refocused our validation strategy to avoid direct comparisons involving

flux variability or individual metabolite concentrations, as we found that fluxes and metabolite levels do not show high correlation to one another.

Instead, we now assess model validity at the pathway level by comparing GEM-predicted pathway coverage (based on subsystem representation in tissue-specific GEMs) with experimentally enriched pathways derived from region-specific targeted metabolomics. To quantify the similarity between these two independently derived pathway sets, we computed weighted Jaccard indices (**Fig. 2f–h**), which measure the degree of overlap while considering pathway size. This provides a robust, statistical measure of agreement between model predictions and experimental data.

To further strengthen this analysis, and in response to the reviewer's suggestion for statistical evaluation, we performed a randomization-based test to assess whether the observed overlap was greater than expected by chance. For both thorax and abdomen, we observed significantly higher overlap between metabolomics- and GEM-derived pathways compared to randomized null distributions (empirical $p < 0.01$), demonstrating that these overlaps are non-random and biologically meaningful.

We have clarified this pathway-level comparison in the revised Results section (**Lines 160–208**) and provided full details of the randomization procedure in the Methods section (**Lines 644–651**). This revised approach avoids potential confusion related to flux variability and supports the utility of tissue-specific GEMs in capturing tissue-relevant metabolic features consistent with metabolomics data.

4) The manuscript suffers from significant issues with clarity, most notably within the Methods section but also in other parts of the text. These issues affect the overall quality and robustness of the paper.

We sincerely thank the reviewer for this important feedback. In response, we have thoroughly revised the manuscript to improve clarity, structure, and overall readability, especially in the Methods section.

Specifically:

- We reorganized the Methods into clearly defined subsections, providing detailed descriptions of model reconstruction. To improve navigation and consistency, we aligned the order of Methods subsections with the sequence of figures in the Results section.

- We included additional statistical analyses that were missing or underdeveloped in the original submission, such as Z-tests for pathway-level flux significance and permutation-based validation for pathway overlap comparisons. These are now clearly described in the revised Methods and applied consistently across key analyses.
- To support transparency and reproducibility, we included raw data, statistical outputs, and source tables used in the figures and key analyses as Supplementary Data Files, including flux sampling distributions, metabolite intensities, and proteomics results.

Importantly, while we made these structural and clarifying revisions, the major findings and conclusions of the study remain unchanged. These revisions have helped us streamline the narrative and better focus the manuscript without altering the main results or biological interpretations presented in the original version. We hope these updates address the reviewer's concerns and improve the manuscript's robustness and clarity.

Minor Comments

1) Ln 124, Figs 1b and S1b: Mentioned clusters are not clear in these figures. S1b does not have a dendrogram to show clusters.

Thank you for pointing this out. We agree that the clustering information was not clearly shown in the original figures. To address this, we have now included the hierarchical clustering in the revised **Supplementary Fig. 7** and provided the corresponding data in **Supplementary Data 1f**.

2) Ln 133: "The analysis revealed that germline cells, male accessory glands, and the fat body were able to complete greater number of metabolic tasks than other tissues" What is the significance of this and how do we know these chosen tasks truly represent the metabolic capabilities of a cell? Isn't this a biased analysis?

Thank you for the insightful comment. The goal of this analysis was to compare tissues based on their ability to perform a common set of predefined metabolic tasks. We agree that these tasks may not fully capture the true metabolic capabilities of each tissue and may include false positives. Furthermore, task outcomes are highly context-dependent, so we intentionally placed less emphasis on task pass/fail counts alone.

To address this, we further categorized the 219 tasks into seven metabolic systems (as in Richelle et al., 2019) and performed Fisher's exact test to identify significant associations between tissue types and metabolic systems (**Line 554 – 555**). This allowed for a more systematic and statistical comparison across tissues. We also validated selected tasks (e.g., trehalose synthesis in the fat body) based on known tissue functions. These results are now detailed in **Supplementary Figs. 11–13** and **Supplementary Data 1k–n**). We agree that a deeper evaluation of task performance, including experimental validation and expanded task sets, would be valuable and is better suited for a dedicated future study.

3) Fig S1e: Color legend missing.

Thank you for the comment. To improve clarity and streamline the presentation, this figure has been removed from the revised version.

4) Fig S1f: I believe y-axis is indicating a fraction, not number of transporters?

Thank you for pointing this out. Indeed, the y-axis represents the fraction of extracellular transport reactions within the transport subsystem, not the absolute number of transporters. We have clarified this in the revised figure legend and main text (now described in **Fig. 1e** and Supplementary Data 1h) to avoid confusion.

5) Fig S2c: what is rho for this plot and how does it compare with S2b? It seems from the plot that there is no real improvement in predicting protein abundance using transformed data instead of original mRNA. It is possible that I am not getting it because S2d is not clear (please see below)

6) Fig S2d: what is plotted as a histogram is not clear from legend or text.

7) Fig. S2j: Please explain gene ratio in the legend.

Thank you for these helpful comments. These figures were part of our attempt to validate model outputs using available whole-organism proteomics data. However, we agree that the predictive power was limited and difficult to interpret, partly due to the mismatch between tissue-specific models and organism-level protein abundance. Given these limitations and for clarity, we have removed Figure 2 and the related Supplementary Figures from the revised manuscript to focus on more robust and interpretable results.

8) Ln 162: "found that nearly 92 % of the maximum rates were within the default

maximum values set in the Fruitfly1, with the median V_{max} of 23.5 mmol/g-DCW/hr" It is not significant that predicted values are consistent with the default maxima. The default setting of upper and lower reaction boundaries (nearly all of which are 1000 and -1000, respectively), just indicate an arbitrarily high flux which you don't expect to reach.

9) Ln 178: "certain tissues with higher V_{max} were also capable of completing more metabolic tasks than those with lower V_{max} ."

What is the significance or biological relevance of this?

10) In multiple figures, the font is too small to even read by zooming. The most notable example is Fig. S3.

11) Lns 214-216: "Further, for the protein degradation pathway, muscle cells showed higher flux variability, an observation that agrees with the established function of muscle as a major site of amino acid breakdown."

Why should flux variability indicate activity? Doesn't it just mean we are uncertain about the flux level? Also, the exact range of the flux is important for interpretation. If zero is within the range, then we cannot even conclude if the reaction is predicted to carry flux. I would think reactions that are predicted to be active with high confidence are those with high mean flux and low variability.

12) Ln 226, Figure 3f: It is not clear how data points are grouped together. There are no apparent clusters.

Thank you for these thoughtful and consistent comments. We agree that the biological interpretation and validation of the V_{max} -based and flux variability analyses were limited in our initial submission. We found it challenging to validate conclusions from these results. Therefore, we have removed these sections—including related figures (Figs. 2, 3f, and S3)—from the revised manuscript to focus on experimentally supported findings and interpretable model-based analyses, such as enriched pathways, flux analysis, and pathway flux variability analysis in muscle in response to high sugar diet.

13) Ln 244: "GABA serves as an inhibitor of neurotransmitters"

I think this is not accurate. GABA is an inhibitory neurotransmitter itself.

Thank you for catching this. Yes, GABA is itself an inhibitory neurotransmitter. We have corrected the sentence in the revised manuscript.

14) Fig 5i: Not clear how FBA is done and what is shown. There are several problems here: (i) UB and LB of some reactions seem to be the standard value, 1000. So, does this mean v_{max} constraints on muscle are not used here? (ii) I think GAPD in other figures is labeled GAP in this one. (iii) Total flux into GAP is more than the flux out of it. Where does excess GAP go? (iv) What is imposed as a constraint and what is

obtained? Please give examples to clarify the method, and provide the data. (v) Flux variability is not shown on the HSD map. The authors can show the numbers as in the NSD pathway figure and still keep the colors to indicate ratios.

Thank you for these detailed and helpful comments. We have clarified the methodology and updated the relevant figures in the revised manuscript. To improve clarity, we split the original Figure 5 into three focused figures: **Figure 3** (flux analysis), **Figure 4** (¹³C glucose tracing validation), and **Figure 5** (redox proteomics). We also added a dedicated section in the **Methods** titled “*Definition of HSD-muscle-GEM*” (**Lines 654–675**), which outlines the constraints imposed on the model to simulate high sugar diet (HSD) conditions. Further details on the flux balance analysis, including the objective function, are now provided in **Lines 677–714**. Additionally, all relevant flux constraints and simulation data are included in **Supplementary Data 3**.

Below we address each point specifically:

- (i) Vmax-based constraints from the earlier version (original Figs. 2 and 3) are no longer used. We imposed constraints based on literature-reported physiological changes under high sugar diet conditions, which are now detailed in **Supplementary Data 3a**.
- (ii) We have now standardized all labels across figures and use "GAP" consistently for clarity.
- (iii) Thank you for pointing this out. While the simplified pathway map may appear imbalanced, the full flux distribution confirms that net production and consumption of GAP is indeed balanced. This information is provided in **Supplementary Data 3i**, which includes all fluxes involving GAP. Specifically, our simulation showed GAP is further metabolized via the TALDO-mediated reaction into E4P and F6P as part of the pentose phosphate pathway. This branch was not shown in the main figure to maintain focus on glycolysis and the TCA cycle, but has now been clarified in supplementary materials.
- (iv) We apologize for the earlier lack of clarity. The specific constraints imposed are now listed in Supplementary Data 3a and 3b, the resulting flux distributions for glycolysis and the TCA cycle are in Data 3c, NADH-dependent reactions are in **Data 3d**, and the flux sampling results are provided in Data 3e.
- (v) We appreciate the suggestion regarding flux variability. To reduce visual complexity in the main figure, we did not include flux ranges in the pathway map. However, the full variability information is now included in **Supplementary Data. 3c**.

15) Ln 360-366: The metabolites discussed in the context of a specific butanoate metabolism pathway, e.g. acetoacetate, are connected to other pathways of the metabolic network, too. So how do we know that the change in proportions are due to enzymatic activities in this particular pathway?

Thank you for this important point. We agree that metabolites like acetoacetate can participate in multiple pathways, and interpreting their changes solely within the context of butanoate metabolism can be challenging. While we did not perform an in-depth analysis of butanoate metabolism in the revised manuscript, we noted a substantial flux increase through the reaction MAR04461 (CG8888), which interconverts (R)-3-hydroxybutanoate and acetoacetate with NAD⁺/NADH (original figure 6d). However, to maintain focus and avoid overinterpretation, we chose not to further pursue mechanistic validation of this pathway in the current study.

16) Ln 521: "T is the mRNA nucleotides per cell."

How is this obtained and how can it be way less than 1 (1E-15)?

17) In methods, equations are not numbered, making it difficult to follow and refer to.

18) In equations of Lns 524 and 526, do mRNA and [mRNA] mean the same thing?

19) Ln 535: "Mw = 110 and mtot = 150 × 10⁻¹² were used"

Please indicate the unit of mtot. Also I think 1.5E-10 would be the correct form in scientific notation.

20) Ln 539: PaxDb appears as abbreviation before this line.

Thank you for these detailed and thoughtful comments. We have removed this result sections from the revised manuscript to improve clarity and focus on experimentally supported findings. We agree that several aspects of this section required further clarification and validation, and we will revisit this direction in future work with improved methodology and validation.

21) Ln 533, ATP demand equation: When this equation is used, what happens to the original maintenance requirement reaction?

22) Ln 597: "The objective function with cT of 1/3 is defined as" cT is a vector, so this statement or the equation needs to be modified.

Thank you for these comments. We agree that these formulations required clarification. In the revised manuscript, we no longer used the ATP, NAD⁺, or NADPH demand equations as objective functions. Instead, these were retained as auxiliary buffer reactions to reflect physiological energy and redox demands. For FBA simulations, we used a biomass objective function (reaction ID: MAR00021) that captures broad biosynthetic and maintenance needs of muscle tissue. We have clarified this in the revised *Definition of HSD-muscle-GEM* (Line 653 – 675) and *Flux Balance Analysis* (Line 678 – 715) method sections.

23) Equation after Ln 609: What is the rationale for using this constraint?

Thank you for the comment. This constraint were removed in the revised manuscript. Instead of using it as a modeling constraint, we now use ¹³C-labeling results as an independent validation of model predictions. We have removed this constraint methods in the revised manuscript.

24) Ln 623: is alpha always taken as 0.9 in all analyses or is it varied as the equations suggest?

Thank you for the question. We used $\alpha = 0.9$ consistently in all Flux Variability Analysis (FVA) simulations. This value was chosen to allow for near-optimal solutions while enabling exploration of alternative biologically plausible flux distributions. We have clarified this in the revised *Flux Variability Analysis* (line 717 – 729) section of the Methods.

25) Ln 633-639:"The perturbation effect was defined by the summed reactions rates in glycolysis, normalized to the change of maximum reaction rates of perturbed enzyme rate."

This paragraph is not clear and this is the most confusing sentence. Is the quantity that is summed "mean reaction rate"? It is not clear what this is normalized with. Please provide an example or specific equations?

Thank you for pointing this out. We agree that the original description was unclear. In the revised manuscript, we have described in the *Methods* under "Perturbation-based sensitivity analysis (Line 737 – 752)", where we now define a normalized sensitivity coefficient with a full equation and clear explanation of each term.

Reviewer #3 (Remarks to the Author):

Reviewer #4 (Remarks to the Author):

Overview:

This manuscript leverages single-nuclei gene expression data to infer tissue-specific genome-scale metabolic models in *Drosophila*. These are constraint-based models that enable prediction of the flux through all metabolic reactions for a given tissue. Additionally, kinetic information (maximum reaction rates, estimated from enzyme abundances and *k_{cat}* values) is used to further constrain the predicted flux values. The authors compare features of the resulting metabolic networks (network size and structure, ability to perform defined metabolic tasks, max reaction rates, flux values). The models are used to predict the metabolic features in the context of a high sugar diet, and metabolite profiling and heavy-labeled isotope tracing are used to qualitatively test some model predictions.

Overall, the methods are well laid out, appropriate and rigorous. The results presented are clear. However, a primary issue with the paper is regarding significance/impact – it is not clear how the many results presented could be leveraged (by the authors or by others in the field). Specific comments are below.

Major issues:

“- A significant weakness is that the paper does not provide a compelling argument that the insights uncovered through metabolic modeling are useful. Thus, it is not clear what the significance and potential impact of this work could be. The discussion mentions cancer cachexia as a future application, but in general, the implications of the metabolic differences between tissues are not presented.”

We thank the reviewer for this important point. In the revised manuscript, we have substantially restructured and elaborated both the *Results* and *Discussion* sections to better highlight the significance and broader impact of our tissue-specific genome-scale metabolic models (GEMs). While the revised manuscript provides detailed explanation, we briefly re-emphasize several key contributions enabled by the reorganized Results:

1. Advance in tissue-level metabolism studies in *Drosophila* (Fig 1 -2):

While generic metabolic models have been developed for *Drosophila* (e.g., Fruitfly1, iDrosophila), our study represents the first systematic reconstruction of *tissue-specific* GEMs for this organism, providing a valuable framework for studying tissue-specific metabolism in this organism.

2. Application of tissue-specific GEM in studying muscle metabolism in response to high sugar diet feeding (**Fig 3 – 5**):

As a proof-of-concept application, we demonstrate the utility of muscle-GEM to not only recapitulate known metabolic perturbations associated with a high sugar diet (HSD) — a widely used model for type 2 diabetes in *Drosophila*, but also reveal new mechanistic insight by identifying altered NADH-dependent reactions and GAPDH as a rate-controlling step in response to HSD.

3. Identification of pathway-level perturbation using PFVI analysis (**Fig 6**):

We present the Pathway Flux Variability Index (PFVI) to prioritize additional dysregulated pathways under HSD conditions. This approach enabled rapid hypothesis generation and allowed us to focus on specific perturbations in muscle in response to HSD. Notably, PFVI identified strong downregulation of fructose metabolism — a pathway not classically associated with muscle — highlighting a previously underappreciated role of muscle in fructose clearance.

While our earlier version briefly mentioned cachexia as a future application of tissue-specific GEMs, we have removed this discussion in the revised manuscript to maintain a focused narrative centered on high sugar diet-induced metabolic remodeling.

“- The title emphasizes the application of studying the effects of high sugar diet; however, most of the paper focuses on comparing the models (p. 6-10; figs 1-4). Only the last part is about high sugar diet (p. 11-13). And related to comment 1 above, the question of how we can use their results remains.”

We thank the reviewer for this thoughtful observation. We agree that, in the original version of the manuscript, a substantial portion of the Results (original figures 1–4) was devoted to comparative analyses of tissue-specific GEMs. In the revised manuscript, we removed the Vmax and flux variability analyses (originally figure 2-3), as they introduced assumptions that were difficult to validate for individual tissues.

In our revised manuscript, we now placed greater emphasis on the application of the muscle-GEM to study high sugar diet (HSD)-induced metabolic perturbations (**Figures 3–6**), while reserving Figures 1 and 2 for the reconstruction and validation of tissue-specific GEMs. These changes better align the manuscript

with the focus suggested by the title and more clearly demonstrate the practical utility of our modeling approach.

Additionally, as now discussed in the revised Discussion (**Lines 375–379**), we highlighted that other tissue-specific GEMs—validated through pathway enrichment and metabolomics—can be applied to investigate tissue inter-organ metabolic communication, as well as to support more focused metabolic studies, such as those presented here for muscle in response to high sugar diet.

“- A central finding for predicting metabolic features with high sugar diet is that GAPDH is a bottleneck reaction (** note, in multiple places, “GADPH” is incorrectly used). This is not a novel result, as GAPDH is widely known to be a limiting reaction in many conditions. This further emphasizes the need to explain why the reader should be excited about the results presented.”

We thank the reviewer for highlighting this point. While GAPDH is a well-known bottleneck enzyme in glycolysis, particularly in cancer and proliferative systems (PMID: 25009227, 28918937), our study provides new mechanistic insight about the role of GAPDH in muscle under high sugar diet (HSD) conditions.

1. Context-specificity: Using muscle-GEM, we identified GAPDH as a rate-controlling step in *Drosophila* muscle in response to HSD — a context in which it has not been widely studied. Previous studies were shown in context of cancer cell lines (**Discussion, line 382-384; Figure 3-4**)
2. Redox-regulation of GAPDH: We identified methionine oxidation of GAPDH occurring near the NAD⁺ binding site, implicating redox modifications, rather than the changes in enzyme abundance, were linked to decreased GAPDH activity under HSD. (**Discussion, line 384-391; Figure 5**)
3. Beyond GAPDH: We also introduced a Pathway Flux Variability Index (PFVI) to systematically identify other dysregulated pathways. Using PFVI, we found a significant downregulation of fructose metabolism in muscle under HSD — a pathway not traditionally associated with muscle function. This expands the utility of our approach by highlighting potentially overlooked muscle metabolism and offering direction for future mechanistic studies (**Discussion, Lines 394–417; Figure 6**).

Together, these findings demonstrate how tissue-specific GEMs can be applied not only to identify known regulatory nodes like GAPDH in HSD condition, but also to identify broader pathway-level metabolic dysregulation in a quantitative and hypothesis-driven manner.

“- The authors should add a paragraph to the Discussion about limitations and assumptions that affect their findings. For example, using estimates of protein abundances and limited availability of k_{cat} values are expected to influence the results and subsequent conclusions.”

We thank the reviewer for this important suggestion. In the revised manuscript, we have added a dedicated paragraph in the Discussion (**Lines 418–434**) to address key limitations and assumptions of our study. Specifically, we recognized that estimating tissue-specific V_{max} values using transcriptomics and whole-organism k_{cat} datasets introduced significant uncertainty and posed challenges for validation. As a result, we have removed all V_{max} -based constraint modeling and associated analyses (originally presented in Figures 2–3).

We also discuss other limitations, including the impact of GEM reconstruction algorithms (e.g., tINIT vs. CORDA) and the difficulty of experimentally validating all 32 tissue-specific models (**Lines 418–434**). While we provided experimental validation for the muscle-GEM using metabolomics, ^{13}C -tracing, and redox proteomics, we acknowledge that comprehensive validation across tissues will require future efforts incorporating tissue-specific proteomics and kinetic measurements to further improve model accuracy and biological relevance.

Additional issues:

- The “clusters” shown in Figure 1b are not distinguishable. A more quantitative/rigorous definition of the clusters is needed, as many appear to overlap.

We thank the reviewer for pointing this out. To address this concern, we performed hierarchical clustering on the same reaction presence/absence matrix and described 13 distinct tissue clusters (**Supplementary Fig. 7, Supplementary Data 1f**), which are now referenced in the main text. To further evaluate the biological relevance of these clusters, we compared them against transcriptomic-based tissue clusters using the Jaccard index (**Fig. 1c**).

- The paper presents that tissues containing enzymes with high V_{max} are able to complete more metabolic tasks (fig. 2b). There appear to be several counter examples to this that should be discussed. Do the points corresponding to low V_{max} represent a low ability to perform tasks? Is this meaningful?

We thank the reviewer for pointing this out. In the revised manuscript, we have removed all analyses related to V_{max} -based constraints, including comparisons

of Vmax values and their relationship to metabolic task performance. These analyses relied on approximated enzyme abundances and kcat values, which introduced substantial uncertainty and were challenging to validate across individual tissues. By removing these sections, the revised manuscript now focuses on GEM simulations using standard constraint-based methods such as FBA, FVA, and sensitivity analysis, thereby avoiding assumptions that could confound interpretation.

- A finding from the flux variability analysis is there are “similarities and differences across tissues” (line 225). This is a very vague sentence that is not insightful or actionable. Again, it aligns with the major critique that the results are not presented in a way that highlights their utility.

We thank the reviewer for this helpful observation. We agree that the original sentence was vague and might not convey significance. In the revised manuscript, we have removed the tissue comparison based on flux variability analysis, as this section relied on assumptions that were difficult to validate and did not contribute meaningfully to our central conclusions. The revised Results now focus on model construction, validation, and the application of muscle-GEM under high sugar diet conditions, which we believe better highlight the utility of our approach (**Figure 1 and 2**).

- It would be helpful to the reader to have a table that presents the major model predictions and whether each aligns with experimental evidence (either from this paper or from published works). This would better support the authors' statements that model predictions are consistent with experimental findings.

We thank the reviewer for this helpful suggestion. We agree that summarizing model predictions alongside supporting literature evidence can improve clarity and reader accessibility. While we considered providing a single summary table, many of the predictions from tissue-specific GEMs are highly context-dependent, making it difficult to generalize validation outcomes across all models in a tabular format.

Instead, we focused on our experimentally measured pathway-level validation strategy. Specifically, we validated our tissue-specific GEM predictions by comparing them with region-specific metabolomics and pathway enrichment analysis results (**Figures 2e–h, Supplementary Data 2–3**). This approach allowed us to assess the biological relevance of each model by evaluating

whether the pathways predicted to be active in each tissue were also enriched in the corresponding metabolite profiles (**Line 625 - 651**).

Additionally, for the high sugar diet (HSD) application, we provided multiple layers of experimental validation — including ¹³C-glucose tracing and metabolomics— to support the predictions made by the muscle-GEM (**Figures 3–6**). We hope this integrative strategy offers the reader a robust framework for evaluating the consistency between model predictions and experimental findings, even in the absence of a single summary table.

Reviewer #5 (Remarks to the Author):

The manuscript by Sun Jin Moon et al. represents a comprehensive study on the use of tissue-specific GEMs to study the effect of a high glucose diet in *Drosophila*. This is quite an impressive study in scope, and will be a good contribution and highly cited study. I think the authors should feel proud of such comprehensive work. However, I think there are still some very important items that need to be addressed regarding the model construction and validation process, along with statistical analyses, which the authors should address, given that others in our community will be using these models, and the models themselves are critical to the claims of the paper. Also there are many critical details that are glossed over, along with community standards in tissue-specific model construction that need to be addressed. However, I hope to see these concerns addressed since I'm excited about this fine work!

Major Comments:

“- We applaud the effort to further refine the *Drosophila* reconstruction. These efforts for improving network curation are valuable to the community.”

We thank the reviewer for this encouraging comment and greatly appreciate the recognition of our efforts to improve the *Drosophila* metabolic reconstruction. It is our hope that these resources will support future studies in the field.

“- Numerous algorithms exist for generating tissue specific models, and a number of decisions need to be made in processing the data. Such decisions can have a huge impact on the content that is obtained in tissue-specific models, and thus any interpretations of model analyses. Why was tINIT selected, as opposed to mCADRE, CORDA, or other high-quality methods? What validations were done to believe that the models obtained by tINIT were valid? Often, community best-practices validate models by using gene knockouts, correlations to experimental metabolic fluxes (though this is confounded by the models used for fitting C-13 data), cell culture growth rates compared to model growth rates, other experimental validations, or comparison of model functions to comprehensive lists of known metabolic functions (not just citing a few papers that suggest a metabolic function could be active in a couple models, as done here for muscle metabolism). This is critical since other studies (e.g., Opdam, Cell Systems, 2017; Machado, PLoS Comp Bio, 2014) showed some methods were quite poor at making reliable models.”

We thank the reviewer for this thoughtful and important comments. We agree that the selection of a tissue-specific model reconstruction algorithm can significantly influence model content and downstream interpretations. We now explicitly acknowledge this in the Discussion section (**lines 418–434**) as a limitation of our study, emphasizing the caution and impact of reconstruction choices on model performance and interpretability. In fact, we used CORDA algorithm to provide as an example, showing some common trends and differences obtained from the same dataset (Error! Reference source not found. - **56**).

Regarding the rationale of using tINIT algorithm, we used it because it incorporates both gene expression data and predefined metabolic tasks, which ensure that the network carries fluxes in the core metabolic reactions. While we believe other algorithms are still powerful as they are continuously used in other studies for context-specific GEM reconstruction, we found tINIT as a suitable and working algorithm especially since the generic *Drosophila* model we used are compatible with this algorithm. Again, we discussed the choice of model algorithm in the discussion section. We wrote details in method section (**Line 476 – 504**).

We thank the reviewer for the comment on validation. We acknowledge that in our original manuscript, we lacked quantitative and rigorous validation with respect to the reconstructed GEMs, as we validate the differences in metabolic networks and functions through literature evidence and a few enriched metabolites identified from metabolomics.

In our revised manuscript, we now include a dedicated results section (**Line 160-208**) where we systematically compare model predictions to experimental data. In brief, we validated the model-predicted enriched pathways with metabolomics-based pathway enrichment analysis using weighted Jaccard indices (**Line 625 - 652**). We chose this pathway-level validation approach because obtaining direct growth rates or performing gene essentiality analysis for all 32 tissues is not feasible in our study. While we also explored correlating model-predicted metabolite demand fluxes with region-specific enriched metabolites—following the concept from Lewis et al., *Cell Systems*, 2021—this approach proved unsuitable as we observed poor correlation across datasets.

With the pathway-level validation strategy, we demonstrated strong agreement between model-predicted and metabolomics-derived enriched pathways, particularly for muscle and fat body (**Figure 2, line 193-208**). These results provide support for the accuracy of tissue-specific GEMs. Nonetheless, we acknowledge the challenges in comprehensively validating all 32 models and

note that future work should include more context-specific validation strategies to further improve model performance (**Discussion, Lines 429–434**).

“- tINIT, as its name suggests, requires defined metabolic tasks. How were these defined? The methods just said they used a list of 57 tasks in building the models. What were these tasks, how were they defined, and what is the support for each task? (see, e.g., Thiele, Nat Biotech, 2013; Blais Nat Commun, 2017; Richelle, PLoS Comp Bio, 2019). They later mentioned using 256 human metabolism tasks to test model functions. How many of these tasks are relevant to *Drosophila*, as opposed to just human?”

We thank the reviewer for this important question. As noted, the tINIT algorithm requires a list of essential metabolic tasks to guide reconstruction. In our study, we used a set of 57 essential tasks adapted from the human metabolic task list presented in Robinson et al., Science signaling, 2020. These tasks cover fundamental biological functions (see **Supplementary Data 1b**).

Although these tasks were originally defined for human GEMs, we reviewed *Drosophila*-specific resources including KEGG pathway annotations and FlyBase, and found no direct evidence that these essential functions would be biologically implausible in *Drosophila melanogaster*. Additionally, while KEGG identified several *Drosophila*-specific pathways (e.g., insect hormone biosynthesis, Toll and Imd signaling, dorso-ventral axis formation), we did not include these as essential tasks in the reconstruction step because their essentiality across all 32 tissues was unclear. However, trehalose biosynthesis, a key metabolic feature in flies, was later included in our metabolic general metabolic tasks. The full list of tasks used for model reconstruction, along with associated inputs, outputs, and categories, is now provided in **Supplementary Data 1b** and described in the revised Methods (**line 545-556**).

In addition, we evaluated metabolic functionality considering both predefined metabolic tasks, curated from Richelle et al. (*PLoS Comput Biol*, 2019) and Robison et al., (Science Signaling, 2020). While these tasks were originally defined for human models, they represent broadly conserved functions. We acknowledge that we did not manually verify each task’s relevance to *Drosophila*. In fact, of the 219 tasks tested, only 93 (42%) were passed in at least one tissue-specific GEM, suggesting that the remaining tasks may reflect functions that are either not conserved in *Drosophila* or not captured due to reconstruction limitations (e.g., unoptimized parameters, incomplete gene annotations, or expression thresholds) (**Supplementary figure 12**). A more systematic evaluation of *Drosophila*-specific task lists and tissue-relevant benchmarks will

be a valuable direction for future refinement and assessment of model functionality.

Nevertheless, we validated the biological relevance of our tissue-specific GEMs through experimentally measured region-specific metabolomics and pathway analysis, confirming the accuracy of model predictions, particularly for muscle metabolism.

“- Many tissue-specific model generation methods require one to define which genes are expressed in a given tissue. How was this performed? Was it done by global or local thresholding? (Richelle, PLoS Comp Bio, 2019)”

We thank the reviewer for this important question. We applied global thresholding based on normalized gene expression values derived from Fly Cell Atlas pseudo-bulk snRNA-seq data. We chose a unified thresholding approach to avoid additional variability that could arise from using tissue-specific (local) thresholds, which may introduce bias or inconsistency across tissues.

For tINIT, we applied a uniform and relaxed gene expression threshold of one. This value was selected based on the distribution of average gene expression levels across tissues and standard deviation (mean – 1xSD = 1.02), ensuring the inclusion of moderately expressed genes while maintaining consistency across tissues (**Line 494 – 504**). We also described the threshold used for CORDA algorithm in method (**Line 498 – 504**). We also acknowledge that these threshold values were not fully optimized and that thresholding strategies can certainly influence the final model content and predictive performance (**Line 431 – 434**).

“- In figure 1c only 37/256 tasks are shown. It is stated that these were significant. How was significance defined and determined? “

We thank the reviewer for pointing this out and apologize for the confusion. Figure 1c (now **Figure 1f**) does not present results from the metabolic task analysis, but instead shows subsystem coverage across the tissue-specific GEMs. In this figure, we highlight 33 metabolic subsystems (not tasks) that showed more than 50% deviation in reaction coverage compared to the average across all models (see Methods, **Lines 536–544**). We acknowledge that our use of the term “significant” may have been misleading, as it could imply statistical significance. In this context, “significant” referred to the magnitude of deviation from the mean, not to a statistical test. To avoid confusion, we have removed this term from the revised manuscript (**Line 140 - 144**)

“- Figure 1d is unclear, and just reporting a Z-score provides limited insight. What % of tasks were functional for the different tissues? Do these conform with known tissue and cell-type specific tasks better than permuting tissue labels and comparing tasks with known tissue-specific functions? On a related topic, it seems that the authors provide some literature support for a couple tissue-specific functions, but can they define tissue-specific functions prior to the analysis so that they can do a proper assessment of true/false positives/negatives to make sure their models are better than randomly-generated models?”

We thank the reviewer for this insightful comment. To address the concern, we have included additional figures and clarifications in the revised manuscript. Specifically, **Supplementary Fig. 12** now shows the metabolic task pass rate per tissue, e.g., the percentage of the 219 tasks successfully carried out by each GEM. Additionally, **supplementary Fig. 13** now also presents the tissue-specific enrichment of metabolic systems, following the procedure from Richelle et al., 2021, to identify nonrandom associations between tissue identity and functional capacity (**Line 151-155, 554 - 556**)

Regarding our metabolic task analysis, we did not perform permutation testing of tissue labels, as this would require a predefined, standardized list of tissue-specific metabolic functions. However, such functions are often context-dependent, making it difficult to define definitive true or false assignments for each tissue. However, we did manually confirm several metabolic tasks with literature support for specific tissues. For example, trehalose biosynthesis is a well-known function of the fat body in *Drosophila*, and this task was successfully passed in our fat-body GEMs (**line 148 – 151**).

Moreover, rather than emphasizing metabolic task functionality, we focused on characterizing enriched metabolic pathways and validating these predictions using metabolomics-based pathway enrichment analysis, incorporating permutation-based statistical analyses (**Line 193 - 208, Figure 2f-h**).

Lastly, we acknowledge that some false positives may exist in the metabolic task analysis. We agree that defining tissue-specific metabolic tasks more rigorously and systematically evaluating true and false positives would be an important next step for future studies building on this work.

“- How much does the Vmax for specific reactions vary across tissues? Do you see, e.g., glycolytic and lactogenic Vmax values to be higher for more metabolically active cells, like neurons?”

We thank the reviewer for this thoughtful comment. In the revised manuscript, we have removed all Vmax-associated calculations and analyses (Original figure 2 and 3). It was because validating Vmax calculations across all 32 tissue-specific GEMs proved challenging given the current lack of comprehensive, tissue-specific kinetic parameters for *Drosophila*.

“- In setting up simulations for FBA and related approaches, one must consider several things, including the objective function. Along these lines, multiple studies have benchmarked these, but I don't see them referenced. For example, in lines 183-186 the authors explain that all flux analyses in the paper are being conducted by setting the objective with a DM reaction that takes into consideration ATP, NAD and NADPH production. The authors state that this is a common metabolic objective across tissues. However, they are not encountering other metabolic activities that the cell is subjected to and is represented by the biomass function. In this regard the authors should cite previous work that have carried a similar approach and also justify the addition of the NAD⁺ to the objective function as they only explained the incorporation of NADPH and ATP to the objective. This could be easily done by referencing the data and rationale in papers such as Savinell and Palsson, J Theor Bio, 1992; Schinn, Metabolic Engineering, 2021; Schuetz, Mol Syst Bio, 2007”

We thank the reviewer for this important point regarding the objective function used in our flux simulations. In our original submission, we used a custom demand reaction that included ATP, NADPH, and NAD⁺, aiming to capture general energetic and redox demands across tissues (**Line 664 - 669**). This formulation was inspired by prior studies modeling basal cellular energy and redox homeostasis (Lewis et al., ANTIOXIDANTS & REDOX SIGNALING, 2018).

However, in the revised manuscript, we did not use this composite ATP/NAD/NADPH objective and instead adopted a biomass objective function (reaction ID: MAR00021) for all simulations. We used this formulation as a general proxy basal maintenance functions, such as protein turnover, lipid remodeling, and nucleotide recycling, which are physiologically relevant in adult *Drosophila* tissues (**Ling 707 – 714**). This adjustment also addressed concerns that including NAD⁺ or NADPH directly in the objective might introduce artificial sinks and lead to unphysiological cofactor cycling. Our simulations now reflect more general flux distributions aligned with basal cellular maintenance needs, rather than imposing specific energy or redox requirements, which may bias the simulations. We have clarified this update in the revised Methods section. We also ensured that the flux through biomass was maintained and confirmed at least two orders of magnitude lower than glycolysis.

Importantly, we ensured that the following GEM-based predictions were validated with orthogonal experimental measurements, such as ^{13}C -glucose tracing and metabolomics-based pathway enrichment analysis. Finally, we acknowledge that different tissues likely pursue distinct physiological objectives depending on their function and context. Thus, we acknowledge the limitation of using this objective function in discussion (**Line 431 - 434**)

“- A lot of flux spans are reported from flux variability analysis. These are merely descriptions of the models and parameters that were inferred. I only see one single citation mentioning how the span on follicle and germ cells could be meaningful. Is there any biological support for all of the differences discussed between lines 194-229?”

We thank the reviewer for this important point. In the revised manuscript, we have removed the flux span–based comparisons across tissues derived from flux variability analysis (FVA), including the content previously discussed between lines 194–229 (previously figure 3). We removed these results, because we found it difficult to validate the resulting flux distributions across all 32 tissue-GEMs. Instead, we focused our analyses on a specific muscle metabolism in response to high sugar diet conditions (**Figure 3 – 6**).

- Supplementary Fig 1 needs to incorporate labels that indicates the meaning of each color. Specifically panel A and E.

We thank the reviewer for pointing this out. The original Supplementary Fig. 1A and 1E have now been reorganized into **Supplementary Figs. 2, 3, and 5**. In the revised figures, we have annotated the color scales to clearly indicate what each color represents. Additionally, we ensured that all heatmaps throughout the supplementary figures now include scale bars to improve clarity and interpretability.

- Supplementary Fig 3 needs to incorporate labels that indicates the meaning of each color. Specifically panel B, H and I.

We thank the reviewer for this helpful comment. The analysis corresponding to the original Supplementary Fig. 3 has been removed from the revised manuscript during the restructuring of our results and supplementary materials. As such, these panels no longer appear in the revised manuscript.

“- In the metabolic profiling section starting with line 231, along with figures 4 and S4, the authors have generated a fantastic dataset and provided a mildly qualitative

analysis, but with no statistical tests to see if there is a significant overlap between the metabolomic data and their flux analyses. However, in theory, metabolic task analysis, which they did for figure 1, should allow one to see, for each tissue, which metabolic tasks are more active (i.e., which metabolites could increase in activity). I believe Sanjay Nigam's lab at UCSD did such an analysis that he presented 3 or 4 years ago. Thus, it would provide much greater confidence in their tissue-specific models, if the authors here did a quantitative analysis comparing the metabolic data to the tissue-specific metabolic task activities."

We thank the reviewer for this insightful suggestion. We agree that a quantitative comparison between metabolomic profiles and predicted model functions is important for assessing the validity of tissue-specific GEMs. While we did not directly compare metabolite levels with metabolic task analysis, we performed an alternative comparison using KEGG pathway analysis.

In the revised version (now **Figure 2**), we addressed this by: (1) performing pathway enrichment analysis on region-specific metabolomics data, and (2) comparing the enriched pathways to those predicted by tissue-specific GEMs. To quantify the agreement, we computed weighted Jaccard indices, which represent pathway overlap similarities (see Methods, **line 626 - 652**). This quantification systematically evaluated the performance of tissue-specific GEMs in identifying enriched pathways for each tissue, with muscle and fat-body showing strong agreement (**Figure 2g-h, Line 197 - 202**). Additionally, we relocated metabolites comparisons found in original figure 4c-f to the **supplementary figure 16 - 21**.

We also considered comparing metabolite levels to the outputs of the metabolic task analysis. However, since the task outputs are binary (pass/fail) and do not reflect the degree of pathway activity or enrichment, we found it difficult to make direct comparisons with metabolomics data, especially the metabolite levels were function of both production and consumption rates. Therefore, we focused our validation efforts on a pathway-level comparison, using metabolomics-based KEGG pathway enrichment analysis.

"- Line 342, the authors say there's a decrease in PFI. What was the magnitude and statistical significance of the decreases? Bordel, et al. PLoS Comp Bio, 2010 and Bar Evan, PNAS, 2010 from the Nielsen and Milo labs, respectively, presented ways to quantify significance of changes in pathway fluxes using different sampling approaches. We've personally seen in many instances increases and decreases in flux are not significant if using empirical p-values, but the approach is valuable for finding the ones that really are."

We thank the reviewer for this valuable suggestion. In response, we have now quantified the statistical significance of pathway-level flux changes between NSD and HSD conditions by implementing a two-sample Z-test, as suggested by the reviewer and in line with prior studies such as Bordel et al., 2010 (PLoS Comput Biol). We described the details in method section (**line 334 – 336** and **line 828 - 837**). Because of the large sample size used for flux sampling, most pathways showed statistically significant differences. However, we acknowledge that reducing the number of simulations may affect the statistical power and alter the significance of some pathway-level differences.

Importantly, we validated the PFVI based analysis using orthogonal experimental analysis, as described in the revised Results (**Figure 6f, Lines 346–358**). Notably, we found that three of the six most significantly altered metabolites (from 283 detected metabolites) were indeed associated with the top differentially affected pathways identified by PFVI, confirming the validity of PFVI-based analysis.

- The PFV subtracts the max and min fluxes for the reactions in a pathway, from what I understand. Shouldn't this consider the absolute value, since sometimes reactions are written backwards from the way they're used, and thus would contribute a negative flux value. On a related note, some reactions will be heavily skewed when doing sampling. Did you try using the mode instead for the mean flux for PFI?

Thank you for these insightful comments. We have clarified the PFVI equation in the Methods section (**line 816 - 826**) to more explicitly explain the rationale. In calculation of the Pathway Flux Variability Index (PFVI), since V_{max} was always greater than V_{min} by definition, this difference is always a positive value or zero, independent of the directionality or sign of the reaction flux.

Regarding your point on skewed flux distributions, we re-analyzed all sampled flux distributions and found that indeed approximately 80% exhibited positive skewness (**Supplementary Fig. 56**). Based on this observation, we revised the PFVI calculation to use the mode of the absolute sampled flux to better reflect the central tendency of flux distributions under skewed conditions. The results sections and figures regarding PFVI are updated accordingly (**Figure 6a–b**).

Importantly, regardless of using mean or mode, the major trend remained mostly similar — for instance, fructose metabolism remained among the most significantly downregulated pathways under HSD conditions using either the mean or mode. However, the mode allowed for improved detection of upregulated pathways with skewed distributions, particularly those associated with fatty acid metabolism. This adjustment accurately captured biologically

known features of metabolic remodeling in diabetic muscle. We thank the reviewer again for this valuable suggestion.

Minor Comments:

- Line 136: Figure incorrectly referenced. Correct reference is Supplementary fig. 1e, instead of Supplementary fig. 1d.

These supplementary figures were removed as they were not essential for supporting the key findings presented in the revised manuscript.

- Line 137: Is not clear what the authors are stating. I suggest rephrasing this sentence for better understanding.

On the same note as above, the associated statement has also been removed in the revised manuscript.

- Figure 2a should be incorporated as an equation in the manuscript and being removed from Figure 2.

We have removed the equations from the figures.

- As in Figure 2a, Supplementary Figure 2 equations in a and e should be addressed as equations rather than a panel in the figure.

We have removed associated equations from the figures.

- Figure legend of Supplementary Fig 4 panels h and I are wrongly referenced. I suggest reorganizing the figure.

Thank you for this suggestion. Supplementary Fig4h-I are reorganized and now shown as supplementary Figure 21.

- Supplementary Fig 5, heatmaps corresponding to panels b and c should be increased in resolution for better readability.

In the revised supplementary figures, these subpanels were removed as they were not essential for supporting the key findings presented.

- Line 289: Add “Glyceraldehyde 3-phosphate” initials for better localization in the bar plot of Supplementary Fig.5a.

We have replaced the abbreviation for glyceraldehyde 3-phosphate with GAP throughout the revised manuscript.

Reviewer #6 (Remarks to the Author):

REVIEWER COMMENTS

Reviewer #1 (Remarks to the Author):

The authors have performed considerable additional experimentation in the revision process. This has addressed many/most of the comments raised. The manuscript is much improved.

Reviewer #2 (Remarks to the Author):

GENERAL COMMENTS

The authors made significant changes in their predictive simulations with the fly tissue GEMs and in the organization of the manuscript. The methodology has been simplified by replacing the previous modeling approach, which relied on kinetic parameters, with regular FBA. Despite this simplification, the predictive power is maintained (or possibly improved), consistent with earlier criticism that the added complexity from inaccurate kinetic parameters was not necessarily beneficial. The methodology is now clearer, aided by better manuscript organization and more straightforward language, in addition to the simplifications in modeling.

However, some methodological concerns remain. A major issue is the treatment of reaction boundaries. As in many GEMs, including the human model used here, reaction flux bounds are uniformly set to +/-1000. This arbitrary value is not biologically meaningful but is meant to represent an effectively unbounded range, under the assumption that biologically relevant fluxes (e.g., those obtained with reasonable nutrient uptake rates and predicting reasonable biomass production rates) will not approach these limits. The authors' approach, however, treats these arbitrary values as literal and perturbs them in some of their tests (see comments below), which raises technical concerns.

In addition, there are some remaining issues related to quality and clarity. Detailed comments are provided below for another round of revision.

We thank the reviewer for the thoughtful summary and for recognizing our efforts to simplify and clarify the modeling framework. We have addressed all methodological and clarity issues, including those related to flux boundaries, sensitivity analysis, and pathway-level flux analysis. We also clarified that the default flux bounds are computational conventions rather than physiological limits and noted this in the *Limitations* section to describe the semi-quantitative nature of our flux predictions given the limited availability of *in vivo* nutrients exchange data (**Pg. 16, line 441 - 445**).

Specifically, in this revision, we have performed additional analyses, including parsimonious FBA (pFBA), NADH demand function maximization, and representative statistical approaches for pathway-level flux analyses, as suggested by the reviewer. We believe these revisions substantially strengthen the methodological rigor and interpretability of our study.

MAJOR COMMENTS

1) The rationale for applying perturbations by altering reaction boundaries is unclear. A 20% reduction in the upper flux bound (or increase in lower bound) is arbitrary and does not affect all reactions equally. For example, if Reaction 1 and Reaction 2 have fluxes of 700 and 850 mmol/gDW/hr, lowering the upper bound from 1000 to 800 will have no impact on Reaction 1, but will constrain Reaction 2, as its reference flux exceeds the new boundary. This inconsistency undermines the interpretation of such perturbations.

A more principled approach would be to apply perturbations directly to the flux values themselves; for example, by imposing a constraint that forces a reaction to carry no more than 80% of its unperturbed flux. While I acknowledge that the authors rely on flux sampling and do not define a single reference flux, it would be more appropriate to apply perturbations based on representative statistics of the sampled distribution (e.g., the mode or median), rather than manipulating arbitrary boundary values.

This would avoid the conceptual problem of treating +/-1000 as meaningful limits and ensure a more uniform and interpretable perturbation strategy.

We thank the reviewer for highlighting this important methodological issue. We agree that applying perturbations by modifying reaction boundaries could lead to non-uniform perturbation effects across reactions, and that directly perturbing flux values based on representative statistics (e.g., mode or median) is a more rigorous approach.

In the revised manuscript, we updated our method by directly applying perturbations to the baseline flux as calculated by the median of the sampled distribution (**Pg 30, Method: "sensitivity analysis"**). We confirmed that the revised perturbation method generated uniform and proportional perturbations across reactions (**Supple fig 3I**). Additionally, instead of focusing on a fixed 20% perturbation value, we also varied perturbation magnitudes (5, 10, 20, 30, and 50%) to further evaluate the effects across different magnitudes (**Fig 3g and h**).

While the earlier sensitivity analysis revealed that GADPH showed the highest sensitivity and potentially served as a rate controlling step, the updated analysis revealed that GAPDH is not a sole rate-controlling enzyme as initially hypothesized, but the control is likely to be distributed among several other enzymes, such as Aldo, P_{gk}, and Gapdh, and varies with the perturbation magnitude under this condition (**Fig 3g-h; Supple Fig. 3m; Pg 10-11, line 250 – 271**). We made a separate result section dedicated for this updated sensitivity analysis. Additionally, All relevant codes, analyses, and documentation have been updated and are available on Github (3_flux_analysis/Sensitivity_analysis).

Overall, we now provide a more consistent and methodologically grounded framework for evaluating flux sensitivities.

2) Related to the issue above, some of the flux sampling distributions shown in Supplementary Figure 49 appear to cluster near the flux boundaries. This raises three concerning possibilities:

(i) The default flux bounds (± 1000), which are conventionally intended to approximate infinity, may be too restrictive relative to the feasible solution space. In this case, increasing the bounds (e.g., to ± 10000) might be appropriate to avoid artificial truncation of the distributions.

(ii) The nutrient uptake rates may not be well constrained, allowing unrealistically high fluxes. As a result, key fluxes, particularly for the uptake of limiting nutrients, may saturate at the default boundary of 1000, leading to biologically implausible solutions.

(iii) The high-magnitude fluxes may reflect the presence of internal flux loops. For example, one reaction ($A + \text{NADH} \leftrightarrow C + \text{NAD}$) may carry a flux of +1000, while a parallel reaction ($A + \text{NADPH} \leftrightarrow C + \text{NADP}$) carries -990, yielding a net flux of 10 while recycling redox cofactors through an internal loop. This scenario could be evaluated by performing parsimonious FBA (pFBA), which minimizes the total flux and suppresses such loops by favoring net conversions over unnecessary cycling.

More broadly, it would be informative to compare the predictive power of simple pFBA solutions to that of flux samples. Given the potential ambiguity in sampled distributions, especially when they include artificial extremes or loops, pFBA could offer a more interpretable baseline for certain predictions.

We thank the reviewer for highlighting these important points regarding the potential flux boundary effects and internal flux loops, and for suggesting the use of parsimonious FBA (pFBA) as a complementary approach. We carefully evaluated each of the three possibilities (**Supplementary Notes 3; supplementary Fig. 3d-g**). Our findings are summarized below.

(i) Boundary magnitude effects:

To examine whether the default flux bounds (± 1000) artificially restricted feasible flux ranges, we repeated FVA-sampling analyses under expanded boundary conditions ($\pm 10,000$ and $\pm 50,000$) (**Supplementary Note 3c**). Using both maximum and median flux criteria (within 1% of boundary values), we found that only a small fraction of reactions (about 2% for upper bounds and 1% for lower bounds) approached saturation across all conditions (**Supplementary Fig. 3g**). Although this percentage depends on the precise threshold used to define boundary proximity, we applied a stringent 1% cutoff to conservatively identify reactions potentially affected by numerical limits. These results indicate that default bounds did not significantly constrain the feasible solution space or artificially truncate flux distributions for most

reactions. All relevant codes, analyses, and documentation have been updated and are available on Github (3_flux_analysis/ Boundary_effects).

(ii) Nutrient uptake constraints:

We agree that physiologically relevant constraints on nutrient uptake and secretion rates are critical for accurate flux predictions and preventing flux boundary effects. However, *in vivo* nutrient exchange rates for *Drosophila* tissues remain largely unavailable, making it difficult to impose constraints with high confidence. Despite this limitation, we applied constraints based on prior literature and metabolomics evidence and validated model-predicted flux changes in glucose metabolism with glucose uptake measurements and isotope tracing experiments (**Supplementary Fig. 3c** and **Fig. 4a-d**). We acknowledge that incomplete nutrient constraints are likely to contribute to boundary saturation, and we note in discussion that obtaining quantitative *in vivo* nutrient exchange data remains a major challenge for accurate flux predictions in *Drosophila* (**Pg 16, line 441 - 445**).

(iii) Internal flux loops and pFBA comparison:

To address the concern of internal flux loops and artificial high fluxes, we implemented parsimonious flux balance analysis (pFBA) (**Pg 28, Methods: “pFBA”, and Supplementary Note 3a-b**). Compared with the sampling analysis, pFBA substantially reduced the total flux within the network (~76–85%) and decreased the number of reactions carrying fluxes (**Supplementary Fig. 3d–e**). Additionally, we identified that several NAD(P)-dependent cycling reactions showed significantly reduced fluxes with pFBA (**Supplementary Fig. 3f**). These confirm that pFBA is effective in suppressing artificial flux extremes and minimizing internal flux loops.

Overall, these analyses indicate that the apparent boundary effects observed in our original results were likely caused by artificial flux extremes and internal flux loops by FVA_sampling analysis rather than restrictive flux limits. We cannot rule out contributions from incomplete nutrient constraints, which are likely to play a role, and this remains important direction for future investigation.

Additionally, based on this analysis, we integrated pFBA throughout the revised study. Specifically, our semi-quantitative nature of flux analyses now focused on identifying commonly perturbed reaction sets that are consistently detected across pFBA, FBA, and FVA-sampling analyses (**Fig. 3a–c; Supplementary Fig. 3h-j; pg 9, line 218 –**

230; pg 29, Methods: differential flux analyses). As our flux analyses are limited in accurately predicting the absolute flux values, we de-emphasized individual flux distributions of specific reactions (e.g., removing originally supplementary fig 49) and changed the unit of flux values to arbitrary unit (a.u.) from mmol/gDCW/hr. All relevant codes, analyses, and documentation have been updated and are available on Github (*3_flux_analysis/pFBA_FBA_FVA*).

Overall, we believe this revision provides a more robust and interpretable framework of our flux analyses, addressing the reviewer's suggestions.

3) The rationale for using flux variability range as a multiplier in the Pathway Flux Variability Index (PFVI) is not well justified. The interpretation of this index appears problematic, particularly in cases where pathways are essential and consistently active. In such cases, reactions may exhibit narrow flux ranges that do not include zero (reflecting obligatory activity) but the PFVI will be suppressed due to the low variability. According to the current interpretation, this could misleadingly suggest low pathway activity.

Conversely, as illustrated in Supplementary Figure 50a, a reaction with a broad flux range that spans zero (e.g., the red curve) may be inactive in many feasible solutions, yet it receives a high PFVI score due to its variability. This undermines the biological interpretation of PFVI as a proxy for pathway activity.

This issue should be addressed either by adjusting the PFVI formulation or clarifying its intended meaning. One might also question whether similar qualitative predictions could be made without using variability at all; e.g., by relying solely on representative flux modes (or pFBA fluxes, as discussed in the previous comment). This would avoid conflating variability with activity and may yield more interpretable results.

We thank the reviewer for valuable insight regarding the formulation of the Pathway Flux Variability Index (PFVI). We agree that conflating flux variability with pathway activity can be problematic, and further investigation is needed to use this metric.

In response, we revised our approach by re-defining pathway flux as the average of non-zero flux magnitudes within each pathway, removing the previous weighting by flux variability (**Pg 30, Methods:** "Pathway-level flux analysis"; **Fig 6a**). The revised formulation follows a more straightforward and widely accepted statistical framework, using two-sample Z-tests on flux distributions to identify pathways with significant flux changes.

To note, results from the revised pathway flux analysis remain generally consistent with those obtained using the previous variability-weighted index, suggesting that variability weighting made less impact than initially assumed. However, further investigation is required to rigorously adapt such index as a new metric.

Moreover, we now compared pathway fluxes across both FVA-sampling and pFBA analyses and highlighted the commonly perturbed pathways identified by both methods, improving flux analysis consistency (**Fig. 6a-b**). The sampling-based method identified a larger number of perturbed pathways compared to the more conservative pFBA approach. Importantly, both methods consistently captured key known pathway perturbations, such as oxidative phosphorylation, along with additional pathways that present interesting opportunities for further investigation (**Pg 14, line 350 – 361**).

In summary, our revised manuscript eliminates the ambiguity introduced by the original PFVI formulation and provides a clearer, statistically supported, and more interpretable framework for identifying pathway-level perturbations. All relevant codes, analyses, and documentation have been updated and are available on Github (*3_flux_analysis/pathway_level_flux_analysis*).

4) Despite the significant improvements, several figures and supplementary tables still suffer from quality and clarity issues. For example, Supplementary Figure 12 has unreadable column labels; Supplementary Data 2a includes #VALUE! errors in the Excel file; Supplementary Figure 15 and 38, as well as Supplementary Data 2b, also require attention. Please see additional issues detailed in the minor comments below.

We thank the reviewer for carefully noting the issues with figure and supplementary file clarity. In the revised submission, we thoroughly reviewed all figures and supplementary datasets to ensure accuracy and readability.

For the supplementary figures, we rotated the column names in **Supplementary Fig. 1h** (previously Fig. 12) to improve legibility, removed the unnecessary metabolite dendrogram from the previous Supplementary Figure 15, and adjusted font sizes in **Supplementary Fig. 5b** (previously Fig. 38) for legibility. We also re-evaluated all remaining supplementary figures to confirm that labeling were legible.

For the supplementary data, we corrected unintended “#VALUE!” errors in **Supplementary Data 2a**, which arose from normalizing raw data containing NA values (representing undetected metabolites). These entries were replaced with zeros, consistent with downstream analyses performed in MATLAB and R. In **Supplementary Data 2b**, we removed redundant summary sections and added a clarifying note

explaining that metabolites labeled with “neg” (e.g., NADH-neg, CDP-neg) were detected in negative ion mode, as described in the Methods (“Intracellular metabolite extraction and LC/MS analysis”).

We appreciate the reviewer’s attention to these details, which helped improve the overall quality and presentation of the supplementary materials.

MINOR COMMENTS

1) Figure 2f (pathway overlap between GEMs and metabolomics): It appears the authors use KEGG pathways as an intermediary to match GEM activity with metabolomics data. Why is this done indirectly through KEGG? Moreover, the method used to map GEM reactions to KEGG pathways is not clearly explained.

We thank the reviewer for this insightful comment. Our pathway comparison was motivated by having already quantified perturbed pathways from both the GEM (**Fig. 1f**) and metabolomics analyses (**Fig. 2e**). We initially attempted to directly compare enriched metabolite sets but found it challenging to systematically quantify metabolite-level enrichment using GEM simulations. We also considered using metabolite demand functions (Lewis and Kemp, *Nature Comm.*, 2021) but found that more investigation was needed to apply this methodology.

To clarify, GEM reactions were not directly mapped to KEGG pathways. Instead, we matched enriched KEGG pathways identified from metabolomics to the GEM’s subsystem annotations, which are largely based on KEGG pathway terminology. Because some subsystem names did not exactly correspond to KEGG pathway names, we manually curated a one-to-one mapping table between KEGG pathways and GEM subsystems to ensure consistent naming (now provided as **Supplementary Data 2e**). For clarity, we also added **Supplementary Note 1** describing the step-by-step pathway overlap methodology.

2) Figure 3b: The numbers and parenthetical expressions in the panel are not explained in the legend. Please clarify.

We thank the reviewer for this comment. We have revised the legend of **Supplementary Fig. 3a** (previous Fig. 3b) to clearly explain the numerical and parenthetical annotations. In the figure, we also indicated that arrows represent average flux values, while parenthetical expressions indicate minimum and maximum flux values.

Additionally, to improve the clarity and focus of the revised main results (**Pg 9, line 207 – 248**), we moved this figure to the **Supplementary Figure 3a**. This change was made because the figure does not depict all fluxes associated with relevant metabolites (the corresponding simulation summary are provided in **Supplementary Data 3c–d**) and due to semi-quantitative nature of our flux analyses given the limited accuracy of absolute flux predictions, as described earlier.

3) Page 9:

“Specifically, glucose uptake rate was reduced to about 55%, while fluxes through glyceraldehyde-3-phosphate dehydrogenase...”

This sounds like a circular argument. If glucose uptake was manually reduced by constraining the upper bound, then observing a reduction in glycolytic flux may simply reflect that constraint. It may still be a valid observation if the flux remains well below the new boundary, but this should be clarified (See Major Comments 1 and 2).

We thank the reviewer for highlighting the need to clarify the relationship between imposed model constraints and observed changes in simulated fluxes. In the revised manuscript, we now clearly distinguish between reactions whose flux changes result directly from imposed constraints (**Pg 9, lines 211–215**) and those identified as model-predicted perturbations through comparisons across pFBA, FBA, and FVA-sampling (**Pg 9, lines 218–230, Fig. 3a–c**). We believe these revisions resolve the circularity concern and clarify the distinction between constrained and model-predicted flux changes.

4) Page 9:

“Overall, TCA cycle fluxes were decreased by approximately 50% in the HSD-muscle-GEM, consistent with decreased TCA flux observed in diabetic muscle.”

This does not appear consistent with Figure 3b, which shows an increase in upper TCA cycle flux. Additionally, please explain where excess α -ketoglutarate goes and where the additional malate comes from, as the alternate pathways consuming or producing these intermediates must account for any internal flux imbalance in the cycle.

We thank the reviewer for this helpful comment. We recognize the potential confusion, as we indicated a 50% reduction in TCA cycle fluxes while the flux map showed increased fluxes in upper-cycle reactions but decreases in lower-cycle reactions. To clarify, 50% reduction was calculated by comparing the average flux across 13 TCA cycle-associated reactions between the conditions (**Supplementary dataset 3c**, and

now **Supplementary Fig 3a**). We added **Supplementary Note 2** to describe the detailed analysis about the flux changes in central carbon metabolism.

Additionally, we evaluated all production and consumption rates involving α -ketoglutarate, malate, fumarate, and citrate, in mitochondria, and confirmed that the associated fluxes remain balanced, supporting steady state assumption used in flux analyses (now included as **Supplementary Data 3d**). Additional α -ketoglutarate and malate were readily exchanged via multiple transporters and some other reactions. We did not include all reactions in the flux map due to space issues, but clarified in the **Supplementary Fig 3a** legends.

5) Page 10:

“Comparison of these four reactions revealed approximately 2-fold increase in NADH generating fluxes and a 2-fold decrease in NAD⁺ regenerating fluxes (Fig. 3f and Supplementary Fig. 27) ... confirming the perturbed NADH metabolism.”

This reasoning is problematic. By mass balance, any NADH generation must be matched by NAD⁺ regeneration elsewhere in the network. How, then, is this imbalance quantified and connected to the NADH/NAD⁺ ratio? Are the authors referring to net electron flow (e.g., from NADH to other carriers)? Furthermore, focusing on just a couple reactions is difficult to justify when the rest of the network may compensate.

To assess differences in the network's capacity to extract electrons, the authors could add a demand reaction (NADH → NAD; in mass and charge balanced form and not in reverse direction as in the original submission) and maximize its flux across conditions.

We thank the reviewer for identifying the potential issues in our original reasoning. We agree that explaining perturbed NADH metabolism by evaluating only a few selected reactions can be problematic, especially since NADH production and consumption must be balanced under steady-state assumption in flux balance analysis.

In the revised manuscript, we re-evaluated our analysis of NAD(H)-dependent fluxes at the systems-level. First, we confirmed that total NADH production and consumption fluxes remain balanced within the model (**Supplementary Data 3j**). Second, rather than focusing on a few major reactions, we profiled and evaluated flux changes of all active NADH-dependent reactions (**Fig 3c**). Third, we focused on identifying commonly perturbed reactions consistently detected by FBA, pFBA, and FVA-sampling analyses, highlighting candidate reactions potentially relevant to perturbed NADH metabolism (**Pg10, line 231 – 238**).

Fourth, as suggested, to quantify the network's maximal NADH production capacity, we introduced a total NADH demand reaction ($\text{NADH} \rightarrow \text{NAD}^+$) encompassing all relevant compartments and maximized its flux (**Fig 3d; Pg 29, Methods:** "Evaluation of maximum NADH production capacity"). This analysis revealed a 27% reduction in maximal NADH production in the HSD-muscle-GEM (**Fig. 3e; Pg 10, 238 – 248**). While this does not directly represent the cellular NAD^+/NADH ratio, it indicates impaired NADH turnover, consistent with our experimental observation of a decreased NAD^+/NADH ratio in thoracic muscle of w^{118} flies fed with HSD (**Fig. 3f**). Moreover, we characterized the altered contributions of individual NADH-producing reactions under this objective (**Supplementary Fig. 3k**). All relevant codes, analyses, and documentation have been updated and are available on Github (3_flux_analysis/NADH_reaction_analysis and NADH_damend_analysis).

Together, these revisions clarify NAD(H)-dependent flux alterations at the network level and remove reliance on a limited set of reactions. We believe this revised approach directly addresses the reviewer's concerns and strengthens the interpretation of perturbed NAD(H) redox metabolism based on our flux analyses.

6) Figure 4c and the like: Normalizing everything with NSD eliminates valuable information. Why not normalize both NSD and HSD with their respective Glc uptake?

We thank the reviewer for this helpful suggestion. While normalization to glucose uptake rate is commonly used in metabolic flux analysis, where fluxes are estimated from fractional labeling data using least-squares methods, such normalization is less straightforward for direct fractional labeling data due to differences in data scale and is, to our knowledge, not commonly performed (Antoniewicz, *Experimental & Molecular Medicine*, 2018). Yet, we normalized labeling fractions to our measured glucose uptake rates and confirmed that the trends were generally maintained (**Supplementary Data 4b-c**). For example, lactate and fumarate remained significantly altered regardless of normalization method.

Additionally, following the reviewer's suggestion, we have now included the original, unnormalized fractional labeling data in the main figure, which improves clarity and enables direct comparison of relative and absolute labeling patterns (**Fig. 4c and d left panels**, previously shown in the Supplementary figures).

7) Supp. Fig. 31: How can downstream metabolites show higher fractional labeling than glucose, the uniformly labeled carbon source used in the experiment?

We thank the reviewer for this insightful observation. While downstream metabolites are not expected to exceed the fractional labeling of input tracer under *in vitro* or injection or infusion conditions, we observed that *in vivo* feeding experiments often resulted in higher labeling in downstream metabolites.

We attribute this to the complexity of the *in vivo* system, where prolonged feeding (5 days to reach pseudo–steady state in our case) enables extensive metabolite exchanges among tissues. Additionally, a fraction of ingested tracers is metabolized in gut and other tissues, generating labeled downstream metabolites that can subsequently circulate and be taken up by muscle. This inter-organ metabolite exchange likely contributes to the higher apparent labeling of downstream metabolites relative to the input tracer. To account for these effects, we normalized the fractional labeling data to the input tracer and compared values relative to the control condition, as shown in **Fig. 4c and d**.

Although direct tracer injection could mitigate some of these effects, we considered this approach too invasive for *Drosophila* and likely to induce stress responses that might confound measurements. Therefore, we adopted the physiologically relevant feeding approach, which is widely used and validated in *Drosophila* isotope tracing studies (Li et al., *PNAS* 2017; Parkhitko et al., *PNAS* 2021; Jouandin et al., *Science* 2022).

8) Fig. 6g: Black arrow means Flux is ND. What does it mean for a model prediction? Please clarify.

We thank the reviewer for noticing this point. We updated our figure panel and the legend to clarify the black arrow's ND, which means that reactions were not detected in our muscle-specific GEM (now **Fig. 6f**). This also highlights the current limitation of the muscle-GEM for further investigation in fructose/sucrose metabolism, as some reactions, such as sorbitol dehydrogenase, are missing. We noted this as an area for future model refinement to improve coverage of this pathway and flux predictions (**Pg 16, line 445 - 447**).

9) Page 15: "Consistent with this, our flux analysis predicted muscle could catabolize fructose and showed a significant decrease in the activity of Hex-A, a *Drosophila*

hexokinase capable of phosphorylating fructose to fructose-6-phosphate⁷⁵ in response to HSD. This aligned with the accumulation of upstream metabolic intermediates, such as sorbitol and trehalose."

An alternative hypothesis is that metabolite levels (rather than enzyme abundance) drive flux in high-sugar conditions. In that case, low enzyme expression (inferred from mRNA) does not necessarily indicate low flux. This complicates interpretation of perturbed states. Do the authors have data to rule out this alternative?

We thank the reviewer for raising this important alternative hypothesis. We agree that higher concentrations of upstream metabolites can increase flux through mass action even when enzyme is at low abundance or post-translationally modified. Since we have not experimentally ruled out this possibility, we have revised our discussion to describe fructose metabolism as 'perturbed' rather than to describe its flux as 'decreased'. We also acknowledged this alternative hypothesis and noted that future fructose isotope tracing experiments would help resolve the direction and magnitude of these flux changes (**Pg 16, line 427 - 430**).

10) Supp. Data 1c: What is the type and unit of the data presented in this table?

We thank the reviewer for pointing out this detail. **Supplementary Data 1c** contains the tissue-specific gene expression matrix, where the values represent normalized gene expression levels expressed in counts per million (CPM). For clarity, we have added a detailed description of the generation and normalization of this data, including the units, in the Methods section (**Pg 19, line 493 – 498**).

11) Page 18, last paragraph: It is not clear what was achieved with *getINITModel2*().

We thank the reviewer for this comment. We agree that it was not clearly described in the original version. We have clarified in the revised text that *getINITModel2* was used to reconstruct tissue-specific GEMs for each *Drosophila* tissue and moved it to the earlier paragraph for clarity (**Pg 19, line 492 – 493**). The description of input gene expression data in the last paragraph was also moved accordingly (**Pg 19, line 498 – 501**). The last paragraph now focuses solely on the CORDA method, avoiding potential confusion between the two reconstruction methods.

12) Pages 25.26: "This formulation was used here as a general proxy for basal cell maintenance, including protein turnover, membrane lipid remodeling, and nucleotide recycling. These processes remain active and physiologically relevant to tissues like muscle^{70,95,96}. Thus, rather than using energy and redox demands as sole objective functions, which could bias the solution space or do not recapitulate other core metabolic activities, we adopted this biomass objective function to capture broader metabolic requirements. We confirmed the flux through this reaction was approximately 1.1 mmol/gDCW/hr, which is orders of magnitude lower than the major glycolytic and TCA cycle fluxes, indicating that the objective function serves as a minimal biosynthetic demand (Supplementary Fig. 25, Supplementary Data 3h)."

I do not understand the reasoning in flux comparisons. Biomass contains pools such as TAG. For example, fatty acid synthesis required for biomass may translate into a very large energetic demand, depending on the biomass composition.

We thank the reviewer for this important comment. We agree that our original justification for the biomass objective function was not clearly articulated. The reviewer correctly points out that simply comparing the low biomass flux value to the higher fluxes in glycolysis and the TCA cycle can be misleading, as the low flux of the biomass reaction can still represent a substantial energetic and precursor demand on the metabolic network.

To address this point, we have revised the text to remove the flux comparison between biomass and glycolytic or TCA cycle fluxes. We also clarified that the energetic and precursor requirements for the biomass are not reflected in a single flux value, but instead distributed among various fluxes throughout the metabolic network (**Pg 27, line 732 – 735**).

13) Page 28: "For visualization of the predicted structures we utilized ChimeraX (PMID: 37774136)."

There appears to be a formatting issue with this reference.

We thank the reviewer for noting this formatting issue. We have corrected the ChimeraX citation to follow the journal's reference style by replacing "(PMID: 37774136)" with the appropriate reference number (112) and corresponding full citation in the reference list (**Pg 32, line 872; Pg 44, line 1212**).

14) Page 30: “To evaluate the statistical significance of pathway-level flux changes between NSD and HSD conditions, we performed a two-sample Z-test using flux distributions obtained from sampling analysis, following an approach described in a previous study¹⁰⁵.”

However, the bibliography does not contain 105 references. Please correct this.

We thank the reviewer for catching this citation error. The reference number was inadvertently omitted during renumbering. We have now corrected it to include the appropriate citation for the study describing the two-sample Z-test on flux distributions (**Pg 29, line 791; Pg 30, line 831**, with new reference of 101 in the revised manuscript). This reference refers to *Bordel et al.*, “Sampling the Solution Space in Genome-Scale Metabolic Networks Reveals Transcriptional Regulation in Key Enzymes,” *PLOS Computational Biology* (2010).

15) Page 30: “where \bar{X} represents the mean pathway flux”

It is unclear how this average is computed. For example, in glycolysis, flux doubles mid-pathway due to carbon splitting (from 6C to 3C intermediates). How is this handled when averaging across a pathway?

We thank the reviewer for this insightful comment. The pathway flux was defined as the average of the non-zero flux magnitudes within each pathway, with equal weights assigned to each reaction. (**Pg 30, line 827 – 838, Methods**: “Pathway-level flux analysis”).

We acknowledge that our current approach assigns equal weights to reactions when averaging flux magnitudes, which may overlook the relative stoichiometric significance of individual reactions. Future work should focus on weighting reactions by stoichiometric coefficients to provide a more nuanced and accurate pathway-level flux metric.

Despite this limitation, our analysis involved comparing the same pathway between different conditions (NSD vs. HSD), where the stoichiometry remain consistent for each reaction. Thus, intrinsic stoichiometric features, such as flux doubling, are held constant and do not bias relative comparisons between the conditions.

Reviewer #3 (Remarks to the Author):

Reviewer #4 (Remarks to the Author):

Overview:

This manuscript leverages single-nuclei gene expression data to infer tissue-specific genome-scale metabolic models in *Drosophila*. These are constraint-based models that enable prediction of the flux through all metabolic reactions for a given tissue. The authors first briefly compare features of the resulting metabolic networks (network size and structure, ability to perform defined metabolic tasks, representation of metabolic subsystems). The muscle-specific model is used to explore the effects in the context of a high sugar diet, and metabolite profiling and heavy-labeled isotope tracing are used to validate some model findings.

Overall, the revised manuscript is much improved. The paper is more streamlined and better demonstrates the utility of tissue-specific GEMs. The methods are well laid out and explained. The results presented are clear, and the title and introduction align with the main message of the paper. Only a few smaller issues remain.

We thank the reviewer for the positive summary of our work. We have carefully reviewed the specific comments below and further improved the clarity, organization, and presentation of the work.

Specific comments:

- The authors have used experimental methods to confirm several model predictions. This is laudable and nicely bridges communities using computational and experimental methods to study metabolism. However, the authors have not sufficiently emphasized what computational modeling using GEMs uniquely offers above and beyond experimental approaches. This should be better highlighted. This is an important point to address.

We thank the reviewer for highlighting this important point. In the revised manuscript, we added an additional paragraph in the Discussion that emphasizes the unique advantages of our tissue-specific GEMs in *Drosophila* (Pg. 15, line 383 – 391). We also streamlined the *Discussion* to emphasize the advantages of our GEMs beyond experimental approaches.

- More justification is needed for the perturbation-based sensitivity analysis. In the first submission, the upper flux bound was decreased by 10%. Here, 20% is used. Either seems arbitrary and should be justified why that perturbation is meaningful. I could not find any support in the paper cited (ref. 99).

We thank the reviewer for highlighting the need to justify the perturbation magnitude used in our sensitivity analysis. We acknowledge that in the previous version, the perturbation values were somewhat arbitrary, and the supporting reference was inadvertently misplaced (now ref. 102, 103 and 104).

To address this, we thoroughly re-evaluated the sensitivity analysis with detailed methodological clarification (**Pg 30, Methods**: “Sensitivity analysis”). Specifically, instead of uniformly decreasing flux bounds, we now apply perturbations directly to the median flux values derived from flux variability and sampling analysis, ensuring that the perturbations are exactly proportional to the baseline fluxes. Additionally, we systematically varied perturbation magnitudes (5%, 10%, 30%, and 50%) to investigate the effect of perturbation magnitudes rather than focusing only one perturbation (**Fig 3g and h**).

While the earlier sensitivity analysis revealed that GADPH showed the highest sensitivity and potentially served as a rate controlling step, the updated analysis revealed that GAPDH is not a sole rate-controlling enzyme as initially hypothesized, but the control is likely to be distributed among several other enzymes, such as Aldo, Pgk, and Gapdh, and varies with the perturbation magnitude under this condition (**Fig 3g-h; Supple Fig. 3m; Pg 10-11, line 250 – 271**). We also made a separate result section dedicated to this updated sensitivity analysis. All relevant codes, analyses, and documentation have also been updated and are available on Github (*3_flux_analysis/Sensitivity_analysis*)

We believe that these methodological improvements address the reviewer’s concern and enhance the rigor and interpretability of our sensitivity analysis.

- The number of supplementary figures grew by nearly 10-fold, from 6 to 56! It is the responsibility of the authors to provide a well-curated story – this task has not been met if there are 56 supplementary figures. I believe the authors must reduce this number to only include those that are absolutely needed to help tell the story (rather than adding figures that are “nice to have”). This is a critical point to address.

We thank the reviewer for this important comment and agree that the supplementary figures should remain concise and well-curated. In the previous revision, we expanded

individual supplementary figures in response to another reviewer's request to improve figure clarity and legibility

In the current revision, we consolidated multiple individual supplementary figures into unified composites, **Supplementary Fig. 1–6**, while maintaining visual clarity and detail. Although most supplementary figures now contain fewer subpanels, **supplementary figure 3** includes additional panels due to additional relevant analyses done in this revision.

We believe this reorganization improves compactness and readability without loss of key information.

Evaluating the response to the points raised by Reviewer 5:

In my reading, most of the comments from Reviewer 5 were addressed appropriately. I believe major points related to comparing to experimental data/observations were useful. Additionally, the changes to the manuscript to remove flux distributions and Vmax calculations made some of the reviewer points no longer relevant.

List of metabolic tasks. This response is still unclear. The authors should mention all essential tasks (57 from human metabolic task list and 219 metabolic functions) in a single place within the methods. As it stands, these two lists are described in different locations in the manuscript. Furthermore, it is not clear whether the 57 is part of the 219. This requires greater clarity from the authors.

We thank the reviewer for the comment and acknowledge that the methodological details were previously less clear.

To clarify, the 57 essential metabolic tasks were used only for the reconstruction of tissue-specific GEMs, whereas the 219 metabolic tasks represented a separate and expanded set used for metabolic task analysis. We added details in method section, describing the curation of the 219-task list and indicating that the 57 essential tasks and 219 tasks were used for different purposes (**Pg 20, Method: "Metabolic Task Analysis", line 562 – 571; Supplementary Dataset 1d and k**).

Specifically, after reconstructing the tissue-specific GEMs using the original 57 tasks, we identified an updated and more comprehensive version of metabolic task list from recent literature (Richelle et al. *Cell Reports Methods*, 2021), which refined previously published tasks and improved the task category by additionally adding system and subsystem descriptions. Using this list as a basis, we further added eight *Drosophila*-related trehalose biosynthesis tasks and 16 energy and carbohydrate metabolism tasks derived from the earlier Human-GEM framework, resulting in a comprehensive set of 219 metabolic functions. Since the 57 essential tasks were always expected to pass as a part of model reconstruction process, we found using this separate list of 219 metabolic tasks remains appropriate for metabolic task evaluation across tissue-specific GEMs.

We clarified this distinction in the updated metabolic tasks analysis section to improve overall clarity (**Pg 21, line 570 - 571**).

Experimental validation. The authors can go further with their validation, as comparing enriched pathways (data) with subsystem coverage (model) does not consider if the reactions present carry flux or not, it just checks if the subsystem is there or not. I believe it is important to perform the same model predicted enrichment after flux values are calculated. It is not clear that the predictions and FEA would agree.

We thank the reviewer for this valuable suggestion. We agree that evaluating subsystem coverage, which only measures the presence of reactions, does not provide information about the fluxes through those pathways.

Performing flux-based pathway enrichment analysis across all tissue-specific GEMs requires well-defined tissue-specific objective functions and environmental constraints. In our preliminary tests, applying a single global objective function without detailed constraints generated high variability and false-positive flux predictions, likely because each tissue has distinct metabolic objectives and environmental constraints. Thus, establishing such tissue-specific constraints requires comprehensive experimental data on parameters such as *in vivo* nutrient availability, metabolite exchange rates, and energy demands, which are currently limited for *Drosophila*. This is an important area for future investigation and we added such limitation in our discussion (**Pg 16, line 437 – 445**).

Despite this limitation, we demonstrated that flux-based pathway enrichments were valid in muscle under high sugar diet (HSD). In this revision we quantified differential fluxes and predicted the affected subsystem using muscle-GEM under HSD (**Fig 3a-b; Pg 9, line 218 - 230**). Additionally, we also showed that muscle-GEM predicted

perturbed glycolytic fluxes, which were confirmed through *In vivo* ¹³C-glucose tracing study (**Fig 4; Pg 11-12, line 273 – 305**). Moreover, using pathway-level flux comparisons, we also predicted perturbed fructose metabolism and confirmed through targeted metabolomics (**Fig 6; Pg.14, line 350 – 374**). Together, these results support the validity of our flux-based pathway analysis in the muscle-GEM.

We acknowledge that extending this flux-based pathway enrichment validation across other tissues represents an important next step to further strengthen GEM-based metabolic analyses in other tissues. We intend to pursue this direction as relevant data and methodologies become available.

REVIEWERS' COMMENTS

Reviewer #2 (Remarks to the Author):

The authors addressed all of my previous comments and I agree that the manuscript has improved further. Below are two minor issues with the updated sections:

1. NADH -> NAD conversion reaction: I could not find this in supplementary tables. The authors should make sure this reaction is mass-balanced using a proton, so that the observed effect is not due to proton consumption (but not charge-balanced, since the reaction represents electron demand).

We thank the reviewer for pointing out this important detail. In fact, we used the NADH demand reaction that was mass balanced by including a proton, represented as $\text{NADH} \rightarrow \text{NAD}^+ + \text{H}^+$. Specifically, the objective function of NADH demand reactions across compartments was defined as:

, where [c], [m], and [p] refer to the cytosol, mitochondria, and peroxisome, respectively.

We have now clarified this in the Methods section ("Evaluation of maximum NADH production capacity") and updated the reaction equation by including the proton in Fig. 3d. The corresponding optimal flux values of NADH -> NAD + H is shown as bar graph in Figure 3e and Source Data Fig 3e.

2. Methods, "Quantification of pathway overlap using weighted Jaccard index": This section is rewritten but is still unclear. In particular, the explanation on $w_B(i)$ (normalized weight for pathway i) is confusing. For example, in the equation $w_B(i) = 1/|B|$, there is no i on the right hand side. What does this mean? Also, why is B in absolute value brackets if it is a number of enriched pathways, which should always be positive?

We thank the reviewer for this valuable comment. We agree that this method required more detail and now expanded the method (Methods: Quantification of pathway overlap using weighted Jaccard index). Specifically, we provided details of $w_B(i)$ as follows:

$w_B(i)$ refers to the normalized weights assigned to GEM-based enriched pathways. Since the enriched pathways identified from GEM do not have statistical significance values, each pathway i in set B is assigned an equal weight, calculated as $w_B(i) = 1/n_B$, where

n_B is the number of enriched pathways in that tissue-specific GEM (set B). For any pathway i that is not part of set B (e.g., $i \in A \cup B$ and $i \notin B$), $w_B(i) = 0$. The sum of all $w_B(i)$ equals 1.

We originally used $|B|$ as standard set notation for the number of elements in set B, but recognize that this may be misleading and have replaced it with the clearer notation n_B . We believe this update resolves the ambiguity and improves clarity.